# EdiVal-Agent: An Object-Centric Framework for Automated, Fine-Grained Evaluation of Multi-Turn Editing

**Tianyu Chen**[1,3,‡,*]**, Yasi Zhang**[2,*]**, Zhi Zhang**[2]**, Peiyu Yu**[2]**, Shu Wang**[2]**, Zhendong Wang**[3]**,
Kevin Lin**[3]**, Xiaofei Wang**[3]**, Zhengyuan Yang**[3]**, Linjie Li**[3]**, Chung-Ching Lin**[3]**, Jianwen Xie**[4]
**Oscar Leong**[2,†]**, Lijuan Wang**[3,†]**, Ying Nian Wu**[2,†]**, Mingyuan Zhou**[1,3,†]

[1]The University of Texas at Austin [2]University of California, Los Angeles
[3]Microsoft AI Superintelligence [4]Lambda, Inc
[*]Equal contribution. [†]Equal advising. [‡] Work done during an internship at Microsoft
Correspondence to `tianyuchen@utexas.edu` and `yasminzhang@ucla.edu`.

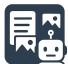 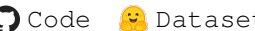

**Website**: `https://tianyucodings.github.io/EdiVAL-page/`
 Code    Dataset

## ABSTRACT

Instruction-based image editing has advanced rapidly, yet reliable and inter-pretable evaluation remains a bottleneck. Current protocols either (i) depend on paired reference images—resulting in limited coverage and inheriting biases from prior generative models—or (ii) rely *solely* on zero-shot vision–language mod-els (VLMs), whose prompt-based assessments of instruction following, content consistency, and visual quality are often imprecise.

To address this, we introduce **EdiVal-Agent**, an automated and fine-grained evalu-ation framework grounded in an object-centric perspective, designed to assess not only standard single-turn but also multi-turn instruction-based editing with preci-sion. Given an input image, **EdiVal-Agent** first decomposes it into semantically meaningful objects, then synthesizes diverse, context-aware editing instructions while dynamically updating object pools across turns. These two stages enable two novel object-centric metrics tailored for multi-turn evaluation and one global metric of visual quality: 1) EdiVal-IF, which measures instruction following by combining open-vocabulary object detectors for symbolic checks with VLMs for semantic verification on detector-guided crops; 2) EdiVal-CC, which evaluates content consistency by calculating semantic similarity of unchanged objects and background using the evolving object pools; and 3) EdiVal-VQ, which quantifies changes in overall visual quality with human preference models.

Instantiating this pipeline, we build **EdiVal-Bench**, a multi-turn editing bench-mark covering 9 instruction types and 16 state-of-the-art editing models spanning in-context [1], flow-matching, and diffusion paradigms. Our results show that *See-dream 4.0* achieves the best overall performance, offering the strongest balance of instruction following, content consistency, and latency. *GPT-Image-1.5* clearly improves over *GPT-Image-1*, especially in content consistency across turns, while *Nano Banana 2* consistently outperforms *Nano Banana* in instruction following and overall score,. Among flow-matching models, *FLUX.2-max* is the strongest baseline, whereas *Qwen-Image-Edit* performs well on the first turn but degrades sharply in later turns, indicating strong exposure bias in multi-turn editing. We demonstrate that **EdiVal-Agent** can be used to identify existing failure modes, thereby informing the development of the next generation of editing models. Our code is available at `https://github.com/TianyuCodings/EdiVal`.

---

[1]In this paper, we label certain closed-source models—GPT-Image-1/1.5, Nano Banana 1/2, and Gemini 2.0 Flash Image—as *in-context*, since they are integrated into autoregressive language models in the web UI and support **in-context multi-turn editing**.

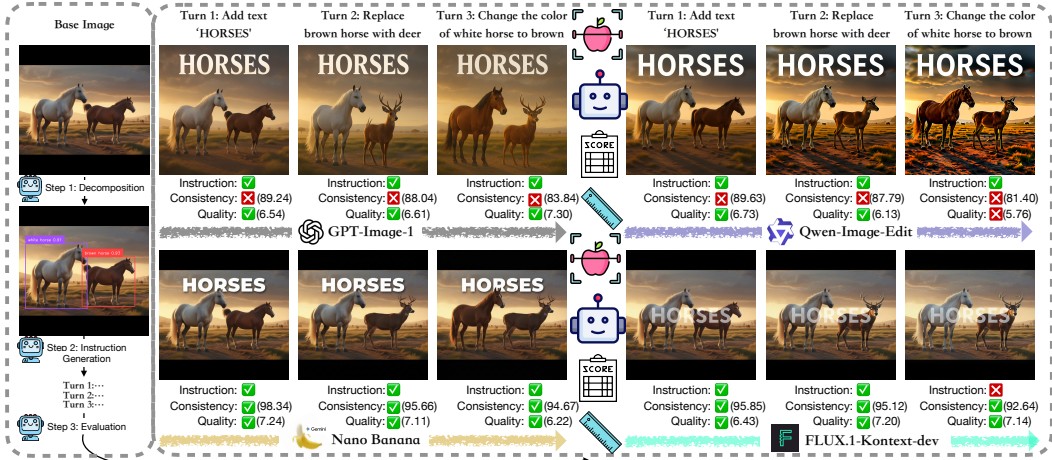

Figure 1: Overview of our workflow and representative model's performance. For visualization, we adopt two thresholds: a consistency score of at least 90 and a visual quality score of at least 6. Details of the automated evaluation pipeline are provided in Figure 2 and Section 2. In multi-turn editing, models exhibit distinct weaknesses: *GPT-Image-1* struggles with content consistency, *Qwen-Image-Edit* underperforms in both visual quality and content consistency, and *FLUX.1-Kontext-dev* lags in instruction following, whereas *Nano Banana* shows no single dominant weakness.

# 1 INTRODUCTION

What truly defines the success of an image editor? At its core, editing requires making targeted, instruction-driven changes while preserving contextual consistency and perceptual realism—often across multiple refinement turns. Yet current evaluation practice struggles to capture this multi-faceted objective.

When ground-truth edited images are available, a common strategy is to compare model outputs against these references (e.g., MagicBrush Zhang et al. (2023), UltraEdit Zhao et al. (2024), AnyEdit Yu et al. (2025), EmuEdit Sheynin et al. (2024)). Typical metrics include pixel-level distances (e.g., L1/L2) and semantic similarities (e.g., DINO Caron et al. (2021) and CLIP Radford et al. (2021)). While informative, such metrics suffer from two structural issues: (i) the space of acceptable edits is inherently large, whereas a single reference provides only one realization; and (ii) references are frequently synthesized by existing editing models (e.g., Prompt-to-Prompt Hertz et al. (2023), SDXL Podell et al. (2024), DALLE-2 Ramesh et al. (2022)), thereby importing their biases and limitations into the evaluation itself. Consequently, high reference similarity does not necessarily imply faithful instruction following, preservation of irrelevant content, or aesthetically plausible outcomes.

A complementary line of work employs zero-shot VLMs as interpretable evaluators (e.g., VIEScore Ku et al. (2023), GEdit-Bench Liu et al. (2025), I²EBench Ma et al. (2024), HQ-Edit Hui et al. (2024), Complex-Edit Yang et al. (2025), and ImgEdit Ye et al. (2025)) and queries VLMs about specific aspects of an edit. While VLMs offer holistic, language-mediated judgments, they remain insufficient for precise editing assessment for several reasons. First, for instruction-following evaluation, they are notoriously poor at spatial reasoning Zhang et al. (2025b); Chen et al. (2024); Chang et al. (2025) and are prone to object hallucinations in existence, category, attributes, and relations Bai et al. (2024). These issues together undermine their ability to assess common object-related edit instructions. Second, they have limited sensitivity to pixel-level changes and frequently miss subtle, localized modifications Vo et al. (2025) (e.g., fine structures, small attribute shifts, etc.), which are crucial for evaluating content consistency. Third, since they are predominantly pretrained on natural images rather than synthetic generations, their priors are miscalibrated for artifacts and aesthetics, leading to failures in detecting common generative defects (e.g., extra fingers) and in modeling perceptual "naturalness" Liang et al. (2024); Xu et al. (2023); Ma et al. (2025), which humans are sensitive to. Consequently, VLM-only scoring lacks the precision and reliability required for fine-grained evaluation across instruction following, content consistency, and visual quality. However, we find recently open-source state-of-the-art editing models (e.g., Qwen-Image-Edit Wu et al. (2025a), Step1X-Edit Liu et al. (2025), BagelDeng et al. (2025)) *solely* rely on VLMs for evaluation.

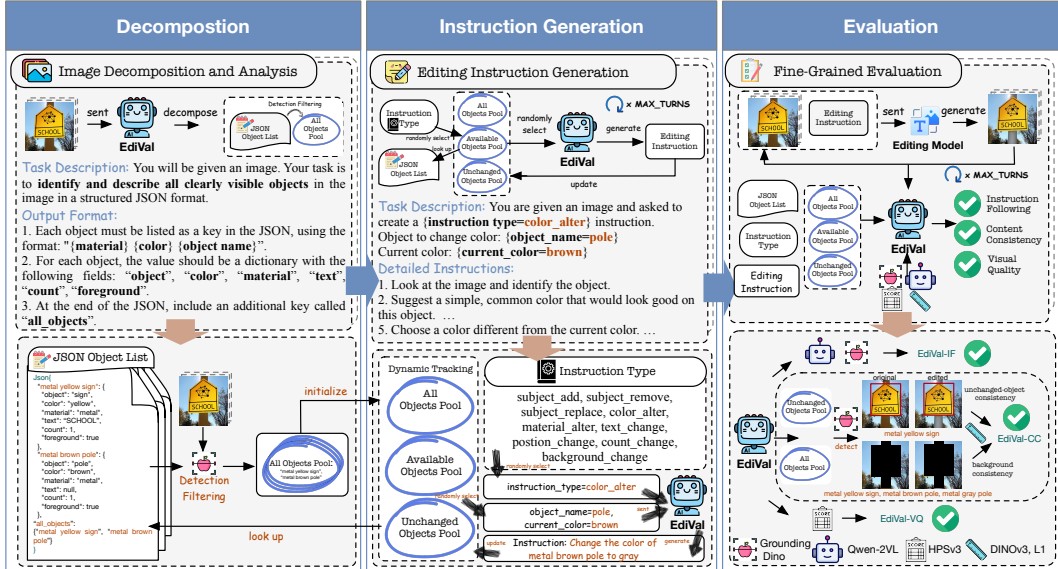

Figure 2: Framework of **EdiVal-Agent**. It first decomposes images into semantically meaningful objects, such as *metal yellow sign* and *metal brown pole*, and identifies their contextual relationships, e.g., they are both in *foreground*. It then generates diverse and proper editing scenarios at scale which are based on the initial analysis, e.g., *Change the color of metal brown pole to gray*. Finally, it systematically evaluates editing model outputs from multiple axes with our proposed metrics: EdiVal-IF, EdiVal-CC, and EdiVal-VQ. Our agentic pipeline is agnostic to the expert tools used and can be readily enhanced with more advanced tools in the future.

To address these challenges, we introduce **EdiVal-Agent**: an automated and fine-grained evaluation agent for multi-turn instruction-based image editing from an object-centric perspective, designed to assess not only standard single-turn but also multi-turn instruction-based editing with precision. As shown in Fig. 2, **EdiVal-Agent** first decomposes it into semantically meaningful objects, then synthesizes diverse, context-aware editing instructions while dynamically updating object pools across turns. These two stages enable two novel object-centric metrics tailored for multi-turn evaluation and one global metric of visual quality: 1) EdiVal-IF, which measures instruction following by combining open-vocabulary object detectors for symbolic checks with VLMs for semantic verification on detector-guided crops; 2) EdiVal-CC, which evaluates content consistency by calculating semantic similarity of unchanged objects and background using the evolving object pools; and 3) EdiVal-VQ, which quantifies changes in overall visual quality with human preference models. We show that EdiVal-IF yields stronger agreement with human judgments in instruction-following evaluation compared to thresholded CLIP directional (CLIP_dir) scores Gal et al. (2022) and using VLMs alone, as evidenced in Sec. 2.5.

Instantiating the agentic pipeline, we curate a new multi-turn image editing benchmark, **EdiVal-Bench**, featuring 9 instruction types and 16 existing editing models—spanning in-context, flow-matching, and diffusion paradigms, across both closed- and open-source systems—conduct fine-grained analyses, and draw actionable insights. Empirically, as demonstrated in Fig. 1 and Tab. 3, *GPT-Image-1* excels at instruction following yet ranks near the bottom in content consistency, whereas *Seedream 4.0* and *Nano Banana* performs strongly on both axes. Besides, open-sourced models like *Qwen-Image-Edit* significantly degrade in instruction following and visual quality when editing turns increase, while *FLUX.1-Kontext-max* and *FLUX.1-Kontext-dev* lags in instruction following. We further contrasts multi-turn editing with single-shot complex prompts Yang et al. (2025), highlighting complementary strengths and failure modes. We hope that our agent pipeline, benchmark, and analyses accelerate the transition of multi-turn editing toward practical applications.

**Key contributions.** 1) *Agent:* **EdiVal-Agent** is a fully automated evaluator that performs object-centric decomposition, generates diverse multi-turn editing instructions, and measures overall editing quality using two object-centric metrics (EdiVal-IF and EdiVal-CC) plus EdiVal-VQ for visual quality. 2) *Benchmark:* using **EdiVal-Agent**, we construct **EdiVal-Bench** with 1,716 instructions

Table 1: Key attributes of open-source edit benchmarks. Note that ImgEdit Ye et al. (2025) does not include multi-turn editing experiments in the paper.

| Benchmark | # Size | Object-centric | Automated | Multi-Turn | Free from Ref. Images | Tools used |
|---|---|---|---|---|---|---|
| EmuEditSheynin et al. (2024) | 3,055 | ✗ | ✓ | ✗ | ✗ | L1, CLIP, DINO |
| MagicBrushZhang et al. (2023) | 1,053 | ✗ | ✗ | ✗ | ✗ | L1, L2, DINO, CLIP |
| AnyEditYu et al. (2025) | 1,250 | ✗ | ✗ | ✗ | ✗ | L1, CLIP, DINO |
| I2EBenchMa et al. (2024) | 2,240 | ✗ | ✗ | ✗ | ✓ | VLM |
| GEdit-BenchLiu et al. (2025) | 606 | ✗ | ✗ | ✗ | ✓ | VLM |
| HQ-EditHui et al. (2024) | 1,651 | ✗ | ✓ | ✗ | ✓ | VLM |
| ImgEdit-BenchYe et al. (2025) | 811 | ✗ | ✓ | ✗ | ✓ | VLM |
| Kontext-BenchLabs et al. (2025) | 1,026 | ✗ | ✗ | ✗ | ✓ | Human Annotation |
| EdiVal-Bench (ours) | 1,716 | ✓ | ✓ | ✓ | ✓ | Detector, VLM, L1, DINO, HPS |

across nine types and three turns on 572 real-world images, with comparisons to prior benchmarks in Tab. 1. 3) *Human agreement:* EdiVal-IF attains 81.3% agreement with human ratings for instruction following, outperforming zero-shot VLMs and CLIP-based baselines. 4) *Evaluation:* we assess 13 editors (diffusion, flow-matching, and close source) along instruction following, content consistency, and visual quality. 5) *Insights:* overall ranking—Seedream 4.0 > Nano Banana > FLUX.1-Kontext-max > GPT-Image-1; the strongest open-source editor, Qwen-Image-Edit, exhibits exposure bias under multi-turn editing. 6) *Artifacts & settings:* we reveal luminance drift across turns, and contrast multi-turn against complex single-shot editing to delineate strengths and weaknesses across model families.

## 2 🤖 EDIVAL-AGENT

### 2.1 OVERVIEW

As illustrated in Fig. 2. The pipeline comprises three stages: (1) *Decomposition* uses a VLM (e.g., GPT-4o; other VLMs are viable alternatives) to extract structured, object-level descriptions—objects, attributes, and relations—enabling symbolic reasoning; (2) *Instruction Generation* produces multi-turn, diverse, compositional prompts by maintaining an explicit object pool and sampling from nine instruction types spanning subject-, attribute-, relational-, text-, count-, and global-level edits; (3) *Evaluation* scores edited images with EdiVal-IF, Edival-CC, and EdiVal-VQ.

### 2.2 STEP 1: DECOMPOSITION

Given an image, a VLM-based agent parses clearly visible foreground objects and returns per-object JSON with fields `object`, `color`, `material`, `text`, `count`, and a boolean `foreground`. Names follow `"{material} {color} {object}"`; unknown fields are omitted; person identity is never recorded (only wearables/accessories). Example: `{"metal yellow sign":{"object":"sign","color":"yellow","material":"metal","text":"SCHOOL","count":1,"foreground":true}}`. An aggregated `all_objects` string concisely lists objects (e.g., "metal yellow sign . metal brown pole"). We apply this stage to GEdit-Bench Liu et al. (2025) (606 images), exclude 34 images with sensitive personal content, and retain 572 images. After extraction, Grounding-DINO validates objects and detects bounding boxes; only reliable detections are kept to seed instruction generation and evaluation. The filtered objects are stored in the `All_Objects_Pool` and later used to initialize three distinct object pools that dynamically track the evolving state of instruction generation.

### 2.3 STEP 2: INSTRUCTION GENERATION

From the decomposed scene, the agent generates multi-turn edits that are grounded in the current object state. The instruction taxonomy (nine types; six categories) appears in Table 2. We maintain three evolving pools at turn $t$: $\mathcal{P}_t^{\text{all}}$ (all objects ever present), $\mathcal{P}_t^{\text{unch}}$ (original objects not edited up to $t$), and $\mathcal{P}_t^{\text{avail}}$ (objects currently editable). With a turn budget `MAX_TURNS`, at each turn the agent (i) selects a type—defaulting to `subject_add` if $\mathcal{P}_t^{\text{avail}} = \emptyset$, otherwise sampling a type not yet used in the chain; (ii) selects object(s) from $\mathcal{P}_t^{\text{avail}}$; (iii) emits a natural-language instruction via GPT-4o referencing those objects and the scene state; and (iv) updates $\mathcal{P}_{t+1}^{\text{all}}$, $\mathcal{P}_{t+1}^{\text{avail}}$, and $\mathcal{P}_{t+1}^{\text{unch}}$ according to the intended edit. When a `background_change` edit ap-

plies at turn $t$, background-consistency scoring is disabled since this turn, and we append "make {objects_in_foreground} unchanged" to the instruction to preserve object-level comparability, where objects_in_foreground = $\{o \in P_t^{\text{avail}} : o.\text{foreground} = \text{true}\}$. The loop is adaptive by expanding/contracting $\mathcal{P}_t^{\text{avail}}$ and naturally compositional. Our default sets MAX_TURNS= 3 (In our implementation, each turn is assigned a distinct instruction type.), though longer chains are easily obtained by allowing repetition or adding types.

Table 2: **Instruction types in EdiVal-Bench** created by **EdiVal-Agent**, grouped by semantic category. Counts are shown per turn (T1–T3).

| Category | Instruction Type | Example Instruction | T1 | T2 | T3 | Total |
|---|---|---|---|---|---|---|
| Subject-centric | subject_add | Add bench on the left of metal red fire hydrant. | 67 | 77 | 93 | 237 |
| | subject_remove | Remove wooden brown door. | 75 | 69 | 61 | 205 |
| | subject_replace | Replace stone gray railing with wooden fence. | 54 | 57 | 55 | 166 |
| Attribute-centric | color_alter | Change the color of metal white airplane to blue. | 56 | 73 | 57 | 186 |
| | material_alter | Change the material of plastic black pen to metal. | 66 | 50 | 72 | 188 |
| Text-related | text_change | Replace the text 'BEARS CONTROL' on cotton black cap with 'WILD PATH'. | 64 | 70 | 54 | 188 |
| Relational | position_change | Change the position of ceramic white cup to right of plastic white laptop. | 52 | 63 | 48 | 163 |
| Counting | count_change | Change the count of fur brown bear to 3. | 73 | 58 | 60 | 191 |
| Global | background_change | Change the background to forest, remain the brown fur bear unchanged. | 65 | 55 | 72 | 192 |

## 2.4 STEP 3: EVALUATION

The first two stages enable two novel object-centric metrics for multi-turn editing evaluation for instruction following and content consistency, respectively, and one global metric for visual quality:

**EdiVal-IF** To evaluate instruction following, we introduce EdiVal-IF, which assesses multi-turn edits by comparing the image from the previous turn, $I^{t-1}$, to the current image, $I^t$. For a given instruction $P^t$ at turn $t$, the score is determined differently for symbolically and semantically verifiable tasks. Symbolically verifiable types ($T_{\text{sym}}$)—such as subject_add, subject_remove, subject_replace, position_change, and count_change—are evaluated using an open-vocabulary object detector $\mathcal{M}_{\text{detect}}$ Liu et al. (2024b). The detector's outputs, including bounding boxes and confidence, are assessed against geometric and logical criteria $\mathcal{F}_{\text{sym}}$ derived from the instruction. For example, for a position_change instruction "Move [A] to the left of [B]", $\mathcal{F}_{\text{sym}}$ verifies that the x-coordinate of A's bounding box $\mathcal{B}$ center is less than that of B in $I^t$, i.e., $\text{center}_x(\mathcal{B}_A^t) < \text{center}_x(\mathcal{B}_B^t)$. In this case,

$$\text{EdiVal-IF}(I^t, I^{t-1}, P^t \in T_{\text{sym}}) = \mathcal{F}_{\text{sym}}(\mathcal{M}_{\text{detect}}(I^{t-1}, I^t | P^t)). \quad (1)$$

Semantically verifiable types ($T_{\text{sem}}$)—color_alter, material_alter, text_change, and background_change—are evaluated with a VLM $\mathcal{M}_{\text{VLM}}$ Yang et al. (2024). To focus the evaluation, the VLM is applied to detector-guided object crops ($I_o$) using instruction-specific templates.

$$\text{EdiVal-IF}(I^t, I^{t-1}, P^t \in T_{\text{sem}}) = \mathcal{M}_{\text{VLM}}(I_o^{t-1}, I_o^t | P^t) = \mathcal{M}_{\text{VLM}}(\mathcal{M}_{\text{detect}}(I^{t-1}, I^t | P^t)). \quad (2)$$

We show that EdiVal-IF achieves superior human agreement (Sec. 2.5). The multi-turn editing success rate is defined as the logical AND of the EdiVal-IF scores across all edits along the chain, whereas the marginal task rate at turn $t$ is defined according to the formulas 1 and 2 provided above.

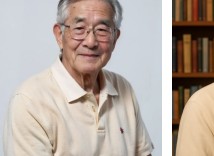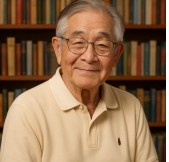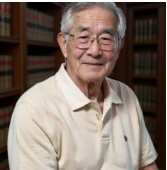

(a) Base image  (b) GPT-Image-1  (c) FLUX.1-max

Figure 3: **Beautification vs. preservation** under the prompt: *"Change the background to a library."* GPT-Image-1 tends to increase HPSv3 via beautification, while FLUX.1-Kontext-max emphasizes fidelity to the input.

**EdiVal-CC** To assess content consistency, EdiVal-CC measures the preservation of non-target content between the base image $I^0$ and the current image $I^t$. Given editing instructions

$P^{1:t}$ from turn 1 to turn $t$, the object pools $\mathcal{P}_t^{\text{unch}}$ and $\mathcal{P}_t^{\text{all}}$ are dynamically updated. Let $\Omega$ denote the entire image area. Using object bounding boxes from the base image ($\mathcal{B}_o^0$) and the current image ($\mathcal{B}_o^t$), extracted by the detector $\mathcal{M}_{\text{detect}}$, the background region is defined as $\Omega_{\text{bg}}^t = \Omega - \bigcup_{o \in \mathcal{P}_t^{\text{all}}}(\mathcal{B}_o^0 \cup \mathcal{B}_o^t)$, i.e., the region obtained by excluding all objects that have appeared. Background consistency is then computed as $s_{\text{bg}}^t = \phi(I_{\text{bg}}^0, I_{\text{bg}}^t)$, where $I_{\text{bg}}^t = \Omega_{\text{bg}}^t \circ I^t$ denotes the background of the image, and $\phi$ is a similarity function such as $L_1$ distance or DINO-based similarity. For unchanged objects, we compute the per-object consistency $s_o^t = \phi(I_o^0, I_o^t)$ for each $o \in \mathcal{P}_t^{\text{unch}}$, and then average them. Formally, the final EdiVal-CC score emphasizes semantic preservation by averaging the feature-level similarities of the background and unchanged objects (see Appendix. M.3 for details):

$$\text{EdiVal-CC}(I^t, I^0, P^{1:t}) = \frac{1}{2}\left( s_{\text{bg}}^t + \frac{1}{|\mathcal{P}_t^{\text{unch}}|} \sum_{o \in \mathcal{P}_t^{\text{unch}}} s_o^t \right). \tag{3}$$

EdiVal-CC aligns with the *intuitive* notion of consistency, providing a precise measurement.

**EdiVal-VQ.** Zero-shot VLMs are not trained for reliable assessment of image quality—particularly artifacts and aesthetics—and we find they are imprecise as scoring functions (see Appx. L). Consequently, we adopt Human Preference Score v3 (HPSv3) Ma et al. (2025) as our visual-quality (VQ) evaluator. In practice, applying preference models to unedited, real photographs often yields relatively low aesthetic scores. We also observe divergent behaviors across editors (See Fig. 3): some (e.g., GPT-Image-1) tend to *beautify* images and increase HPSv3, whereas others (e.g., FLUX.1-Kontext-max) *preserve* the original appearance with minimal aesthetic change. Because aesthetic preference is inherently user- and task-dependent, and beautification can trade off with content consistency (already incorporated into our overall score), we report EdiVal-VQ separately and do not fold it into the aggregate metric.

**EdiVal-O.** We aggregate *Instruction Following* (EdiVal-IF) and *Content Consistency* (EdiVal-CC) into a single overall score. Since both metrics are unit-free and normalized to $[0, 1]$ but capture complementary aspects, we follow prior work and use the geometric mean to balance them and penalize imbalance (Liu et al., 2025; Ku et al., 2023). Formally, EdiVal-O $= \sqrt{\text{EdiVal-IF} \cdot \text{EdiVal-CC}}$.

**Design Scope and Limitations.** We omit `style_change` from our taxonomy because style categories are inherently ill-defined, which makes instruction-following (EdiVal-IF) difficult to evaluate reliably. Extending **EdiVal-Agent** with style-aware recognition is promising future work. After language-based extraction, we validate objects using Grounding-DINO Liu et al. (2024a) and prune low-confidence or ambiguous detections. This stabilizes the object pool and reduces error propagation during instruction generation and IF evaluation. By default, we employ Grounding-DINO as the open-vocabulary detector, Qwen2-VL as the VLM, and DINOv3 Siméoni et al. (2025) as the image feature extractor due to their state-of-the-art performance and open-source availability, which facilitates community use. The agentic pipeline is tool-agnostic and can be readily strengthened by substituting more advanced modules in the future.

## 2.5 Measuring Human Agreement

**Setup.** We conduct human study on edits made by four exemplary models, Step1X-Edit, AnyEdit, Gemini 2.0 Flash and Flux.1-Kontext-dev, on **EdiVal-Bench**, generated by **EdiVal-Agent** as described in Sec. 2.3. In total, we collect 4,576 human annotations of edits. During evaluation, raters were shown the original image, the edited image, and the corresponding instruction, and asked a binary question: *"Evaluate whether the edited image successfully follows the given instruction."*

**Results.** Figure 4 summarizes the findings. EdiVal-IF achieves a human agreement accuracy of **81.3%**, significantly higher than VLM-only (*Qwen-2-VL*, 75.2%), CLIP_dir (65.4%), and other zero-shot VLMs. These results verify that integrating VLMs reasoning with object detection leads to better alignment with human judgment compared to existing methods. The inter-annotator's agreement rate (85.5%) indicates the best performance any evaluation tool can reach.

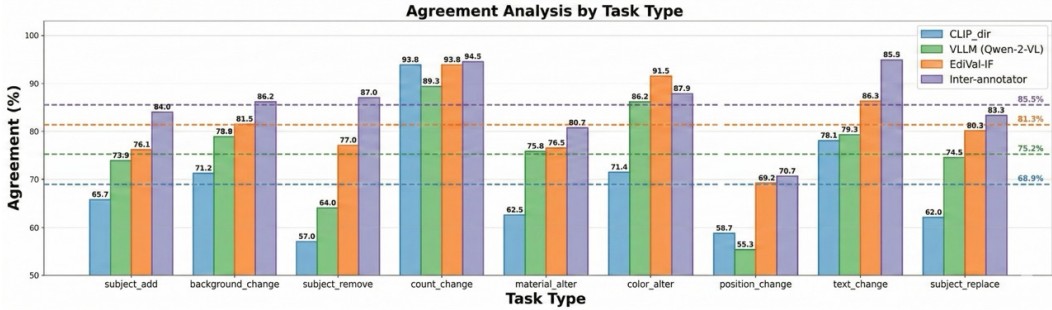

Figure 4: Results of human agreement. Dashed lines represent the average accuracy of each method. EdiVal-IF achieves 81.3% human agreement accuracy, significantly outperforming the VLM (Qwen2-VL) at 75.2% and thresholded CLIP_dir at 65.4%. Note that the CLIP_dir threshold is tuned separately for each task.

We attribute the improvement in instruction-following evaluation to two factors. First, for symbolically verifiable instruction types—subject_add, subject_remove, subject_replace, position_change, and count_change—EdiVal-IF relies solely on Grounding-DINO. It determines the success of an edit by checking object presence/absence, the positions of object centers, and the number of bounding boxes. Results for position_change and subject_remove show that these fixed rules, combined with Grounding-DINO, can significantly outperform Qwen2-VL in edit evaluation. We hypothesize that errors in position_change stem from poor spatial reasoning, while failures in subject_remove are due to hallucinations regarding object existence. Second, semantically verifiable types—color_alter, material_alter, text_change, and background_change—are evaluated using Qwen2-VL combined with Grounding-DINO. The decomposition stage in **EdiVal-Agent** can supports evaluation by localizing text regions, enabling the LLM to reason more precisely about text edits. These findings indicate that EdiVal-IF not only enhances interpretability but also improves the practical applicability of evaluation pipelines in real-world settings that demand human-like understanding. Nonetheless, EdiVal-IF has failure modes, which we document and analyze in Appendix. I.

## 3  BENCHMARKING MULTI-TURN EDITING

Table 3: **Results of multi-turn editing.** EdiVal-IF, EdiVal-CC, and EdiVal-O across three sequential editing turns. Best per column in **dark red**; second-best in lighter red. * indicates close-sourced model, and † indicates open-sourced model.

| Technique | Model | In-Context | Date | Latency (s/img) | EdiVal-IF T1 | EdiVal-IF T2 | EdiVal-IF T3 | EdiVal-CC T1 | EdiVal-CC T2 | EdiVal-CC T3 | EdiVal-O T1 | EdiVal-O T2 | EdiVal-O T3 | Rank |
|---|---|---|---|---|---|---|---|---|---|---|---|---|---|---|
| Unknown | * Seedream 4.0 ⚥ | ✗ | 25.09.10 | 14.55 | **75.93** | 55.58 | **41.59** | 92.51 | 88.03 | 85.86 | **83.81** | 69.95 | 59.76 | 1 |
| | *GPT-Image-1.5 ⚥ | ✓ | 25.12.16 | 35.55 | 75.19 | **55.92** | 40.08 | 94.49 | 91.20 | 88.49 | 84.29 | 71.41 | 59.55 | 2 |
| | *Nano Banana 2 ⚥ | ✓ | 26.03.01 | 23.79 | 73.89 | 54.17 | 38.61 | 93.54 | 90.52 | 88.61 | 83.14 | 70.02 | 58.49 | 3 |
| | *Nano Banana | ✓ | 25.08.26 | 9.20 | 70.70 | 50.66 | 35.35 | 93.91 | 90.48 | 89.48 | 81.48 | 67.70 | 56.24 | 5 |
| | *GPT-Image-1 | ✓ | 25.07.16 | 26.47 | 73.12 | 54.89 | 38.35 | 81.00 | 77.78 | 75.50 | 76.96 | 65.34 | 53.81 | 6 |
| | *Gemini 2.0 Flash | ✓ | 25.02.05 | 8.34 | 68.07 | 45.96 | 28.42 | 90.58 | 85.10 | 80.88 | 78.52 | 62.54 | 47.94 | 8 |
| Flow Matching | *FLUX.2-max | ✗ | 25.12.16 | 36.87 | 75.55 | 55.27 | 39.36 | 92.91 | 88.30 | 85.78 | 83.78 | 69.86 | 58.10 | 4 |
| | *FLUX.1-Kontext-max | ✗ | 25.06.03 | 10.13 | 69.49 | 46.89 | 31.83 | 93.93 | 90.90 | 88.40 | 80.79 | 65.29 | 53.04 | 7 |
| | †Qwen-Image-Edit | ✗ | 25.08.04 | 115.08 | 72.90 | 44.06 | 22.55 | 84.22 | 80.52 | 77.98 | 78.36 | 59.56 | 41.93 | 9 |
| | †Step1X-Edit | ✗ | 25.04.25 | 20.42 | 61.89 | 34.97 | 17.83 | 92.76 | 88.52 | 85.21 | 75.77 | 55.64 | 38.98 | 10 |
| | †FLUX.1-Kontext-dev | ✗ | 25.06.25 | 29.21 | 59.97 | 32.69 | 16.61 | **95.32** | **92.24** | **90.22** | 75.61 | 54.91 | 38.71 | 11 |
| | †OmniGen | ✗ | 24.09.11 | 19.70 | 54.72 | 24.48 | 10.66 | 93.00 | 88.42 | 83.92 | 71.34 | 46.52 | 29.91 | 12 |
| Diffusion | †AnyEdit | ✗ | 24.11.24 | 3.93 | 41.07 | 16.32 | 7.22 | 86.42 | 78.91 | 70.10 | 59.58 | 35.89 | 22.50 | 14 |
| | †UltraEdit | ✗ | 24.07.07 | 3.15 | 51.37 | 17.70 | 6.36 | 86.80 | 84.50 | 82.40 | 66.78 | 38.67 | 22.89 | 13 |
| | †MagicBrush | ✗ | 23.06.16 | 4.08 | 42.31 | 15.73 | 4.90 | 86.96 | 81.26 | 76.86 | 60.66 | 35.75 | 19.41 | 15 |
| | †IP2P | ✗ | 23.12.15 | 4.09 | 37.41 | 10.66 | 2.80 | 76.85 | 68.36 | 60.30 | 53.62 | 26.99 | 12.99 | 16 |

**Summary of Results.**  Table 3 shows that the benchmark has a very tight top tier. *Seedream 4.0* remains the best overall model (Rank 1), combining the strongest instruction-following performance at T1/T3 (75.93/41.59), strong cross-turn consistency (92.51/88.03/85.86), and relatively low latency (14.55 s/img). *GPT-Image-1.5* is a close second and achieves the highest overall *EdiVal-O* at T1/T2 (84.29/71.41), substantially outperforming *GPT-Image-1*; compared with its predecessor,

it improves *EdiVal-CC* by 13.49/13.42/12.99 points and *EdiVal-O* by 7.33/6.07/5.74 points across the three turns, although at higher latency (35.55 vs. 26.47 s/img). A similar trend appears in the Nano series: *Nano Banana 2* consistently improves over *Nano Banana* in instruction following and overall quality, raising *EdiVal-O* from 81.48/67.70/56.24 to 83.14/70.02/58.49, but these gains come with a large speed penalty (23.79 vs. 9.20 s/img) and only marginal changes in consistency. Among the flow-matching baselines, *FLUX.2-max* is the strongest competitor, while *Qwen-Image-Edit* still degrades sharply as the number of turns increases (78.36→59.56→41.93 in *EdiVal-O*), indicating pronounced error accumulation. Overall, newer model versions improve multi-turn robustness and instruction retention, but often by trading off latency. Among open-source systems, *Qwen-Image-Edit* performs well initially (*EdiVal-O* 78.36 at T1) but degrades rapidly with additional turns, likely due to exposure bias as discussed below. We can see that there is a clear gap between the performance of closed-source and open-source systems. With the exception of *Qwen-Image-Edit*, our model rankings exactly match those reported on the Artificial Analysis leaderboard (rank by human vote) as of September 12, 2025; see Appendix. H.

## 3.1 INSTRUCTION FOLLOWING

**Marginal Task Success Rate.** For a given turn, the *marginal task success rate* (Eqns. 1 and 2) is the proportion of prompts for which the edit requested at that turn is successfully executed. By contrast, the *instruction-following* score in Table 3 reports the *multi-turn task success rate* at turn $i$: the logical AND of the EdiVal-IF scores across all edits along the chain. Figure 5 summarizes per-turn performance. High-ranking models—such as Seedream 4.0, Nano Banana, and FLUX.1-Kontext-max—maintain relatively stable EdiVal-IF across turns, even though Seedream 4.0 and FLUX.1-Kontext-max are *not* in-context editors (they do not condition on prior prompts or intermediate images). In contrast, several other models exhibit a clear decline in marginal success as the number of turns increases.

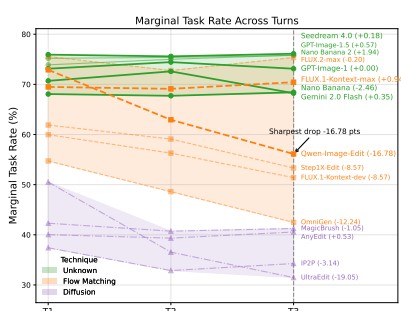

Figure 5: Marginal Task Success rate across turns.

A particularly salient case is **Qwen-Image-Edit**. Although it is the strongest open-source system at turn 1 (EdiVal-O 78.36 vs. 81.48 for Nano Banana), its performance degrades more rapidly over subsequent turns. We hypothesize that this stems from *exposure bias* (Ning et al., 2023; Schmidt, 2019): many single-turn editors are trained to operate on real images and ground-truth inputs rather than on their own previous outputs. When asked to edit their own generations, small distributional mismatches compound across turns, reducing stability; this effect is further aggravated when the model can only attend to a limited history.

**Marginal Task Success Rate Across Instruction Types.** We analyze marginal subtask success rates across turns for different instruction types. The results for Nano Banana are shown in Fig. 6. Other editing models exhibit similar behavior. Nano Banana performs relatively well on attribute-centric tasks such as `color_alter` and `material_alter`, but poorly on `position_change` and `count_change`, indicating weaknesses in spatial and numerical reasoning, respectively.

## 3.2 CONTENT CONSISTENCY

We evaluate two aspects: (i) **unchanged-object consistency** (Fig. 8), which measures whether objects that are not edited up to turn $i$ remain unchanged, and (ii) **background consistency** (Fig. 7), which assesses whether the background remains stable when it is not explicitly modified. When calculating consistency, the grounding box is extracted from the raw input image and applied to all edited images. We therefore choose to report DINOv3 over $L_1$ distance for consistency computation because even small shifts in object location can lead to large variations in pixel-wise $L_1$ loss, even if unchanged objects are well preserved. By relying on DINO features, we ensure that consistency is measured semantically, capturing attributes such as object identity, attributes, and texture, etc. Nevertheless, the consistency scores from DINOv3 remain highly correlated with those computed using pixel-wise $L_1$ loss (See results in the Appendix. N). Based on the results, the most consistent

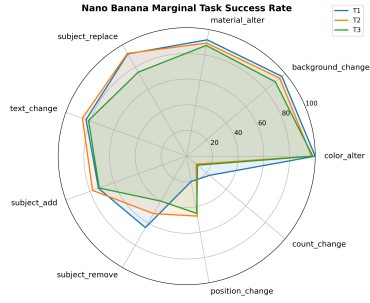

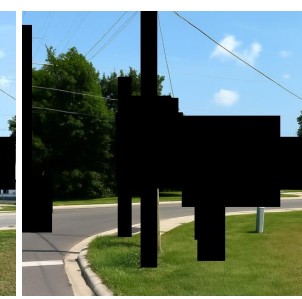

Base image          Qwen-Image-Edit

Figure 6: Marginal task success rate grouped by task types for Nano Banana.

Figure 7: **Illustration of background consistency.** Instruction: "Remove beige brick house." The grounding box is the union of all object regions from the raw and edited images.

editing model is **FLUX.1-Kontext-dev**, followed by **Nano Banana** and **FLUX.1-Kontext-max**. In contrast, **GPT-Image-1** ranks near the bottom, showing notably poor consistency across turns. Representative qualitative examples are shown in Figure 8 and Figure 7.

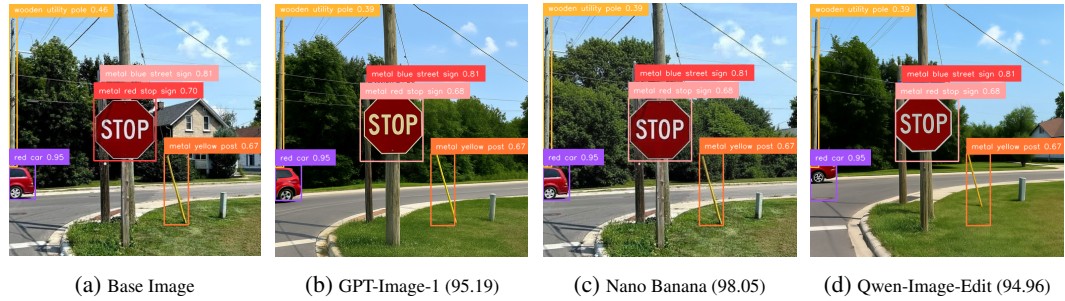

(a) Base Image          (b) GPT-Image-1 (95.19)          (c) Nano Banana (98.05)          (d) Qwen-Image-Edit (94.96)

Figure 8: **Illustration of object consistency.** Instruction: "Remove brick beige house." The grounding box, extracted from the raw input image, highlights the localized region used to compute unchanged-object consistency. The corresponding consistency score is shown in brackets.

## 3.3 VISUAL QUALITY

Besides EdiVal-VQ, we also report the absolute change in visual quality (VQ) relative to the base image: EdiVal-VQ$\Delta_i$ = $\left| \text{EdiVal-VQ}_{\text{turn } i} - \text{EdiVal-VQ}_{\text{base}} \right|$. A smaller $\Delta$ indicates stronger style fidelity to the base image, whereas a larger $\Delta$ suggests greater beautification or stylistic drift. As summarized in Table 4, **GPT-Image-1** achieves the highest aesthetic scores across turns and the largest $\Delta$, indicating a substantial stylistic shift (Fig. 3). In contrast, **GPT-Image-1.5** places greater emphasis on consistency, reducing $\Delta$ from 2.18 to 0.15. For preserving the base image's appearance (small $\Delta$), **Gemini 2.0 Flash** exhibits the least drift, with **Nano Banana** also performing strongly. We provide a low-level exposure statistics analysis in Appendix K.

## 3.4 MULTI-TURN EDITING VS. COMPLEX EDITING

We compare two strategies for composing multiple edits. In *multi-turn* editing, instructions are executed sequentially—apply instruction 1, then apply instruction 2 to the result, and so on. In *complex* editing, we concatenate $C$ instructions into a single prompt and perform one edit ("complex level" $C$, with $C \in \{1, 2, 3\}$). Empirically (Fig. 9), when a model does *not* suffer from exposure bias, multi-turn editing tends to yield higher success rates, consistent with a step-by-step "chain of edits" (analogous to chain-of-thought in reasoning). For instance, Nano Banana benefits from the multi-turn formulation. Conversely, when exposure bias is pronounced, compressing instructions into a single, complex prompt can perform better; see Qwen-Image-Edit in Fig. 9.

Table 4: EdiVal-VQ and EdiVal-VQΔ results across turns. dark red denotes the *best* value in the column; lighter red denotes the *second-best*. For HPS, higher values are stronger aesthetics. For Δ, smaller values are stronger fidelity preservation.

| Technique | Model | EdiVal-VQ | | | EdiVal-VQΔ | | |
|---|---|---|---|---|---|---|---|
| | | T1 | T2 | T3 | T1 | T2 | T3 |
| Unknown | Seedream 4.0 | 5.14 | 5.15 | 5.15 | 0.76 | 0.77 | 0.77 |
| | GPT-Image-1.5 | 4.16 | 4.08 | 4.23 | 0.22 | 0.30 | 0.15 |
| | GPT-Image-1 | 6.65 | 6.59 | 6.56 | 2.27 | 2.21 | 2.18 |
| | Nano Banana 2 | 6.59 | 6.51 | 5.95 | 2.21 | 2.13 | 1.57 |
| | Nano Banana | 4.94 | 5.12 | 5.26 | 0.56 | 0.73 | 0.88 |
| | Gemini 2.0 Flash | 4.44 | 4.23 | 4.07 | 0.05 | 0.15 | 0.32 |
| Flow Matching | FLUX.2-max | 5.03 | 5.21 | 5.36 | 0.65 | 0.83 | 0.98 |
| | FLUX.1-Kontext-max | 5.12 | 5.07 | 5.04 | 0.74 | 0.69 | 0.66 |
| | Qwen-Image-Edit | 5.86 | 5.72 | 5.15 | 1.47 | 1.34 | 0.77 |
| | Step1X-Edit | 4.06 | 3.34 | 2.76 | 0.33 | 1.04 | 1.63 |
| | FLUX.1-Kontext-dev | 5.12 | 5.07 | 5.04 | 0.73 | 0.69 | 0.65 |
| | OmniGen | 4.61 | 4.07 | 3.50 | 0.23 | 0.31 | 0.89 |
| Diffusion | AnyEdit | 3.66 | 2.80 | 1.95 | 0.72 | 1.58 | 2.44 |
| | UltraEdit | 4.79 | 4.68 | 4.36 | 0.41 | 0.30 | 0.02 |
| | MagicBrush | 3.85 | 3.08 | 2.36 | 0.54 | 1.30 | 2.02 |
| | IP2P | 3.20 | 2.38 | 1.44 | 1.18 | 2.01 | 2.94 |

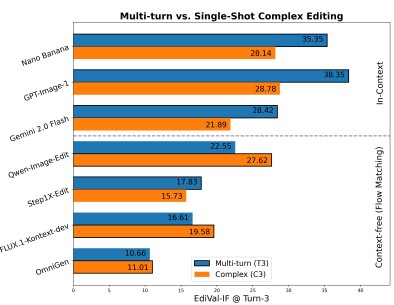

Figure 9: Turn-3 instruction following: Multi-turn vs. single-shot complex prompts.

## 3.5 PARETO FRONT

After constructing the leaderboard using EdiVal-O, we further analyze the trade-offs between different evaluation dimensions. To ensure that no model "games" the benchmark by excelling in only one dimension, we plot the Pareto boundary at Turn 3 for all pairwise combinations of our three evaluated dimensions: EdiVal-IF, EdiVal-CC, and EdiVal-VQ (see Figure 10). Additional Pareto plots for Turn 1 and Turn 2 are provided in Figures 11 and 12.

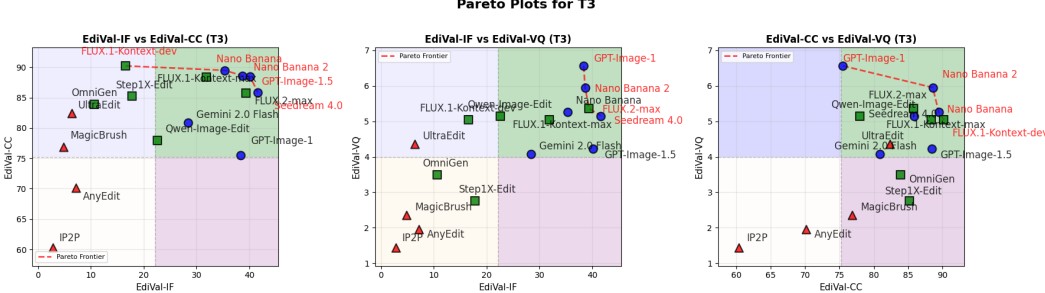

Figure 10: Pareto front plot for Turn 3 editing across EdiVal-IF, EdiVal-CC, and EdiVal-VQ.

## 3.6 TOOL SWAP ANALYSIS AND PROMPT COMPRESSION STRATEGIES

We evaluate the robustness of our evaluation stack by systematically swapping key components, including the VLM (Appendix E.1), the detector and its threshold (Appendices E.3 and E.2), and the image feature extractor (Appendix E.4). Our analysis indicates that these modifications have only a negligible impact on the final evaluation outcomes. Additionally, we conduct an ablation study on prompt compression by testing three variants: random shuffling, sequential connectors, and explicit "keep-unchanged" constraints. The results demonstrate that these structural variations yield only a mild effect on the overall complex editing success rate, suggesting the model is relatively insensitive to the specific formatting of compressed instructions. More details are in Appendix D.

## 4 CONCLUSION

We introduced **EdiVal-Agent**, an automated, and interpretable framework for evaluating instruction-based image editing. By leveraging symbolic object decomposition, structured instruction generation, and a hybrid evaluation pipeline integrating both specialist tools and vision-language reasoning models, **EdiVal-Agent** enables fine-grained, object-centric assessment of modern multi-turn editing systems. Limitations and discussions are deferred to Appendix B.

ETHICS STATEMENT

Our work focuses on developing reliable and interpretable evaluation methods for instruction-based image editing. While such technology holds promise for creative design, accessibility, and efficient content creation, it may also be misused for harmful purposes such as generating misleading, deceptive, or inappropriate content. We emphasize that our benchmark and evaluation framework are intended solely for advancing research in safe and trustworthy generative AI. To mitigate risks, we build on publicly available datasets, apply safety filters to generated images, and encourage responsible use aligned with ethical standards and community guidelines.

REPRODUCIBILITY STATEMENT

We provide complete prompting templates and pseudo-code in the Appendix, along with implementation details and API links. Comprehensive results, datasets, and evaluation metrics are also documented. To ensure full reproducibility, we will release all code, data, and model checkpoints upon acceptance of this manuscript.

ACKNOWLEDGMENTS

Ying Nian Wu was supported in part by NSF DMS-2415226, DARPA W912CG25CA007, and research gifts from Amazon and Qualcomm. Tianyu Chen and Mingyuan Zhou acknowledge the support of NSF-IIS 2212418 and NIH-R37 CA271186. The authors thank Wenyi Yin from Stanford and Binwei Yao from UW-Madison for their constructive and insightful feedback. The authors gratefully acknowledge the support of Lambda, Inc. and Microsoft for providing compute resources for this project.

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

## A    RELATED WORK

**Instruction-based editing models.**    InstructPix2Pix (IP2P) Brooks et al. (2023) introduced a two-stage recipe that converts a text-to-image diffusion model Rombach et al. (2022); Zhang et al. (2025a) into an editor: (i) synthesize paired editing data using Stable Diffusion Rombach et al. (2022) and training-free techniques such as Prompt-to-Prompt Hertz et al. (2023); (ii) fine-tune the diffusion model on these pairs. Subsequent systems—MagicBrush Zhang et al. (2023), UltraEdit Zhao et al. (2024), and AnyEdit Yu et al. (2025)—scale this paradigm to large, fine-grained real-image editing. More recent work (e.g., OmniGen Xiao et al. (2025); Wu et al. (2025b), Step1X-Edit Liu et al. (2025), FLUX.1 Kontext Labs et al. (2025), and Qwen-Image-Edit Wu et al. (2025a), Seedream Gao et al. (2025)) adopts task-aware architectures and increasingly leverages flow matching Liu et al. (2022); Lipman et al. (2022); Zhang et al. (2024).

A complementary line explores autoregressive (AR) editors such as Gemini 2.0 Flash Image Gemini2 (2025), Gemini 2.5 Flash Image ("Nano Banana") Deepmind (2025), and GPT-Image-1 OpenAI (2025). These models enable **in-context multi-turn editing**: users iteratively refine an image within a conversational interface, with the model maintaining a coherent editing history. To our knowledge, we provide the first systematic comparison of in-context multi-turn AR editing versus context-free multi-turn editing with non-AR models across instruction following, content consistency, and visual quality.

**Editing evaluation.**    Early evaluations (e.g., Brooks et al. (2023)) rely on CLIP-based similarity Radford et al. (2021), including directional variants Gal et al. (2022), to approximate editing quality. However, CLIP emphasizes semantic alignment and is less sensitive to fine, pixel-level changes. When ground-truth edited images exist, it is natural to compare model outputs against references using pixel distances ($L_1$) and semantic similarities (DINO Caron et al. (2021), CLIP Radford et al. (2021)) Zhang et al. (2023); Zhao et al. (2024); Yu et al. (2025); Sheynin et al. (2024). Yet references are imperfect: the space of valid edits is inherently multimodal, while a single reference captures only one realization; moreover, many references are themselves synthesized by prior editors (e.g., Prompt-to-Prompt Hertz et al. (2023), SDXL Podell et al. (2024), DALLE-2 Ramesh et al. (2022)), importing their biases into evaluation.

Recent work relies exclusively on VLMs as interpretable judges—e.g., VIEScore Ku et al. (2023), HQ-Edit Hui et al. (2024), and Complex-Edit Yang et al. (2025)—by querying models such as GPT-4o OpenAI (2025) about specific aspects of an edit. While VLMs offer holistic, language-mediated assessments, they are insufficient on their own: they are notoriously poor at spatial reasoning Zhang et al. (2025b); Cheng et al. (2024); Chen et al. (2024); Qharabagh et al. (2024); Chang et al. (2025) and are prone to object hallucinations in existence, category, attributes, and relations Bai et al. (2024); they have limited sensitivity to pixel-level changes and frequently miss subtle, localized modifications Vo et al. (2025) (e.g., fine structures, small attribute shifts, etc.), which are crucial for evaluating content consistency; they are miscalibrated for artifacts and aesthetics Liang et al. (2024); Xu et al. (2023); Ma et al. (2025), which humans are sensitive to. Our approach, **EdiVal-Agent**, addresses these gaps by integrating VLM-based reasoning with grounding tools, symbolic, object-centric pixel- and semantic-level tools, and human preference models, yielding a precise and interpretable evaluation of instruction-based editing.

**Editing tasks.**    We consider three settings: (i) **Single-turn vs. multi-turn.** Multi-turn editing Zhang et al. (2023); Zhao et al. (2024) is more demanding than single-turn, as the model must maintain consistency across sequential instructions. In contrast to *context-free* multi-turn pipelines (each turn consumes the previous image and the next instruction), AR models Gemini2 (2025); Deepmind (2025); OpenAI (2025) support *in-context* multi-turn editing by conditioning on the full conversational history. (ii) **Complex single-shot vs. multi-turn.** Following Yang et al. (2025), a sequence of edits can be concatenated into a single complex prompt and executed in one pass; we compare this setting to genuine multi-turn editing. (iii) **Other tasks.** We focus on instruction-based editing, the most common regime; other scenarios (e.g., prompt-to-prompt/caption-to-caption Hertz et al. (2023)) are outside our scope. To the best of our knowledge, this paper offers the first comprehensive comparison covering single-turn, multi-turn, and complex single-shot editing within a unified framework.

## B LIMITATION AND DISCUSSION

Given the object-centric evaluations conducted in this study, several limitations warrant consideration. First, our instruction types are limited to object-centric prompts, which may not capture the full range of creative editing requests typical in real-world scenarios. Future research should explore a broader spectrum of instructions, including those involving stylistic changes or complex narrative elements. Additionally, while our work provides a reliable and comprehensive evaluation framework for multi-turn editing, it does not apply the evaluation results to improve the editing models themselves. A straightforward extension would be to use evaluation scores for Best-of-N selection to improve inference-time performance. Future work could also explore post-training methods such as reinforcement learning, treating the evaluation scores as reward signals.

Please note that we have included performance evaluations for GPT-Image-1.5, Nano Banana 2, and FLUX.2-max in this camera-ready version. These models were released following the initial submission to ICLR 2026. While the main text has been updated to provide a self-contained analysis of these models, please be advised that the appendix may not reflect exhaustive comparative analysis for these three newly additions due to space and timing constraints.

## C PARETO PLOT

In addition to presenting a leaderboard in Table 13, we also visualize the trade-offs among the three EdiVal dimensions—EdiVal-IF, EdiVal-CC, and EdiVal-VQ—using Pareto plots. These plots illustrate the performance frontier for each turn, providing a more concrete view of the trade-off and allowing users to select the most suitable model according to their preference.

We present the Pareto plots for Turn 1 in Figure 11, Turn 2 in Figure 12, and Turn 3 in Figure 10.

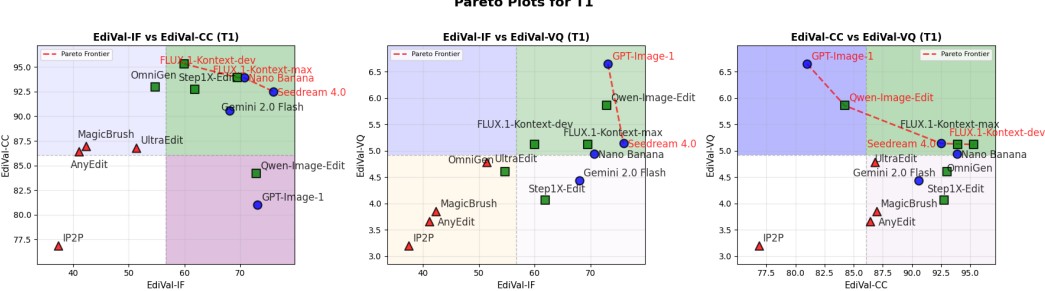

Figure 11: Pareto plot for Turn 1.

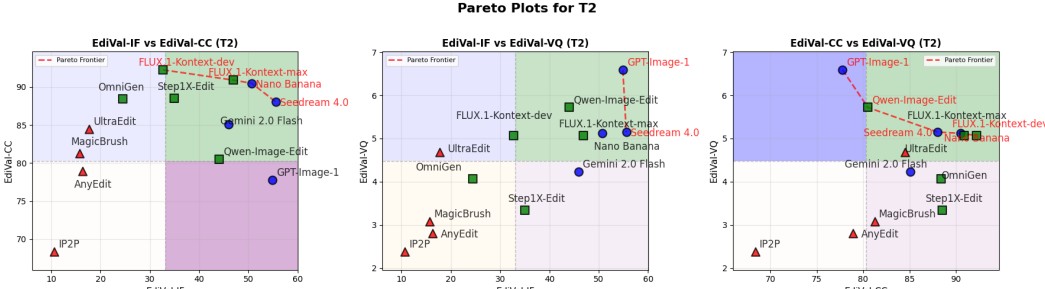

Figure 12: Pareto plot for Turn 2.

## D ABLATION STUDY ON COMPLEX EDITING COMPRESSION

We conduct an ablation study on how to compress three-turn instructions into a single "complex edit" prompt. In the previous experiment, we adopt the simplest concatenation strategy: {prompt

Table 5: Complex Editing Performance for compression variants.

| Connector Variant | Complex(C3) (%) |
|---|---|
| Default | 27.62 |
| Random shuffle | 27.10 |
| Sequential connector | 26.92 |
| Keep-unchanged | 25.87 |

Table 6: Tool swap analysis (correlations).

| Type | Default | Variant | Metric | Pearson | Spearman |
|---|---|---|---|---|---|
| VLM | Qwen2-7B-VL | Qwen2.5-7B-VL | IF | 0.9544 | 0.9298 |
| | | Qwen2.5-32B-VL | IF | 0.9790 | 0.9544 |
| | | InternVL3-8B | IF | 0.9660 | 0.9228 |
| Threshold | $[0.3, 0.4]$ | Threshold = 0.4 | IF | 0.9817 | 0.9860 |
| | | Threshold = 0.3 | IF | 0.9772 | 0.9457 |
| Filter | Has filter | No filter | IF | 0.9982 | 0.9930 |
| Detector | Grounding DINO | OWL-ViT | IF | 0.8157 | 0.7929 |
| Feature | DINOv3 | DINOv2 | CC | 0.9987 | 1.0000 |

T1}. {prompt T2}. {prompt T3}. We further evaluate three alternative variants on Qwen-Image-Edit: 1) *Random shuffle*: randomly shuffle the three per-turn prompts before concatenation. 2) *Sequential connector*: explicitly encode the order as: first, {prompt T1}, then, {prompt T2}, last, {prompt T3}. 3) *Keep-unchanged objects*: append an explicit constraint: {prompt T1}. {prompt T2}. {prompt T3}. Keep {unchanged objects} unchanged. Table 5 reports the resulting complex editing success rate(%) at $C = 3$. The results show these compression variants have only a very mild effect on performance.

## E  TOOL SWAP ANALYSIS

We analyze the effect of swapping individual components in our evaluation stack, including the VLM, detector, detector threshold, and image feature extractor. We find that modifying the VLM (Appendix E.1), adjusting the detector threshold (Appendix E.2), changing the detector (Appendix E.3), or changing the image feature extractor (Appendix E.4) has only a minor impact on the final evaluation outcomes. The summary statistics are shown in Tab. 6, which demonstrates that for each configuration replacement, our evaluation results remain highly correlated with those obtained under the default stack. As for **EdiVal-VQ**, we found that HPSv3 is the only human preference model trained on images generated after SD3.5. This suggests that other preference models, such as HPSv2, cannot provide reliable assessments of recent, more advanced generations, as they are only sensitive to earlier-stage generations that are significantly lower in quality.

However, we note that replacing Grounding DINO with OWL-ViT Minderer et al. (2022) (Appendix E.3) fails to filter out failure cases originally caused by Grounding DINO and also reduces agreement with human annotations. This highlights that detector accuracy should be prioritized when considering detector substitutions.

We further examine component-specific tasks, such as counting, where we compare the performance of density-map estimation methods (Appendix F). Finally, we include preliminary experiments on style transfer (Appendix **??**), which indicate that existing VLMs still struggle to reliably judge whether a style transfer succeeds.

**VLM swap.** We replace the default VLM in our stack (Qwen2-7B-VL) with several alternatives: Qwen2.5-7B-VL, Qwen2.5-32B-VL, and InternVL3-8B. For each configuration, we recompute the per-turn instruction-following success rates for four representative editors: GPT-Image-1, Seedream 4.0, Nano Banana, and FLUX.1-Kontext-max.

Across all VLM choices, the ranking of these editors remain the same. While the absolute numbers shift slightly (e.g., the Qwen2.5- and InternVL3-based stacks tend to assign somewhat lower scores than the default Qwen2-7B-VL stack). When we compare per-turn success rates between the default Qwen2-7B-VL stack and each swapped VLM, both Pearson and Spearman correlations remain high over all (model, turn) pairs. Concretely, we show the results in Tab 7.

### E.1  VLM SWAP

To assess how our evaluation results depend on the choice of VLM, we swap the default VLM in our stack (Qwen2.5-7B-VL) with several alternatives: Qwen2-7B-VL, Qwen2.5-32B-VL, and InternVL3-8B. For each configuration, we recompute the per-turn instruction-following success rates for four representative editors: GPT-Image-1, Seedream 4.0, Nano Banana, and FLUX.1-Kontext-max.

Across all VLM choices, the relative ranking of these editors remains unchanged. While the absolute scores shift slightly (e.g., the Qwen2.5- and InternVL-based stacks tend to assign somewhat lower scores than the Qwen2-7B-VL stack), the overall ordering is stable. When we compare per-turn/marginal success rates between the default Qwen2.5-7B-VL stack and each swapped VLM, both Pearson and Spearman correlations remain high over all (model, turn) pairs.

Concretely, Table 7 reports correlations between the per-turn success rates under our default VLM (Qwen2.5-7B-VL) and those obtained with each alternative VLM. All Pearson correlations exceed $0.95$ and all Spearman correlations exceed $0.92$, indicating that changing the VLM has only a modest effect on the absolute scores and largely preserves the relative ordering of the four models. In particular, on the EdiVal-IF leaderboard we consistently observe

$$\text{Seedream 4.0} > \text{GPT-Image-1} > \text{Nano Banana} > \text{FLUX.1-Kontext-max.}$$

Table 7: Correlations between per-turn instruction-following success rates under the default VLM (Qwen2.5-7B-VL) and three strong vision–language baselines. All correlations are computed over 12 (model, turn) points.

| Baseline Model | Pearson | Spearman |
|---|---|---|
| Qwen2-7B-VL | 0.9544 | 0.9298 |
| Qwen2.5-32B-VL | 0.9790 | 0.9544 |
| InternVL3-8B | 0.9660 | 0.9228 |

### E.2 GROUNDING DINO THRESHOLD AND LARGE-BOX FILTER SWAP

In this section, we ablate the effect of changing Grounding DINO's detection threshold and disabling the large-box filter. By default, we use a threshold of $0.30$ for `subject_remove`, $0.40$ for `position_change`, and $0.35$ for all other tasks. In addition, during detection we discard any box whose normalized height *and* width are both larger than $0.98$, in order to filter out degenerate predictions that cover almost the entire image.

To investigate sensitivity to these choices, we re-run the evaluation under three alternative settings: (1) a global threshold of $0.30$, (2) a global threshold of $0.40$, and (3) the default thresholds but with the large-box filter disabled. The results are reported in Table 8. Across all settings, the instruction-following metrics (image success, task success, and overall scores) change only slightly, and the ranking of models remains unchanged. This suggests that our conclusions are robust to reasonable variations in the detector threshold and the large-box filtering heuristic.

Table 8: Per-turn image success, task success, and overall scores under different evaluation settings (ranked).

| Config | Model | Img. Success | | | Task Success | | | Overall | | | Rank |
|---|---|---|---|---|---|---|---|---|---|---|---|
| | | T1 | T2 | T3 | T1 | T2 | T3 | T1 | T2 | T3 | |
| Default | Seedream 4.0 | 75.930 | 55.580 | 41.590 | 75.930 | 75.580 | 76.110 | 83.811 | 69.948 | 59.757 | 1 |
| | Nano Banana | 70.700 | 50.660 | 35.350 | 70.700 | 72.590 | 68.240 | 81.483 | 67.703 | 56.242 | 2 |
| | GPT-Image-1 | 73.120 | 54.890 | 37.970 | 73.120 | 74.440 | 72.740 | 76.959 | 65.340 | 53.542 | 3 |
| | FLUX.1-Kontext-max | 69.490 | 46.890 | 31.830 | 69.490 | 69.110 | 70.430 | 80.791 | 65.286 | 53.045 | 4 |
| Threshold 0.3 | Seedream 4.0 | 73.450 | 53.270 | 38.940 | 73.450 | 74.870 | 74.870 | 82.431 | 68.479 | 57.822 | 1 |
| | Nano Banana | 69.940 | 49.340 | 33.460 | 69.940 | 71.640 | 67.490 | 81.044 | 66.815 | 54.717 | 2 |
| | GPT-Image-1 | 71.430 | 52.820 | 37.030 | 71.430 | 74.060 | 72.180 | 76.065 | 64.096 | 52.875 | 3 |
| | FLUX.1-Kontext-max | 68.170 | 45.570 | 30.510 | 68.170 | 68.360 | 68.930 | 80.020 | 64.361 | 51.933 | 4 |
| Threshold 0.4 | Seedream 4.0 | 73.980 | 52.570 | 38.940 | 73.980 | 73.450 | 74.510 | 82.728 | 68.027 | 57.822 | 1 |
| | Nano Banana | 68.810 | 47.260 | 31.950 | 68.810 | 69.570 | 64.460 | 80.386 | 65.392 | 53.469 | 2 |
| | GPT-Image-1 | 70.110 | 50.190 | 35.150 | 70.110 | 72.560 | 71.240 | 75.359 | 62.480 | 51.515 | 3 |
| | FLUX.1-Kontext-max | 66.290 | 43.690 | 28.630 | 66.290 | 67.230 | 67.420 | 78.909 | 63.019 | 50.308 | 4 |
| No Large-Box Filter | Seedream 4.0 | 75.930 | 55.220 | 41.240 | 75.930 | 75.220 | 76.110 | 83.811 | 69.721 | 59.505 | 1 |
| | Nano Banana | 70.700 | 50.660 | 35.350 | 70.700 | 72.590 | 68.240 | 81.483 | 67.703 | 56.242 | 2 |
| | GPT-Image-1 | 72.930 | 54.890 | 38.350 | 72.930 | 74.440 | 73.120 | 76.859 | 65.340 | 53.809 | 3 |
| | FLUX.1-Kontext-max | 69.490 | 46.890 | 31.830 | 69.490 | 69.110 | 70.430 | 80.791 | 65.286 | 53.045 | 4 |

### E.3 DETECTOR SWAP

We also perform an ablation study by swapping the open-vocabulary detector in our pipeline from Grounding DINO to alternative detectors, such as OWL-ViT Minderer et al. (2022) and GLIP Li et al. (2022). For this swap, detector accuracy is the primary factor we must prioritize, since a more accurate detector directly translates to higher agreement with human annotations.

Table 9 summarizes the performance of several popular open-vocabulary detectors on standard detection benchmarks. In the open-set setting ODinW (object detection in the wild), which most closely resembles our scenario, Grounding DINO outperforms GLIP. It also achieves the best AP on COCO, a widely used benchmark with 80 common objects. LVIS is a challenging benchmark with more than 1k categories spanning rare, common, and frequent objects; on LVIS$^{\text{val}}$, OWL-ViT attains strong performance, but Grounding DINO still provides a better trade-off for our open-world editing setting, particularly when ODinW performance is considered.

We do not adopt other available detectors such as Grounding DINO 1.5, because they are closed-source and thus unsuitable for a community benchmark where users may need to run evaluation many times or adapt the pipeline to their own models. Moreover, our current configuration already achieves better alignment with human judgment (81.3%) than strong zero-shot VLM baselines, so switching to a closed-source detector would reduce reproducibility without a clear benefit.

When we swap Grounding DINO for OWL-ViT in our EdiVal-IF pipeline, the *relative* ranking of models (based on Img.Success at Turn 3) remains unchanged, but the *absolute* scores drop. Nevertheless, the per-turn/marginal success rates still exhibit high correlation with those under Grounding DINO (Pearson $0.82$, Spearman $0.79$). OWL-ViT frequently fails to detect objects that are clearly present in the image, which leads to many spurious "reject" decisions and thus lower Task.Success and Img.Success across all models. This bias is largely consistent across models, so EdiVal-IF preserves the same ordering, but the absolute values are shifted downward. A qualitative example is shown in Figure 14, where OWL-ViT fails to detect objects that are clearly present in the image.

We therefore do not recommend replacing Grounding DINO with OWL-ViT, GLIP, or other weaker open-vocabulary detectors in our framework. Grounding DINO remains a widely adopted state-of-the-art open-vocabulary detector, and weaker detectors not only reduce absolute performance scores but also harm agreement with human judgments. For example, when swapping Grounding DINO for OWL-ViT, the human agreement of EdiVal-IF drops from $81.30\%$ to $53.67\%$. In general, upgrading to a stronger open-vocabulary detector should preserve the relative EdiVal-IF ranking while improving absolute scores and human agreement, whereas downgrading to significantly weaker detectors has the opposite effect and is therefore undesirable.

| Method | Backbone | Pre-training data | COCO | LVIS$^{\text{minival}}$ | | | | LVIS$^{\text{val}}$ | | | | ODinW35 | ODinW13 |
|---|---|---|---|---|---|---|---|---|---|---|---|---|---|
| | | | AP$_{\text{all}}$ | AP$_{\text{all}}$ | AP$_{\text{r}}$ | AP$_{\text{c}}$ | AP$_{\text{f}}$ | AP$_{\text{all}}$ | AP$_{\text{r}}$ | AP$_{\text{c}}$ | AP$_{\text{f}}$ | AP$_{\text{avg}}$ | AP$_{\text{avg}}$ |
| OWL-ViT | ViT-L | O365, OID, VG, LiT | 42.2 | - | - | - | - | **34.6** | **31.2** | - | - | - | - |
| GLIP | Swin-L | FourODs, GoldG, Cap24M | 49.8 | 37.3 | 28.2 | 34.3 | 41.5 | 26.9 | 17.1 | 23.3 | 35.4 | - | 52.1 |
| Grounding DINO | Swin-L | O365, OID, GoldG | **52.5** | - | - | - | - | - | - | - | - | 26.1 | **56.9** |

Table 9: Performance of popular detectors including OWL-ViT, GLIP, and Grounding DINO (Swin-L). Numbers are copied from Table 1 of Grounding DINO 1.5 Ren et al. (2024). In the open-set setting ODinW, which is most similar to our scenario, Grounding DINO outperforms GLIP. It also achieves the best AP on COCO. LVIS is a large-scale benchmark with over 1k categories spanning rare, common, and frequent objects.

**Can detector swapping avoid failure cases?** No. We also examine the same failure cases of Grounding DINO under OWL-ViT and GLIP. In these examples, OWL-ViT behaves differently from Grounding DINO but does not fix the underlying issue: regardless of whether the object truly exists in the image, OWL-ViT often detects nothing at all. GLIP, on the other hand, exhibits similar false-positive behavior to Grounding DINO, but with denser and less precise bounding boxes. Thus, switching to OWL-ViT or GLIP does not eliminate such failure cases; it merely changes them into systematic missed detections or more cluttered false positives.

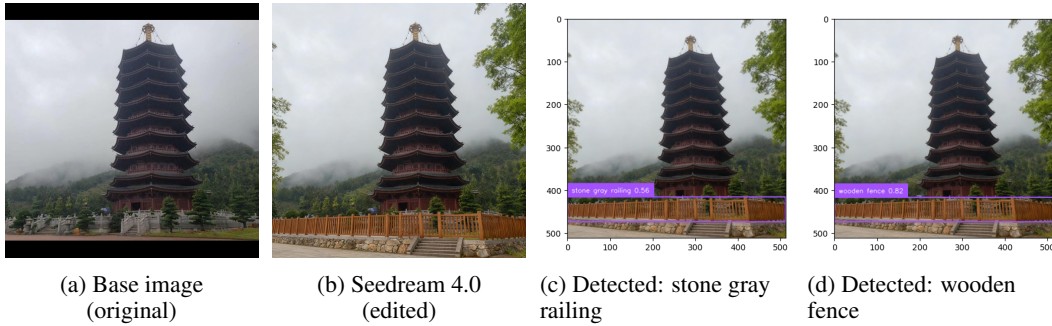

Figure 13: **Failure due to detector false positives.** Although the edit visually replaces the railing with a wooden fence, Grounding DINO fires on both "stone gray railing" and "wooden fence" in overlapping regions, causing an incorrect failure in our instruction-following metric.

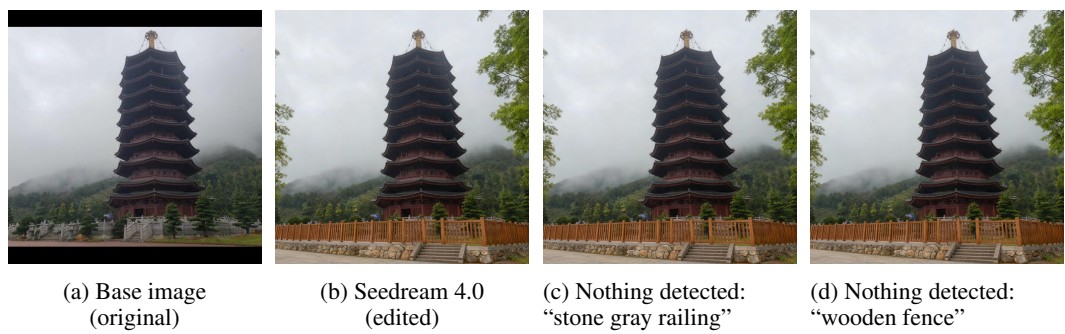

Figure 14: OWL-ViT fails to detect the queried objects regardless of whether they are actually present in the image, and therefore does not resolve this failure case.

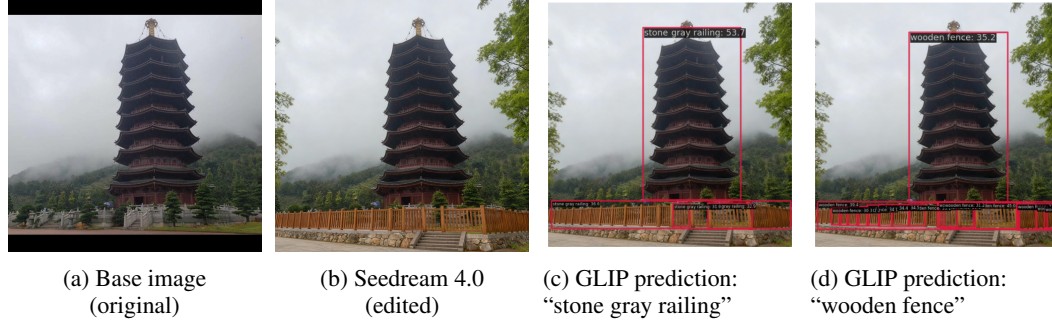

Figure 15: GLIP produces substantially more false positives than Grounding DINO, hallucinating objects such as a "stone gray railing" and misclassifying the tower region as a "wooden fence" or "stone gray railing".

### E.4 IMAGE FEATURE EXTRACTOR SWAP: DINOV3 TO DINOV2

We further ablate the image feature extractor by swapping the default backbone (DINOv3) with DINOv2. As with our other component swaps (VLM, detector, and threshold), we observe the same qualitative behavior: the *relative* ranking of models remains unchanged, while the *absolute* consistency scores shift.

Concretely, DINOv2 systematically produces lower EdiVal-CC values than DINOv3, but the ordering across models is identical. When we compare per-turn consistency scores across the two backbones over all (model, turn) pairs, we obtain Pearson correlation 0.9987 and Spearman correlation 1.0000, indicating that DINOv2 and DINOv3 induce essentially the same ranking and very similar relative differences between models, despite the shift in absolute scale.

Table 10: DINO-v2 feature backbone results with EdiVal-CC Rank.

| Model | Turn 1 | Turn 2 | Turn 3 | EdiVal-CC Rank |
|---|---|---|---|---|
| Seedream 4.0 | 89.455 | 84.185 | 81.120 | 3 |
| Nano Banana | 90.820 | 86.900 | 85.070 | 1 |
| GPT-Image-1 | 76.360 | 72.555 | 69.845 | 4 |
| FLUX.1-Kontext-max | 91.530 | 87.535 | 84.460 | 2 |

Table 11: DINO-v3 feature backbone results with EdiVal-CC Rank.

| Model | Turn 1 | Turn 2 | Turn 3 | EdiVal-CC Rank |
|---|---|---|---|---|
| Seedream 4.0 | 92.51 | 88.03 | 85.86 | 3 |
| Nano Banana | 93.91 | 90.48 | 89.48 | 1 |
| GPT-Image-1 | 81.00 | 77.78 | 75.50 | 4 |
| FLUX.1-Kontext-max | 93.93 | 90.90 | 88.40 | 2 |

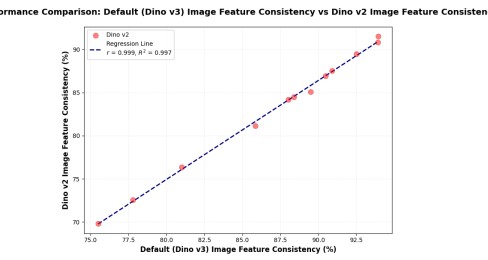

Figure 16: Per-turn EdiVal-CC scores under DINOv3 (x-axis) vs. DINOv2 (y-axis) for the four representative models. Each point is a (model, turn) pair; Pearson correlation is 0.9987 and Spearman correlation is 1.0000, confirming that the feature-extractor swap only shifts the absolute scale while preserving the ranking.

## F IMPROVEMENT TOWARDS COUNTING

When we generate prompts, the target count is never higher than 10. Under this regime, Grounding DINO is sufficiently reliable for counting the relevant objects.

Potentially there are two other families of methods to do counting: tracking-based counting and density-map estimation. Tracking-based methods are typically designed for videos, where they exploit multiple frames to track objects over time. In our case, we only use single edited images to do counting, so such tracking-based approaches are not applicable.

Density-map estimation methods (e.g., CSRNet Zhang et al. (2016); Li et al. (2018)) are usually developed for crowd counting of a *fixed* target category (such as humans) on specific datasets, rather than open-vocabulary object counting. When the objects to be counted are not aligned with the training data distribution, these methods can fail dramatically. To make this concrete, we apply CSRNet to one of our examples: CSRNet outputs an estimated count of 44.86, whereas Grounding DINO correctly predicts 3 bounding boxes for the target objects.

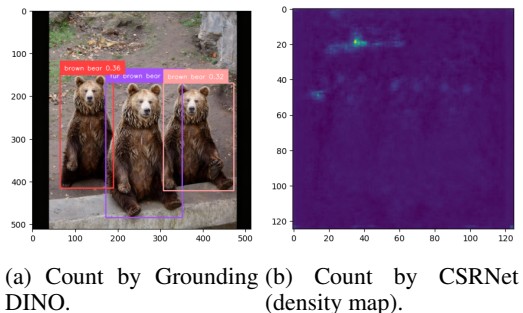

(a) Count by Grounding DINO.

(b) Count by CSRNet (density map).

Figure 17: Illustration of counting with an open-vocabulary detector (Grounding DINO) vs. a density-map-based crowd counter (CSRNet). CSRNet severely overestimates the count in this open-vocabulary setting.

# G  MORE METRICS FOR HUMAN AGREEMENT

The most straightforward metric is accuracy. Here, we provide more metrics measuring human agreement: Pearson Linear Correlation Coefficient (PLCC), Cohen's Kappa Coefficient (Kappa) and F1 scores as shown in Tab. 12. However, we note that for 0/1 predictions, correlation metrics like PLCC and Kappa may be considered not suitable for measuring the agreement with human annotators.

Table 12: **Task-specific and overall performance comparison across models.** Metrics reported: Pearson Linear Correlation Coefficient (PLCC), Cohen's Kappa, and F1. Best per column highlighted in **bold**.

| Task Type | Model | PLCC | Kappa | F1 |
|---|---|---|---|---|
| subject_add | CLIP_dir | 0.2110 | 0.1764 | 0.6914 |
| | Qwen2.5-VL | 0.5331 | 0.5264 | **0.7893** |
| | EdiVal-IF | **0.5365** | **0.5364** | 0.7786 |
| background_change | CLIP_dir | -0.0329 | -0.0076 | 0.8745 |
| | Qwen2.5-VL | **0.5792** | **0.5686** | **0.8889** |
| | EdiVal-IF | 0.5244 | 0.5157 | 0.8763 |
| subject_remove | CLIP_dir | -0.0011 | -0.0002 | 0.6592 |
| | Qwen2.5-VL | 0.1891 | 0.1758 | 0.4896 |
| | EdiVal-IF | **0.5473** | **0.5409** | **0.7837** |
| count_change | CLIP_dir | 0.0456 | 0.0142 | 0.0782 |
| | Qwen2.5-VL | 0.1998 | 0.1748 | 0.2162 |
| | EdiVal-IF | **0.3431** | **0.3274** | **0.3571** |
| material_alter | CLIP_dir | 0.2086 | 0.2038 | 0.4561 |
| | Qwen2.5-VL | **0.8658** | **0.8616** | **0.8971** |
| | EdiVal-IF | 0.4778 | 0.4624 | 0.6364 |
| color_alter | CLIP_dir | 0.1841 | 0.0874 | 0.8542 |
| | Qwen2.5-VL | **0.8409** | **0.8407** | **0.9573** |
| | EdiVal-IF | 0.7820 | 0.7744 | 0.9338 |
| position_change | CLIP_dir | 0.0996 | 0.0430 | 0.3285 |
| | Qwen2.5-VL | -0.0381 | -0.0374 | 0.1798 |
| | EdiVal-IF | **0.3907** | **0.3271** | **0.5000** |
| text_change | CLIP_dir | 0.6178 | 0.6173 | 0.8063 |
| | Qwen2.5-VL | 0.7161 | 0.6947 | **0.8651** |
| | EdiVal-IF | **0.7438** | **0.7347** | 0.8571 |
| subject_replace | CLIP_dir | 0.0420 | 0.0121 | 0.8219 |
| | Qwen2.5-VL | **0.6028** | **0.5994** | **0.8699** |
| | EdiVal-IF | 0.5533 | 0.5429 | 0.8410 |
| **Overall** | CLIP_dir | 0.3186 | 0.2568 | 0.6858 |
| | Qwen2.5-VL | 0.6162 | 0.6161 | 0.7922 |
| | EdiVal-IF | **0.6278** | **0.6273** | **0.8030** |

## H    ARTIFICIAL ANALYSIS LEADERBOARD

We report the leaderboard from the Artificial Analysis website as of **September 12, 2025** (Fig. 18). To ensure a fair comparison, we align on the intersection of models evaluated by both platforms and *exclude* Qwen-Image-Edit. Among the overlapping systems—Seedream 4.0, Nano Banana (Gemini 2.5 Flash), GPT-Image-1 (GPT-4o), FLUX.1-Kontext-max, and Gemini 2.0 Flash—the relative ordering of human votes on Artificial Analysis matches our EdiVal rankings exactly (Table 13), supporting the accuracy of our methodology.

Table 13: **Model rankings on the overlapping set.** Relative ranks from Artificial Analysis (human votes) vs. EdiVal (ours) as of Sep 12, 2025.

| Model | Artificial Analysis (Rank) | EdiVal (Rank) |
|---|---|---|
| Seedream 4.0 | 1 | 1 |
| Nano Banana (Gemini 2.5 Flash) | 2 | 2 |
| GPT-Image-1 (GPT-4o) | 3 | 3 |
| FLUX.1-Kontext-max | 4 | 4 |
| Gemini 2.0 Flash | 5 | 5 |

| Creator | Model | ELO | 95% CI | Appearances |
|---|---|---|---|---|
| ByteDance Seed | Seedream 4.0 | 1,205 | −20/+23 | 1,607 |
| Google | Gemini 2.5 Flash | 1,201 | −13/+13 | 5,783 |
| Black Forest Labs | FLUX.1 Kontext [pro] | 1,089 | −12/+12 | 5,993 |
| OpenAI | GPT−4o | 1,088 | −12/+13 | 5,781 |
| Alibaba | Qwen−Image−Edit | 1,087 | −12/+12 | 6,103 |
| Black Forest Labs | FLUX.1 Kontext [max] | 1,083 | −12/+12 | 5,947 |
| ByteDance seed | SeedEdit 3.0 | 1,076 | −21/+21 | 1,372 |
| HiDream | HiDream−E1.1 | 1,005 | −13/+13 | 5,112 |
| Google | Gemini 2.0 Flash Preview | 1,000 | +0/+0 | 5,521 |
| Black Forest Labs | FLUX.1 Kontext [dev] | 995 | −13/+13 | 5,679 |

Figure 18: **Artificial Analysis leaderboard (Sep 12, 2025).** Screenshot of the public leaderboard used for comparison in Table 13.

## I  FAILURE CASE

We discuss a representative failure mode of our evaluation. The most severe errors arise from false positives in *Grounding-DINO*, despite its strong open-vocabulary performance. Consider the prompt: *"Replace [stone gray railing] with [wooden fence]."* As shown in Fig. 19, Seedream 4.0 produces an edit that is visually correct. Our rule for subject_replace declares success if, on the edited image, the *source* object (stone gray railing) is no longer detected while the *target* object (wooden fence) is detected. However, Grounding-DINO occasionally reports both the source and target objects in the same region with high confidence, incorrectly suggesting that the source object remains and thereby degrading the measured instruction-following accuracy. Improving the reliability of open-vocabulary detection—particularly reducing false positives—would directly improve the fidelity of our evaluation.

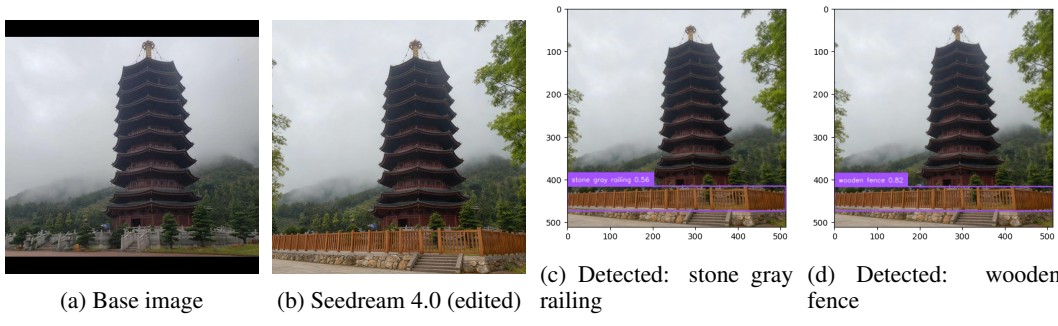

(a) Base image    (b) Seedream 4.0 (edited)    (c) Detected: stone gray railing    (d) Detected: wooden fence

Figure 19: **Failure due to detector false positives.** Although the edit visually replaces the railing with a wooden fence, Grounding-DINO fires on both "stone gray railing" and "wooden fence" in overlapping regions, causing an incorrect failure in our instruction-following metric.

## J    DISCUSSION ON SINGLE-SHOT COMPLEX EDITING

Figure 20 shows that marginal success for the final instruction remains largely stable as complex prompt length increases. Together with the multi-turn drops seen in Figure 5, this pattern supports an *exposure-bias* explanation: performance degradation primarily stems from error accumulation across sequential edits rather than an intrinsic inability to handle multiple instructions in a single prompt.

## K    DISCUSSION ON VISUAL QUALITY

Beyond instruction following and content consistency, the perceptual *quality* of the edited image is a key dimension. We therefore report (i) a learned aesthetic score and (ii) several low-level image statistics that can surface systematic artifacts and drift in multi-turn editing pipelines.

Table 14: Turn-3 instruction following: Multi-turn vs. single-shot complex prompts, grouped by technique. **Bold** indicates which setting is higher for each model.

| Technique | Model | Multi-turn (T3) | Complex (C3) |
|---|---|---|---|
| In-Context | Nano Banana | **35.35** | 28.14 |
| | GPT-Image-1 | **38.35** | 28.78 |
| | Gemini 2.0 Flash | **28.42** | 21.89 |
| Flow Matching | Qwen-Image-Edit | 22.55 | **27.62** |
| | Step1X-Edit | **17.83** | 15.73 |
| | FLUX.1-Kontext-dev | 16.61 | **19.58** |
| | OmniGen | 10.66 | **11.01** |
| Diffusion | AnyEdit | 7.22 | 2.80 |
| | UltraEdit | 6.36 | 8.22 |
| | MagicBrush | 4.90 | 4.55 |
| | IP2P | 2.80 | 2.80 |

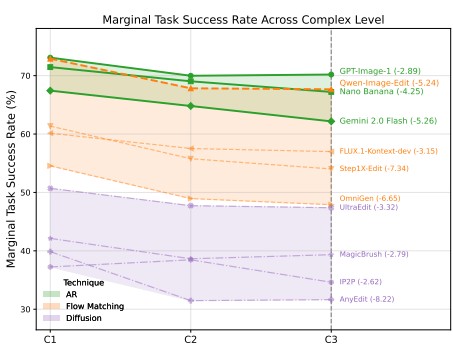

Figure 20: Marginal task-success rate of the *last* instruction as a function of complex prompt length (levels $C = 1, 2, 3$).

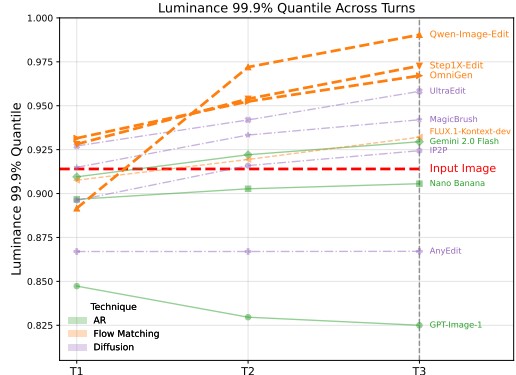

Figure 21: Per-image 99.9% luminance quantile across turns. Higher values indicate more extreme bright pixels and greater risk of over-exposure.

**Low-level image statistics**    In addition to learned aesthetic scores, we compute several low-level image statistics that help reveal systematic, multi-turn editing artifacts. Concretely, we convert RGB pixels to luminance using the Rec. 709 luma coefficients: $Y = 0.2126\,R + 0.7152\,G + 0.0722\,B$, and for each edited image we extract the **99.9% luminance quantile** (the per-image pixel value below which 99.9% of pixels fall). The 99.9% quantile is sensitive to high-exposure pixels and therefore highlights over-exposure and bright streaks while being robust to single-pixel outliers. In Figure 21 we plot the trend of this statistic across turns.

The measured trend shows a clear pattern: **Qwen-Image-Edit** and several other flow-matching models (with the notable exception of **FLUX.1-Kontext-dev**) exhibit a pronounced increase in the

99.9% luminance quantile over turns, indicating progressive brightening and increased risk of over-exposure. By contrast, regeneration-style editors such as **GPT-Image-1** tend to produce lower luminance values than the input (reflecting darker, more conservative reconstructions), and several models remain stable across turns.

Figure 22 provides qualitative examples from Qwen-Image-Edit. The edited images exhibit elevated luminance and noticeable high-frequency bright artifacts (e.g., white streaks or "line" textures) that degrade perceptual quality, with luminance quintiles increasing substantially. Correspondingly, HPS drops from 6.19 to 4.19 and 3.34, suggesting that HPS is sensitive to over-exposure to some extent. In contrast, when querying VLMs about the visual quality of these images, the returned scores do not change in the first two turns and remain consistently above 50, reflecting a *positive* evaluation under the [0, 100] scale, while the T2/T3 edited images show significant artifacts.

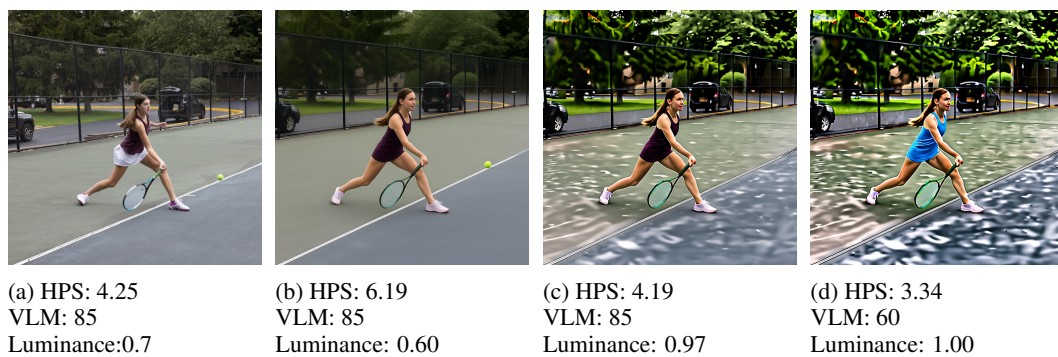

(a) HPS: 4.25
VLM: 85
Luminance:0.7

(b) HPS: 6.19
VLM: 85
Luminance: 0.60

(c) HPS: 4.19
VLM: 85
Luminance: 0.97

(d) HPS: 3.34
VLM: 60
Luminance: 1.00

Figure 22: Representative Qwen-Image-Edit examples illustrating over-exposure and bright artifact formation across turns. Although editing instructions are often satisfied, the images show elevated luminance and high-frequency bright streaks that accompany the edits (visible especially in T2/T3). Editing instructions: [Remove polyester white skirt, Change the count of tennis ball to 4, Change the color of tank top to blue]. Note that VLM gives a *positive* score to all the images.

## L  VLMs FAILING TO JUDGE VISUAL QUALITY

The following is the zero-shot prompt for visual quality with VLMs. The example results are shown in Fig. 22.

```
You are an expert at evaluating image visual quality and naturalness.

I will show you an image.

Please analyze whether the image is visually pleasing and natural. Consider:
1. Is the image visually pleasing?
2. Is the image natural?
3. Does the image look natural and coherent?

Respond only with a score between 0 and 100, where 100 is the highest score.
100 means the image is visually pleasing and natural.
0 means the image is not visually pleasing and natural.
50 means the image is neutral.
```

---

**Algorithm 1** Object Listing and Grounding Filter

---

**Require:** image $I$
**Ensure:** object pool $\mathcal{O}$ with grounding metadata
1: $J \leftarrow \text{LISTOBJECTS}(I)$                       ▷ see Sec. M.2.1
2: $\mathcal{O} \leftarrow \emptyset$
3: **for all** $(\text{name}, \text{attrs}) \in J$ excluding key `All Objects` **do**
4:      $(\text{boxes}, \text{phrases}, \text{scores}) \leftarrow \text{GROUND}(I, \text{name})$       ▷ thresholds 0.3–0.4
5:      **if** boxes $\neq \emptyset$ **and** each box has $w, h < 0.9$ **and** area $\leq 0.4$ **then**
6:          $\mathcal{O}[\text{name}] \leftarrow \text{attrs}$; attach grounding metadata (count, boxes, phrases, scores)
7:      **end if**
8: **end for**
9: $\mathcal{O}[\texttt{Filtered All Objects}] \leftarrow \text{JOIN}(\text{KEYS}(\mathcal{O}), ". ")$ and then append "."

---

# M ALGORITHMIC DETAILS

This appendix provides the algorithmic details of our pipeline: object discovery and grounding-based filtering (decomposition), instruction generation for multi-turn editing, and evaluation (instruction following, consistency, and perceptual quality). We also list the exact prompts and implementation specifics needed for reproducibility, and summarize the model-generation configurations.

## M.1 DECOMPOSITION

We first enumerate visible objects in an input image using a vision-language model (VLM) prompt, then filter these objects via visual grounding.

- Object listing: We use `GPT-4o` with the prompt in Section M.2.1. The model returns a JSON with one entry per object and a terminal aggregated string key `''All Objects''`.

- Grounding filter: We use GroundingDINO SwinT-OGC Liu et al. (2024a) to retain only objects that can be visually grounded. We resize images to $512 \times 512$. We keep detections meeting text/box thresholds (0.35) and reject oversized boxes by checking width/height in normalized coordinates; we use `max_box_size=0.9` and filter large regions if area $> 0.4$. The output augments each kept object with grounding counts, phrases, boxes, and scores, and creates a `''Filtered All Objects''` string listing retained objects.

## M.2 INSTRUCTION GENERATION

We generate multi-turn editing instructions from the grounded object pool. We support nine task types: local edits {`subject_replace`, `subject_remove`, `material_alter`, `color_alter`, `subject_add`, `text_change`, `position_change`, `count_change`} and the global edit {`background_change`}. We set `MAX_TURNS=3`. At each turn, we sample a new task type without repetition where feasible. Feasibility is checked against the current object pool (e.g., `position_change` requires at least two objects). If a sampled task is infeasible, we fall back to `subject_add`. We maintain an available-objects pool that is updated after each instruction according to its semantics (adds, removes, or modifies attributes). If a background change occurs, we mark `bg_consistency=false` for subsequent turns and restrict the pool to foreground objects for the remainder of the episode.

**Prompts (Full Text)** Below we reproduce the prompts used by our generators, reformatted for readability in print (content preserved).

## M.2.1 OBJECT LISTING PROMPT

You will be given an image. Your task is to identify and describe all clearly visible objects in the image in a structured JSON format.
**Output rules:**

---

**Algorithm 2** Multi-Turn Instruction Generation

---

**Require:** grounded pool $\mathcal{O}_0$, turns $T{=}3$
**Ensure:** tasks $\{\tau_t\}$, instructions $\{I_t\}$, formats $\{F_t\}$, flag has_bg, set all_objects_ever
 1: used $\leftarrow \emptyset$; $\mathcal{O} \leftarrow \mathcal{O}_0$; has_bg $\leftarrow$ false
 2: all_edited $\leftarrow \emptyset$; all_objects_ever $\leftarrow$ keys($\mathcal{O}_0$)
 3: **for** $t = 1$ to $T$ **do**
 4:     cand $\leftarrow \{$all task types$\} \setminus$ used
 5:     $\tau_t \leftarrow$ sample(cand);
 6:     **if** not feasible($\tau_t, \mathcal{O}$) **then** $\tau_t \leftarrow$ `subject_add`
 7:     **end if**
 8:     $F_t \leftarrow$ format_instruction($\tau_t, \mathcal{O}$)            $\triangleright$ Query VLM by prompts in Section M.2.2
 9:     $I_t \leftarrow$ render_instruction($F_t, \tau_t, \mathcal{O}$)     $\triangleright$ strip brackets; add unchanged list for background
10:     used $\leftarrow$ used $\cup \{\tau_t\}$; append $F_t, I_t$
11:     Update *all_object_pool* by adding any objects introduced in instruction $I_t$.
12:     Update *available_object_pool* by adding or removing objects as specified in $I_t$.
13:     Update *unchanged_objects_pool* by removing any objects affected by $I_t$.
14:     **if** $\tau_t =$ `background_change` **then**
15:         has_bg $\leftarrow$ true; $\mathcal{O} \leftarrow$ filter_foreground($\mathcal{O}$)
16:     **end if**
17: **end for**
18: **return** $\{\tau_t\}, \{I_t\}, \{F_t\}$, has_bg, all_objects_ever

---

1. Each object must be listed as a key in the JSON, using the format: "{material} {color} {object name}". If the material or color is unknown, omit that part. Do not include any visible text in the key. Do not use "person" as an object name; instead, describe wearable items (e.g., "blue cotton shirt").

2. For each object, the value is a dictionary with fields: "object" (type, e.g., shirt, cup), "color" (dominant color, use null if unknown), "material" (likely material, use null if unknown), "text" (visible text, null if none), "count" (number of instances), and "foreground" (boolean).

3. Do not include objects that are too small to describe, mostly occluded/incomplete, or only background scenery (e.g., distant sky, wall, floor).

4. Add a final key "All Objects" whose value is a single string listing all object names, formatted as: "{material} {color} {object name}. {color} {object name}. {material} {object name}. {object name}." Exclude "null"/"None" and separate entries by ". " (period + space). Do not include any text content in this list.

**Example output (abridged JSON):**

- "cotton blue shirt": {object: "shirt", color: "blue", material: "cotton", text: null, count: 1, foreground: true}
- "ceramic white cup": {object: "cup", color: "white", material: "ceramic", text: "GOOD DAY", count: 1, foreground: false}
- "leather bag": {object: "bag", color: null, material: "leather", text: null, count: 2, foreground: true}
- "red scarf": {object: "scarf", color: "red", material: null, text: null, count: 1, foreground: true}
- "All Objects": "cotton blue shirt. ceramic white cup. leather bag. red scarf."

### M.2.2 TASK PROMPTS

**Subject Replace**

You are given an image and asked to suggest a replacement object for a specific object in the scene.

**Given object to replace:** *object_name*

**Your task:**

1. Understand the scene context.
2. Suggest a new object that naturally replaces "*object_name*".

3. Ensure the suggestion is realistic for the scene.

4. Respond with only the object name (e.g., "chair", "lamp", "book").

**Examples:** In a kitchen: "bowl", "mug"; on a street: "bus", "truck"; in an office: "stool", "bench".

**Answer format:** New object name:

## Material Alter

You are given an image and asked to suggest a new material for a specific object.

**Object:** *object_name*    **Current material:** *current_material*

**Your task:**

1. Identify the object.

2. Suggest a realistic alternative material that is easy to distinguish from the current one.

3. Respond with only the material name (e.g., "wood", "metal", "plastic", "leather").

**Examples:** cup: ceramic, glass, metal, plastic; chair: wood, metal, plastic, fabric; bag: leather, canvas, nylon, fabric.

**Answer format:** New material:

## Position Change

You are given an image and asked to create a position change instruction.

**Available objects:** *available objects*    **Positions:** left, right, above, below

**Your task:**

1. Select a target object to move and a reference object.

2. Choose a relative position (left, right, above, below).

3. Ensure the instruction is physically reasonable.

4. **Format:** "Change the position of [target object] to [position] of [reference object]".

**Examples:** "Change the position of [cup] to [right] of [book]"; "Change the position of [lamp] to [above] of [table]".

**Answer format:** Position change instruction:

## Count Change

You are given an image and asked to create a count change instruction.

**Available objects:** *available objects*    **Target count:** *target count*

**Your task:**

1. Identify a suitable object for the requested count.

2. Ensure the target count is realistic for the scene.

3. **Format:** "Change the count of [object name] to [target count]".

**Examples:** "Change the count of [cup] to [3]"; "Change the count of [book] to [2]".

**Answer format:** Count change instruction:

## Text Change

You are given an image and asked to generate new text content.

**Context:** *text situation*

**Your task:**

1. Generate text that fits the scene.

2. Keep text short: *max 2 words in English or 4 Chinese characters*.

3. Respond with only the text content (no quotes or extra words).

**Examples:** coffee shop: "COFFEE", "OPEN"; book: "NOVEL", "GUIDE"; sign: "EXIT", "STOP"; Chinese: " 咖啡 ", " 出口 ".

**Answer format:** New text:

## Color Alter

You are given an image and asked to suggest a new color for a specific object.

**Object:** *object name*    **Current color:** *current color*

**Your task:**

1. Suggest a simple, common color that fits the object.
2. Use only basic colors: red, blue, green, yellow, black, white, brown, gray, orange, purple, pink.
3. Choose a color different from the current color and answer with the color name only.

**Answer format:** New color:

### Subject Add

You are given an image and asked to suggest a new object to add to the scene.

**Reference object:** *reference object*    **Position:** *position*

**Your task:**

1. Propose an object that would naturally fit at the specified position relative to the reference object.
2. Ensure the suggestion is realistic and contextually appropriate.
3. Respond with only the object name (e.g., "lamp", "book", "cup").

**Examples:** next to a desk: "chair", "lamp", "computer"; near a kitchen counter: "bowl", "plate", "mug"; by a window: "plant", "curtain", "book".

**Answer format:** New object:

### Background Change

You are given an image and asked to suggest a new background for the scene. The existing objects should remain unchanged.

**Your task:**

1. Propose a new background that works with the current setting.
2. Keep it simple and realistic; use 1–2 words (e.g., "kitchen", "office", "garden", "beach", "forest").
3. Respond with only the background name.

**Answer format:** New background:

## M.3   EVALUATION

We evaluate in two modes: (i) **Multi-turn** (each turn edits the output of the previous turn), and (ii) **Complex Editing** (compress all instructions to a single prompt).

**Instruction Following.**   We compute a binary success per instruction with a detector combining GroundingDINO Liu et al. (2024a) and a VLM (Qwen2-VL-7B) Bai et al. (2025). Representative details:

- Detector thresholds. Unless noted per task, GroundingDINO thresholds are 0.3–0.4; detections return normalized boxes $[x_1, y_1, x_2, y_2]$.
- Cropping and small objects. For object-level checks we crop by detected boxes; very small boxes ($<0.05$ in width/height) can be enlarged before VLM queries.
- Replace. Detect old and new objects in source/target; success if both are detected and any IoU between a source box (old) and a target box (new) is $> 0$. A VLM pre-check rejects obvious non-replacements. See details in Alg 3.
- Remove. Detect the object in the source; success if the object is absent in the target. See details in Alg 4.
- Position change. Detect target and reference objects and verify the requested spatial relation using object centers; also ensure the object count did not increase spuriously. See details in Alg 6.
- Count change. Use the detector to locate instances of the target object and take the number of validated detections as the count. See details in Alg 7.

---

**Algorithm 3** Evaluate Subject Replace

---

**Require:** base $B$, target $T$, old object name $o$, new object name $n$
**Ensure:** success flag succ
 1: $S \leftarrow \text{DETECT}(B, o, \tau); \quad T_n \leftarrow \text{DETECT}(T, n, \tau)$
 2: **if** $S \neq \emptyset \land T_n \neq \emptyset$ **then**
 3: $\quad$ succ $\leftarrow \max_{b \in S, t \in T_n} \text{IOU}(b, t) > 0$
 4: **else**
 5: $\quad$ succ $\leftarrow$ false
 6: **end if**
 7: **return** succ

---

**Algorithm 4** Evaluate Subject Remove

---

**Require:** base $B$, target $T$, object name $o$
**Ensure:** success flag succ
 1: $S \leftarrow \text{DETECT}(B, o, \tau); \quad T_o \leftarrow \text{DETECT}(T, o, \tau)$
 2: succ $\leftarrow (S \neq \emptyset \land T_o = \emptyset)$
 3: **return** succ

---

- Color/material. Crop the object in the target and ask the VLM a yes/no question about the new color/material. See details in Alg 8 and Alg 9.

- Text change. If the instruction adds text anywhere, run the VLM on the whole image; if it replaces text on a specific object, first crop that object's box, ask the VLM to extract the text, and compare it to the requested text. See details in Alg 10.

- Background change. Ask the VLM yes/no whether the requested background category is present. See details in Alg 11.

**Consistency.** We measure object and background stability as follows:

- Object consistency (unchanged objects): DINOv3 ViT-B/16 Siméoni et al. (2025) feature similarity between crops of unchanged objects in base vs. target; we also report pixel L1 consistency and average across objects per image.

- Background consistency: detect objects in all_objects_pool in base/target (Ground-ingDINO), mask them to isolate background, then compute masked L1 between backgrounds (optionally DINOv3 masked similarity). Background consistency is evaluated only when no background change occurred earlier (bg_consistency=true).

**Perceptual Quality.** We report HPSv3 Ma et al. (2025) plausibility and aesthetics, plus luminance metrics. Quality is *not* folded into the *Overall* score.

## M.4 Overall Score and Aggregation Details

Let $\alpha_t$ be the image success rate at turn $t$: the fraction of images for which *all* edits up to and including turn $t$ are successful (aggregated per task type, then averaged). Let $\kappa$ denote the average content-consistency score combining object and background DINOv3 similarities when applicable.

- Overall score. We report

$$\text{Overall} = \left[\text{mean}_t(\alpha_t) \times \text{mean}(\kappa)\right]^{1/2}.$$

- Missing outputs across turns. For summary tables, we include only images that produce all required outputs for the evaluated mode. If a model fails to generate a later turn, that image is omitted from later-turn aggregates for that mode. Some edits will be rejected by some models since the sensitive content flag.

- No unchanged objects. If the unchanged-object list is empty, object consistency is recorded as None and excluded from averages; background consistency is still computed when bg_consistency=true.

---

**Algorithm 5** Evaluate Subject Add

---

**Require:** base $B$, target $T$, new object name $n$, optional reference object name $r$, optional position $p \in \{\text{left}, \text{right}, \text{above}, \text{below}\}$
**Ensure:** success flag
1: $B_n \leftarrow \text{DETECT}(B, n, \tau); \quad T_n \leftarrow \text{DETECT}(T, n, \tau)$
2: **if** $T_n = \emptyset \lor B_n \neq \emptyset$ **then**
3:      **return** false
4: **end if**
5: **if** $r$ and $p$ are provided **then**
6:      $B_r \leftarrow \text{DETECT}(B, r, \tau); \quad T_r \leftarrow \text{DETECT}(T, r, \tau)$
7:      **if** $T_r = \emptyset$ **then**
8:          **return** false
9:      **end if**
10:     Choose max logits boxes $t \in T_n$, $u \in T_r$
11:     $(x_t, y_t) \leftarrow \text{CENTER}(t); \quad (x_u, y_u) \leftarrow \text{CENTER}(u)$
12:     **if** $p = \text{left} \land x_t < x_u - \varepsilon_x$ **then**
13:         **return** true
14:     **end if**
15:     **if** $p = \text{right} \land x_t > x_u + \varepsilon_x$ **then**
16:         **return** true
17:     **end if**
18:     **if** $p = \text{above} \land y_t < y_u - \varepsilon_y$ **then**
19:         **return** true
20:     **end if**
21:     **if** $p = \text{below} \land y_t > y_u + \varepsilon_y$ **then**
22:         **return** true
23:     **end if**
24:     **return** false
25: **else**
26:     **return** true
27: **end if**

---

- Turn-level reporting. We also report per-turn (T1, T2, T3) instruction-following and consistency, and per-task-type success rates $\alpha_{t,\text{type}}$. Quality metrics are reported separately and are not folded into *Overall*.

## M.5 MODEL GENERATIONS

We evaluate a mix of closed- and open-source editors using each model's default settings (no hyperparameter tuning):

- GPT-Image-1, Nano Banana, and Gemini 2.0 Flash: called via their APIs with default parameters.
- QWEN Image Edit: default settings from `https://huggingface.co/Qwen/Qwen-Image-Edit`.
- InstructPix2Pix (IP2P): settings from `https://github.com/timothybrooks/instruct-pix2pix`.
- Magicbrush: same settings as IP2P; model weights from `https://huggingface.co/vinesmsuic/magicbrush-jul7`.
- UltraEdit: settings from `https://github.com/HaozheZhao/UltraEdit`; we apply a black mask since no explicit mask is provided.
- AnyEdit: repository at `https://github.com/weichow23/AnySD/tree/9e7d36ef88e237b527695efc90b1abc18fa51218` with `edit_type` set to `general`.
- Step1X-Edit: repository at `https://github.com/stepfun-ai/Step1X-Edit`; weights at `https://huggingface.co/stepfun-ai/Step1X-Edit`.

---

**Algorithm 6** Evaluate Position Change

---

**Require:** base $B$, target $T$, target object name $a$, reference object $r$, position $p$
**Ensure:** success flag
  1: $B_a \leftarrow \text{DETECT}(B, a, \tau);$    $T_a \leftarrow \text{DETECT}(T, a, \tau)$
  2: $B_r \leftarrow \text{DETECT}(B, r, \tau);$    $T_r \leftarrow \text{DETECT}(T, r, \tau)$
  3: **if** $T_a = \emptyset \ \vee \ T_r = \emptyset$ **then**
  4:     **return** false
  5: **end if**
  6: **if** $|T_a| > |B_a|$ **then**
  7:     **return** false                                                  ▷ No count inflation
  8: **end if**
  9: Select max logits boxes $t \in T_a$, $u \in T_r$
 10: $(x_t, y_t) \leftarrow \text{CENTER}(t);$    $(x_u, y_u) \leftarrow \text{CENTER}(u)$
 11: **if** $p = \text{left}$ **then**
 12:     **return** $x_t < x_u - \varepsilon_x$
 13: **end if**
 14: **if** $p = \text{right}$ **then**
 15:     **return** $x_t > x_u + \varepsilon_x$
 16: **end if**
 17: **if** $p = \text{above}$ **then**
 18:     **return** $y_t < y_u - \varepsilon_y$
 19: **end if**
 20: **if** $p = \text{below}$ **then**
 21:     **return** $y_t > y_u + \varepsilon_y$
 22: **end if**
 23: **return** false

---

**Algorithm 7** Evaluate Count Change

---

**Require:** target $T$, name $o$, requested count $c^*$
**Ensure:** success flag
  1: $\hat{c} \leftarrow |\text{DETECT}(T, o)|$
  2: **return** $(\hat{c} = c^*)$

---

- OmniGen: repository at `https://github.com/VectorSpaceLab/OmniGen`.
- FLUX: default settings from `https://huggingface.co/black-forest-labs/FLUX.1-Kontext-dev`.

**Modes.** For clarity in the paper: we report both **Multipass** and **Complex Editing** (renamed from *singlepass* for consistency with the rest of the paper).

**Reproducibility Notes.** Prompts are provided in full (Section M.2); thresholds are specified above. Grounding uses SwinT-OGC weights; consistency uses DINOv3 ViT-B/16; the quality head follows our RAHF implementation, and HPSv3 is included when available. All other parameters are left at defaults.

---

**Algorithm 8** Evaluate Color Alter

---

**Require:** target image $T$, object name $o$, color $k$
 1: **return** VLMYESNO($T$, "Is the $o$ $k$?")

---

---

**Algorithm 9** Evaluate Material Alter

---

**Require:** target image $T$, object name $o$, material $m$
**Ensure:** success flag
 1: **return** VLMYESNO($T$, "Is the $o$ made of $m$?")

---

---

**Algorithm 10** Evaluate Text Change

---

**Require:** target $T$, desired text $t^*$ (optionally object name)
**Ensure:** success flag
 1: $t \leftarrow$ VLMTEXT($T$)
 2: Normalize $t$ and $t^*$ (case, punctuation, whitespace)
 3: **return** TEXT-MATCH($t, t^*$)

---

---

**Algorithm 11** Evaluate Background Change

---

**Require:** target $T$, category $g$
**Ensure:** success flag
 1: **return** VLMYESNO($T$, "Does the background show $g$?")

---

Table 15: Image success rates and overall task success rates across turns. (Multi-turn model)

| Model | Image Success Rate | | | Overall Task Rate | | |
|---|---|---|---|---|---|---|
| | T1 | T2 | T3 | T1 | T2 | T3 |
| Seedream 4.0 | 75.93 | 55.58 | 41.59 | 75.93 | 75.58 | 76.11 |
| Nano Banana | 70.70 | 50.66 | 35.35 | 70.70 | 72.59 | 68.24 |
| GPT-Image-1 | 73.12 | 54.89 | 38.35 | 73.12 | 74.44 | 73.12 |
| FLUX.1-Kontext-max | 69.49 | 46.89 | 31.83 | 69.49 | 69.11 | 70.43 |
| Gemini 2.0 Flash | 68.07 | 45.96 | 28.42 | 68.07 | 67.72 | 68.42 |
| Qwen-Image-Edit | 72.90 | 44.06 | 22.55 | 72.90 | 62.94 | 56.12 |
| Step1X-Edit | 61.89 | 34.97 | 17.83 | 61.89 | 59.09 | 53.32 |
| FLUX.1-Kontext-dev | 59.97 | 32.69 | 16.61 | 59.97 | 56.29 | 51.40 |
| OmniGen | 54.72 | 24.48 | 10.66 | 54.72 | 48.60 | 42.48 |
| UltraEdit | 51.37 | 17.70 | 6.36 | 50.52 | 36.54 | 31.47 |
| AnyEdit | 41.07 | 16.32 | 7.22 | 40.03 | 39.34 | 40.56 |
| MagicBrush | 42.31 | 15.73 | 4.90 | 42.31 | 40.73 | 41.26 |
| IP2P | 37.41 | 10.66 | 2.80 | 37.41 | 32.87 | 34.27 |

Table 16: Task success rates (%) across five instruction types and three turns (multi-turn mode).

| Model | Subject Replace | | | Subject Remove | | | Material Alter | | | Color Alter | | | Subject Add | | |
|---|---|---|---|---|---|---|---|---|---|---|---|---|---|---|---|
| | T1 | T2 | T3 | T1 | T2 | T3 | T1 | T2 | T3 | T1 | T2 | T3 | T1 | T2 | T3 |
| Seedream 4.0 | 90.74 | 91.23 | 88.89 | 68.92 | 47.69 | 50.00 | 95.31 | 96.00 | 95.77 | 100.00 | 98.59 | 100.00 | 83.08 | 89.61 | 81.52 |
| Nano Banana | 91.84 | 92.31 | 75.47 | 64.18 | 51.61 | 40.35 | 91.94 | 89.36 | 87.50 | 100.00 | 97.18 | 98.11 | 73.02 | 77.94 | 72.41 |
| GPT-Image-1 | 84.31 | 94.64 | 85.71 | 70.77 | 55.93 | 47.37 | 96.83 | 95.65 | 87.88 | 100.00 | 97.06 | 100.00 | 80.95 | 72.46 | 72.41 |
| FLUX.1-Kontext-max | 92.31 | 88.89 | 88.00 | 67.16 | 55.56 | 56.36 | 87.30 | 79.07 | 80.30 | 100.00 | 95.65 | 98.04 | 77.05 | 71.23 | 72.73 |
| Gemini 2.0 Flash | 83.33 | 92.98 | 78.18 | 58.67 | 53.62 | 50.82 | 90.91 | 82.00 | 83.33 | 100.00 | 89.04 | 98.21 | 77.61 | 72.73 | 75.27 |
| Qwen-Image-Edit | 87.04 | 82.46 | 70.91 | 70.67 | 31.88 | 37.70 | 93.94 | 90.00 | 79.17 | 100.00 | 97.26 | 94.74 | 77.61 | 55.84 | 39.78 |
| Step1X-Edit | 90.74 | 96.49 | 67.27 | 53.33 | 30.43 | 21.31 | 95.45 | 80.00 | 87.50 | 100.00 | 100.00 | 91.23 | 64.18 | 57.14 | 45.16 |
| FLUX.1-Kontext-dev | 85.19 | 80.70 | 72.73 | 54.67 | 42.03 | 32.79 | 84.85 | 74.00 | 73.61 | 100.00 | 98.63 | 94.74 | 67.16 | 61.04 | 39.78 |
| OmniGen | 88.89 | 84.21 | 58.18 | 46.67 | 21.74 | 19.67 | 84.85 | 72.00 | 70.83 | 100.00 | 90.41 | 91.23 | 53.73 | 51.95 | 37.63 |
| UltraEdit | 88.89 | 63.16 | 38.18 | 26.67 | 5.80 | 6.56 | 87.88 | 66.00 | 63.89 | 98.21 | 80.82 | 78.95 | 38.81 | 23.38 | 9.68 |
| AnyEdit | 74.07 | 66.67 | 61.82 | 37.33 | 39.13 | 36.07 | 78.79 | 68.00 | 68.06 | 78.57 | 68.49 | 78.95 | 22.39 | 38.96 | 25.81 |
| MagicBrush | 83.33 | 75.44 | 63.64 | 28.00 | 18.84 | 18.03 | 83.33 | 86.00 | 80.56 | 94.64 | 87.67 | 91.23 | 37.31 | 41.56 | 37.63 |
| IP2P | 75.93 | 66.67 | 65.45 | 25.33 | 8.70 | 18.03 | 74.24 | 70.00 | 65.28 | 87.50 | 82.19 | 75.44 | 23.88 | 28.57 | 19.35 |

## N  ADDITIONAL EVALUATION RESULTS

In this section, we provide extended evaluation results. We separate the analysis into two modes: *multi-turn editing* and *complex editing*. Each mode is evaluated across three aspects: instruction following, consistency, and quality.

For the multi-turn editing mode, the overall instruction-following success rate is reported in Table 15, while success rates for individual instruction types appear in Tables 16 and 17. Consistency results are summarized in Table 21. We also observed that some input images are non-square after resizing, which can leave black padding on the top/bottom or left/right edges. Certain editing models, such as GPT-Image-1 and Qwen-Image-Edit, attempt to fill these areas, whereas others preserve them. To account for this, we separately report consistency for square (Table 22) and non-square inputs (Table 23). The conclusions remain consistent with the overall evaluation. Quality results for multi-turn editing are presented in Table 25.

For the complex editing mode, the overall instruction-following success rate is shown in Table 18, and per-instruction-type results are in Tables 19 and 20. Consistency and quality results are reported in Tables 24 and 26, respectively.

In consistency table, p99 means $99\%$ quantile of luminance value, and p999 means $99.9\%$ quantile of luminance value.

Table 17: Task success rates (%) across four instrcution types and three turns (multi-turn mode).

| Model | Text Change | | | Position Change | | | Count Change | | | Background Change | | |
|---|---|---|---|---|---|---|---|---|---|---|---|---|
| | T1 | T2 | T3 | T1 | T2 | T3 | T1 | T2 | T3 | T1 | T2 | T3 |
| Seedream 4.0 | 95.31 | 97.14 | 96.23 | 39.22 | 39.68 | 48.94 | 18.06 | 19.30 | 20.00 | 98.46 | 96.36 | 93.06 |
| Nano Banana | 83.33 | 86.36 | 81.25 | 20.00 | 47.37 | 45.24 | 23.19 | 9.62 | 10.91 | 96.67 | 94.44 | 90.00 |
| GPT-Image-1 | 88.33 | 97.01 | 97.96 | 31.11 | 40.68 | 50.00 | 11.27 | 18.52 | 18.18 | 98.39 | 94.44 | 91.30 |
| FLUX.1-Kontext-max | 80.36 | 86.15 | 82.69 | 18.75 | 38.18 | 44.44 | 8.82 | 9.09 | 12.28 | 96.88 | 92.59 | 92.54 |
| Gemini 2.0 Flash | 90.32 | 94.29 | 96.30 | 21.15 | 28.57 | 31.91 | 10.96 | 7.14 | 10.00 | 86.15 | 83.64 | 80.56 |
| Qwen-Image-Edit | 98.44 | 92.86 | 72.22 | 21.15 | 34.92 | 33.33 | 12.33 | 1.72 | 0.00 | 98.46 | 80.00 | 77.78 |
| Step1X-Edit | 60.94 | 51.43 | 44.44 | 13.46 | 31.75 | 27.08 | 0.00 | 1.72 | 1.67 | 87.69 | 87.27 | 83.33 |
| FLUX.1-Kontext-dev | 50.00 | 41.43 | 27.78 | 15.38 | 26.98 | 33.33 | 0.00 | 1.72 | 5.00 | 90.77 | 80.00 | 77.78 |
| OmniGen | 29.69 | 35.71 | 18.52 | 17.31 | 22.22 | 20.83 | 5.48 | 5.17 | 0.00 | 76.92 | 56.36 | 56.94 |
| UltraEdit | 28.12 | 15.71 | 7.41 | 21.15 | 36.51 | 35.42 | 5.48 | 6.90 | 5.00 | 75.38 | 38.18 | 43.06 |
| AnyEdit | 3.12 | 10.00 | 11.11 | 21.15 | 25.40 | 27.08 | 0.00 | 1.72 | 1.67 | 56.92 | 40.00 | 52.78 |
| MagicBrush | 7.81 | 12.86 | 3.70 | 19.23 | 15.87 | 20.83 | 0.00 | 0.00 | 3.33 | 43.08 | 34.55 | 43.06 |
| IP2P | 1.56 | 8.57 | 5.56 | 13.46 | 15.87 | 25.00 | 1.37 | 0.00 | 5.00 | 47.69 | 20.00 | 31.94 |

Table 18: Image rates, overall task rates, and marginal means across three turns (complex mode).

| Model | Image Success Rate | | | Overall Task Rate | | | Marginal Task Rate | | |
|---|---|---|---|---|---|---|---|---|---|
| | T1 | T2 | T3 | T1 | T2 | T3 | T1 | T2 | T3 |
| GPT-Image-1 | 73.08 | 48.45 | 28.78 | 73.08 | 69.77 | 68.25 | 73.08 | 69.98 | 70.19 |
| Nano Banana | 71.46 | 46.56 | 28.14 | 71.46 | 68.83 | 67.27 | 71.46 | 69.03 | 67.21 |
| Gemini 2.0 Flash | 67.43 | 40.63 | 21.89 | 67.43 | 64.54 | 61.94 | 67.43 | 64.80 | 62.17 |
| Qwen-Image-Edit | 72.90 | 46.15 | 27.62 | 72.90 | 69.23 | 68.07 | 72.90 | 67.83 | 67.66 |
| Step1X-Edit | 61.36 | 32.34 | 15.73 | 61.36 | 57.69 | 55.01 | 61.36 | 55.77 | 54.02 |
| FLUX.1-Kontext-dev | 60.14 | 33.74 | 19.58 | 60.14 | 59.53 | 57.87 | 60.14 | 57.52 | 56.99 |
| OmniGen | 54.55 | 23.43 | 11.01 | 54.55 | 50.96 | 49.83 | 54.55 | 48.95 | 47.90 |
| AnyEdit | 39.86 | 10.31 | 2.80 | 39.86 | 34.79 | 34.27 | 39.86 | 31.47 | 31.64 |
| UltraEdit | 50.70 | 22.03 | 8.22 | 50.70 | 48.34 | 46.62 | 50.70 | 47.73 | 47.38 |
| MagicBrush | 42.13 | 14.86 | 4.55 | 42.13 | 38.46 | 38.81 | 42.13 | 38.64 | 39.34 |
| IP2P | 37.24 | 12.41 | 2.80 | 37.24 | 37.76 | 35.14 | 37.24 | 38.46 | 34.62 |

Table 19: Success rates (%) for five instruction types across three turns (complex mode).

| Model | Subject Replace | | | Subject Remove | | | Material Alter | | | Color Alter | | | Subject Add | | |
|---|---|---|---|---|---|---|---|---|---|---|---|---|---|---|---|
| | T1 | T2 | T3 | T1 | T2 | T3 | T1 | T2 | T3 | T1 | T2 | T3 | T1 | T2 | T3 |
| GPT-Image-1 | 82.22 | 80.65 | 78.99 | 70.31 | 63.87 | 58.54 | 96.49 | 90.20 | 84.66 | 100.00 | 97.27 | 91.88 | 81.67 | 70.73 | 69.00 |
| Nano Banana | 91.67 | 88.89 | 79.33 | 67.16 | 55.83 | 59.88 | 92.59 | 85.15 | 83.33 | 97.87 | 98.15 | 94.19 | 69.84 | 71.21 | 67.77 |
| Gemini 2.0 Flash | 85.19 | 82.88 | 75.90 | 57.33 | 55.56 | 54.15 | 93.94 | 75.00 | 74.87 | 100.00 | 96.90 | 93.01 | 70.15 | 65.97 | 66.67 |
| Qwen-Image-Edit | 87.04 | 86.49 | 80.72 | 70.67 | 58.33 | 54.63 | 93.94 | 86.21 | 85.64 | 100.00 | 99.22 | 98.39 | 77.61 | 75.69 | 73.84 |
| Step1X-Edit | 90.74 | 84.68 | 76.51 | 52.00 | 41.67 | 42.93 | 93.94 | 82.76 | 79.26 | 100.00 | 93.02 | 90.86 | 64.18 | 59.03 | 54.01 |
| FLUX.1-Kontext-dev | 85.19 | 82.88 | 74.10 | 54.67 | 47.92 | 40.00 | 86.36 | 75.00 | 76.06 | 100.00 | 99.22 | 98.39 | 67.16 | 68.75 | 63.29 |
| OmniGen | 88.89 | 82.88 | 75.90 | 46.67 | 39.58 | 41.95 | 86.36 | 73.28 | 72.34 | 100.00 | 96.12 | 93.01 | 53.73 | 48.61 | 49.37 |
| AnyEdit | 66.67 | 58.56 | 49.40 | 33.33 | 20.14 | 22.44 | 77.27 | 76.72 | 70.21 | 78.57 | 68.99 | 72.04 | 28.36 | 22.92 | 21.10 |
| UltraEdit | 88.89 | 87.39 | 78.31 | 26.67 | 30.56 | 31.22 | 87.88 | 79.31 | 77.66 | 98.21 | 93.02 | 90.32 | 38.81 | 46.53 | 43.46 |
| MagicBrush | 83.33 | 74.77 | 69.28 | 28.00 | 27.78 | 29.27 | 83.33 | 69.83 | 73.94 | 92.86 | 84.50 | 83.87 | 37.31 | 35.42 | 32.91 |
| IP2P | 75.93 | 73.87 | 61.45 | 25.33 | 31.25 | 24.88 | 71.21 | 63.79 | 59.57 | 87.50 | 82.95 | 79.03 | 23.88 | 31.94 | 29.11 |

Table 20: Success rates (%) for four instruction types across three turns (complex mode).

| Model | Text Change | | | Position Change | | | Count Change | | | Background Change | | |
|---|---|---|---|---|---|---|---|---|---|---|---|---|
| | T1 | T2 | T3 | T1 | T2 | T3 | T1 | T2 | T3 | T1 | T2 | T3 |
| GPT-Image-1 | 87.50 | 92.24 | 93.75 | 30.95 | 22.11 | 25.36 | 13.11 | 10.38 | 13.12 | 98.15 | 96.08 | 93.37 |
| Nano Banana | 84.21 | 85.34 | 84.24 | 24.44 | 26.00 | 22.54 | 20.34 | 14.15 | 13.75 | 98.15 | 93.40 | 93.71 |
| Gemini 2.0 Flash | 85.94 | 86.57 | 84.04 | 17.31 | 23.68 | 16.67 | 11.11 | 10.77 | 6.84 | 90.77 | 84.17 | 80.73 |
| Qwen-Image-Edit | 98.44 | 97.01 | 93.62 | 21.15 | 22.61 | 24.54 | 12.33 | 3.05 | 3.66 | 98.46 | 95.83 | 93.75 |
| Step1X-Edit | 57.81 | 59.70 | 52.13 | 15.38 | 21.74 | 25.15 | 1.37 | 3.05 | 1.57 | 86.15 | 80.00 | 73.44 |
| FLUX.1-Kontext-dev | 50.00 | 48.51 | 47.87 | 15.38 | 26.96 | 27.61 | 0.00 | 1.53 | 3.66 | 90.77 | 90.00 | 88.54 |
| OmniGen | 28.12 | 32.84 | 27.13 | 15.38 | 21.74 | 17.18 | 6.85 | 1.53 | 2.09 | 75.38 | 70.00 | 69.79 |
| AnyEdit | 3.12 | 5.22 | 7.98 | 25.00 | 26.96 | 29.45 | 1.37 | 1.53 | 1.57 | 56.92 | 44.17 | 40.62 |
| UltraEdit | 29.69 | 16.42 | 11.70 | 21.15 | 26.09 | 22.70 | 5.48 | 2.29 | 3.14 | 75.38 | 65.00 | 64.06 |
| MagicBrush | 7.81 | 4.48 | 6.38 | 19.23 | 22.61 | 17.18 | 0.00 | 0.00 | 5.24 | 43.08 | 36.67 | 35.42 |
| IP2P | 3.12 | 5.22 | 6.38 | 13.46 | 20.00 | 20.25 | 1.37 | 2.29 | 1.57 | 47.69 | 37.50 | 38.54 |

Table 21: Consistency scores (%) across DINOv3-based and L1-based object/background metrics. (multi-turn mode)

| Model | Object DINOv3 Consistency | | | Background DINOv3 Consistency | | | Object $1 - L_1$ Consistency | | | Background $1 - L_1$ Consistency | | |
|---|---|---|---|---|---|---|---|---|---|---|---|---|
| | T1 | T2 | T3 | T1 | T2 | T3 | T1 | T2 | T3 | T1 | T2 | T3 |
| Seedream 4.0 | 89.50 | 83.68 | 81.18 | 95.52 | 92.38 | 90.54 | 93.31 | 88.76 | 86.22 | 94.30 | 89.04 | 85.00 |
| Nano Banana | 90.17 | 85.25 | 84.38 | 97.65 | 95.70 | 94.58 | 92.39 | 90.00 | 89.10 | 95.95 | 94.70 | 93.88 |
| GPT-Image-1 | 73.26 | 68.80 | 67.21 | 88.74 | 86.76 | 83.78 | 79.65 | 78.32 | 77.60 | 78.07 | 76.53 | 75.79 |
| FLUX.1-Kontext-max | 90.91 | 86.66 | 83.68 | 96.95 | 95.15 | 93.11 | 94.16 | 91.26 | 89.46 | 95.88 | 93.43 | 91.46 |
| Gemini 2.0 Flash | 85.53 | 77.12 | 72.02 | 95.63 | 93.07 | 89.74 | 90.72 | 86.32 | 84.11 | 95.04 | 93.15 | 91.73 |
| Qwen-Image-Edit | 77.12 | 71.56 | 68.51 | 91.31 | 89.47 | 87.45 | 83.48 | 79.15 | 76.32 | 84.57 | 81.18 | 78.43 |
| Step1X-Edit | 88.17 | 81.65 | 77.33 | 97.34 | 95.40 | 93.09 | 93.92 | 90.64 | 88.80 | 98.24 | 97.10 | 95.73 |
| FLUX.1-Kontext-dev | 92.66 | 87.92 | 85.29 | 97.97 | 96.55 | 95.14 | 94.39 | 91.59 | 89.91 | 96.36 | 95.06 | 94.13 |
| OmniGen | 88.34 | 80.77 | 73.64 | 97.66 | 96.08 | 94.21 | 93.87 | 91.02 | 89.43 | 97.44 | 97.00 | 96.34 |
| UltraEdit | 78.81 | 75.11 | 72.24 | 94.80 | 93.89 | 92.57 | 91.86 | 90.47 | 89.65 | 97.12 | 96.62 | 96.19 |
| AnyEdit | 82.02 | 73.41 | 63.04 | 90.82 | 84.42 | 77.17 | 92.52 | 88.96 | 84.97 | 93.72 | 89.98 | 86.05 |
| MagicBrush | 79.70 | 70.71 | 65.46 | 94.22 | 91.81 | 88.27 | 91.13 | 87.13 | 85.56 | 96.31 | 94.52 | 92.87 |
| IP2P | 68.24 | 56.83 | 48.01 | 85.47 | 79.89 | 72.59 | 84.44 | 79.74 | 77.21 | 91.31 | 87.21 | 83.51 |

Table 22: Consistency and $1 - L_1$ metrics across three turns (multi-turn mode for square image).

| Model | Object DINOv3 Consistency (Mean) | | | Background DINOv3 Consistency | | | Object $L_1$ Consistency (Mean) | | | Background $L_1$ Consistency | | |
|---|---|---|---|---|---|---|---|---|---|---|---|---|
| | T1 | T2 | T3 | T1 | T2 | T3 | T1 | T2 | T3 | T1 | T2 | T3 |
| Seedream 4.0 | 89.04 | 85.39 | 80.74 | 95.76 | 92.43 | 88.27 | 91.66 | 88.02 | 84.43 | 95.34 | 92.45 | 89.69 |
| Nano Banana | 88.68 | 85.19 | 83.31 | 96.88 | 93.79 | 92.41 | 89.73 | 87.64 | 88.48 | 94.21 | 93.77 | 93.36 |
| GPT-Image-1 | 73.62 | 71.27 | 67.27 | 90.00 | 85.72 | 82.16 | 80.33 | 78.80 | 79.88 | 85.69 | 81.86 | 80.85 |
| FLUX.1-Kontext-max | 91.34 | 89.90 | 86.04 | 95.99 | 94.18 | 90.10 | 93.81 | 92.80 | 90.58 | 97.27 | 95.77 | 93.94 |
| Gemini 2.0 Flash | 84.16 | 79.91 | 71.13 | 91.50 | 91.83 | 87.07 | 88.98 | 87.07 | 85.13 | 95.19 | 93.35 | 93.34 |
| Qwen-Image-Edit | 75.36 | 72.68 | 68.85 | 88.50 | 88.98 | 82.71 | 81.66 | 78.47 | 77.04 | 91.16 | 87.85 | 85.78 |
| Step1X-Edit | 87.40 | 84.03 | 78.27 | 97.43 | 92.42 | 88.15 | 93.14 | 91.57 | 89.40 | 98.10 | 97.12 | 96.01 |
| FLUX.1-Kontext-dev | 92.55 | 88.92 | 84.04 | 96.49 | 93.57 | 92.19 | 92.83 | 90.61 | 88.34 | 96.91 | 95.74 | 94.73 |
| OmniGen | 89.41 | 84.51 | 77.77 | 97.34 | 93.13 | 86.64 | 93.26 | 91.48 | 89.53 | 95.79 | 96.44 | 95.36 |
| UltraEdit | 79.43 | 76.51 | 71.54 | 92.83 | 89.88 | 86.54 | 91.74 | 90.16 | 89.24 | 95.87 | 95.14 | 94.63 |
| AnyEdit | 82.25 | 72.40 | 53.59 | 86.12 | 78.60 | 70.09 | 92.33 | 88.00 | 82.10 | 94.36 | 91.68 | 87.55 |
| MagicBrush | 79.02 | 75.60 | 70.07 | 92.07 | 87.33 | 81.08 | 89.67 | 87.53 | 86.90 | 95.29 | 93.88 | 92.38 |
| IP2P | 76.38 | 66.12 | 54.94 | 85.29 | 81.46 | 65.50 | 86.90 | 82.25 | 77.79 | 92.45 | 88.01 | 84.51 |

Table 23: Consistency and $L_1$ metrics across three turns (multi-turn model for unsquared image).

| Model | Object DINOv3 Consistency (Mean) | | | Background DINOv3 Consistency | | | Object $1 - L_1$ Consistency (Mean) | | | Background $1 - L_1$ Consistency | | |
|---|---|---|---|---|---|---|---|---|---|---|---|---|
| | T1 | T2 | T3 | T1 | T2 | T3 | T1 | T2 | T3 | T1 | T2 | T3 |
| Seedream 4.0 | 89.55 | 83.48 | 81.23 | 95.49 | 92.38 | 90.86 | 93.50 | 88.85 | 86.44 | 94.17 | 88.62 | 84.35 |
| Nano Banana | 90.34 | 85.26 | 84.50 | 97.74 | 95.94 | 94.88 | 92.69 | 90.29 | 89.17 | 96.16 | 94.81 | 93.95 |
| GPT-Image-1 | 73.22 | 68.53 | 67.20 | 88.60 | 86.88 | 83.99 | 79.57 | 78.27 | 77.35 | 77.20 | 75.91 | 75.14 |
| FLUX.1-Kontext-max | 90.86 | 86.31 | 83.43 | 97.06 | 95.27 | 93.51 | 94.20 | 91.09 | 89.34 | 95.73 | 93.16 | 91.14 |
| Gemini 2.0 Flash | 85.69 | 76.79 | 72.12 | 96.14 | 93.23 | 90.10 | 90.91 | 86.23 | 83.99 | 95.02 | 93.12 | 91.51 |
| Qwen-Image-Edit | 77.32 | 71.43 | 68.47 | 91.65 | 89.53 | 88.10 | 83.68 | 79.23 | 76.23 | 83.78 | 80.37 | 77.43 |
| Step1X-Edit | 88.26 | 81.37 | 77.22 | 97.33 | 95.77 | 93.77 | 94.01 | 90.53 | 88.73 | 98.26 | 97.09 | 95.70 |
| FLUX.1-Kontext-dev | 92.67 | 87.80 | 85.44 | 98.15 | 96.92 | 95.54 | 94.57 | 91.70 | 90.10 | 96.30 | 94.98 | 94.05 |
| OmniGen | 88.22 | 80.33 | 73.14 | 97.70 | 96.45 | 95.25 | 93.94 | 90.97 | 89.42 | 97.64 | 97.07 | 96.47 |
| UltraEdit | 78.74 | 74.95 | 72.32 | 95.04 | 94.39 | 93.40 | 91.87 | 90.51 | 89.70 | 97.26 | 96.80 | 96.40 |
| AnyEdit | 82.00 | 73.53 | 64.18 | 91.39 | 85.51 | 78.14 | 92.54 | 89.07 | 85.31 | 93.64 | 89.78 | 85.84 |
| MagicBrush | 79.78 | 70.14 | 64.90 | 94.48 | 92.37 | 89.27 | 91.30 | 87.08 | 85.40 | 96.44 | 94.59 | 92.94 |
| IP2P | 67.32 | 55.75 | 47.18 | 85.50 | 79.69 | 73.57 | 84.17 | 79.45 | 77.14 | 91.18 | 87.12 | 83.38 |

Table 24: Consistency scores (%) across object/background DINOv3 and $L_1$ metrics (complex mode).

| Model | Object DINOv3 | | | Object $1 - L_1$ | | | Background DINOv3 | | | Background $1 - L_1$ | | |
|---|---|---|---|---|---|---|---|---|---|---|---|---|
| | T1 | T2 | T3 | T1 | T2 | T3 | T1 | T2 | T3 | T1 | T2 | T3 |
| GPT-Image-1 | 73.23 | 70.02 | 67.57 | 79.52 | 78.03 | 77.29 | 88.72 | 86.77 | 84.79 | 77.96 | 77.38 | 76.51 |
| Nano Banana | 89.23 | 87.20 | 86.46 | 92.15 | 91.08 | 90.40 | 97.39 | 96.69 | 95.38 | 96.39 | 95.75 | 95.33 |
| Gemini 2.0 Flash | 85.41 | 80.37 | 77.38 | 90.60 | 88.45 | 86.53 | 96.03 | 94.18 | 92.83 | 95.03 | 94.77 | 93.46 |
| Qwen-Image-Edit | 77.12 | 76.09 | 76.69 | 83.48 | 83.08 | 83.11 | 91.31 | 91.32 | 90.51 | 84.57 | 84.93 | 85.35 |
| Step1X-Edit | 88.14 | 85.31 | 84.38 | 93.93 | 92.34 | 92.11 | 97.34 | 96.37 | 95.44 | 98.24 | 98.02 | 98.04 |
| FLUX.1-Kontext-dev | 92.66 | 90.30 | 89.19 | 94.39 | 92.79 | 91.61 | 97.97 | 96.74 | 95.40 | 96.36 | 95.57 | 94.04 |
| OmniGen | 88.37 | 85.15 | 83.14 | 93.88 | 92.46 | 91.06 | 97.62 | 97.10 | 96.07 | 97.45 | 97.58 | 97.40 |
| AnyEdit | 81.90 | 82.94 | 84.92 | 92.34 | 92.43 | 93.72 | 90.97 | 92.63 | 93.78 | 94.15 | 95.11 | 95.87 |
| UltraEdit | 78.81 | 72.75 | 71.67 | 91.86 | 89.51 | 88.99 | 94.80 | 93.01 | 92.03 | 97.12 | 96.35 | 96.02 |
| MagicBrush | 79.70 | 75.64 | 75.53 | 91.13 | 89.23 | 88.69 | 94.22 | 94.34 | 93.13 | 96.31 | 96.14 | 95.83 |
| IP2P | 68.24 | 67.31 | 69.49 | 84.44 | 82.93 | 83.72 | 85.47 | 85.88 | 86.45 | 91.31 | 89.94 | 90.19 |

Table 25: Human preference scores, p999, and p99 across three turns (multi-turn mode).

| Model | Human Preference Score | | | p999 | | | p99 | | |
|---|---|---|---|---|---|---|---|---|---|
| | T1 | T2 | T3 | T1 | T2 | T3 | T1 | T2 | T3 |
| GPT-Image-1 | 6.6519 | 6.5898 | 6.5609 | 84.73 | 82.96 | 82.50 | 74.38 | 71.91 | 70.54 |
| Nano Banana | 4.9431 | 5.1179 | 5.2638 | 89.67 | 90.27 | 90.56 | 81.31 | 82.01 | 82.09 |
| Gemini 2.0 Flash | 4.4386 | 4.2332 | 4.0677 | 90.95 | 92.21 | 92.95 | 83.08 | 84.79 | 86.24 |
| Qwen-Image-Edit | 5.8591 | 5.7198 | 5.1502 | 89.16 | 97.20 | 99.04 | 79.60 | 90.51 | 95.28 |
| Step1X-Edit | 4.0577 | 3.3443 | 2.7569 | 92.81 | 95.39 | 97.27 | 85.10 | 88.46 | 91.21 |
| FLUX.1-Kontext-dev | 5.1192 | 5.0701 | 5.0354 | 90.76 | 91.94 | 93.21 | 82.05 | 83.03 | 84.58 |
| OmniGen | 4.6099 | 4.0743 | 3.4958 | 93.15 | 95.24 | 96.71 | 85.55 | 88.24 | 90.50 |
| AnyEdit | 3.6609 | 2.8017 | 1.9457 | 86.70 | 86.70 | 86.71 | 77.54 | 76.53 | 75.82 |
| UltraEdit | 4.7934 | 4.6806 | 4.3598 | 92.71 | 94.19 | 95.82 | 85.42 | 86.76 | 88.34 |
| MagicBrush | 3.8465 | 3.0805 | 2.3606 | 91.49 | 93.33 | 94.20 | 83.42 | 84.70 | 85.32 |
| IP2P | 3.2020 | 2.3779 | 1.4418 | 89.61 | 91.59 | 92.44 | 81.79 | 83.78 | 84.77 |

Table 26: Updated human preference scores, p999 scores, and p99 scores across three turns (complex mode).

| Model | Human Preference Score | | | p999 | | | p99 | | |
|---|---|---|---|---|---|---|---|---|---|
| | T1 | T2 | T3 | T1 | T2 | T3 | T1 | T2 | T3 |
| GPT-Image-1 | 6.6328 | 6.8428 | 6.9655 | 85.33 | 84.14 | 84.44 | 74.73 | 73.92 | 73.07 |
| Nano Banana | 4.9444 | 5.1700 | 5.3632 | 89.65 | 90.67 | 91.79 | 81.02 | 81.93 | 82.75 |
| Gemini 2.0 Flash | 4.4511 | 4.5428 | 4.5732 | 91.27 | 92.66 | 93.48 | 83.85 | 85.69 | 86.75 |
| Qwen-Image-Edit | 5.8591 | 5.8769 | 5.9155 | 89.16 | 90.92 | 92.23 | 79.60 | 81.36 | 82.62 |
| Step1X-Edit | 4.0534 | 3.9063 | 3.8648 | 92.82 | 93.55 | 94.05 | 85.11 | 85.49 | 86.01 |
| FLUX.1-Kontext-dev | 5.1192 | 5.2446 | 5.4645 | 90.76 | 91.01 | 91.29 | 82.05 | 81.53 | 81.55 |
| OmniGen | 4.5976 | 4.3070 | 3.8122 | 93.15 | 93.74 | 95.65 | 85.56 | 86.28 | 88.57 |
| AnyEdit | 3.7020 | 3.7601 | 3.8382 | 86.47 | 87.16 | 87.43 | 77.82 | 78.60 | 79.12 |
| UltraEdit | 4.7934 | 4.7647 | 4.8117 | 92.71 | 93.06 | 93.24 | 85.42 | 86.09 | 86.47 |
| MagicBrush | 3.8465 | 3.6029 | 3.5523 | 91.49 | 91.52 | 91.64 | 83.42 | 83.31 | 83.07 |
| IP2P | 3.2020 | 3.3552 | 3.5640 | 89.61 | 90.46 | 90.86 | 81.79 | 82.57 | 82.71 |

## O  HUMAN AGREEMENT

The human study was conducted online through Gradio[2]. Annotators were asked to answer a 2-way multiple-choice problem (Yes/No) about an editing instruction, an original image, and an edited image. There were very limited potential participant risks, if they were to be exposed to an image that was disturbing or not safe for work (NSFW). It is because the source images we used were from GEit-Bench Liu et al. (2025), which were not in themselves offensive. Also, our agent already filtered out unsafe images during the first decomposition stage. Furthermore, all edited images from the models were passed through its own NSFW filters which blacked out any potentially unsafe content.

We conducted human study on edits made by four exemplary models—Step1X-Edit, AnyEdit, Gemini-Flash 2.0, and Flux.1-Kontext-dev—on **EdiVal-Bench**, generated by **EdiVal-Agent** as described in Section M.2. For each edit, we collected two human ratings, yielding a total of $572 \times 4 \times 2 = 4{,}576$ annotations. Depending on the prompt (which affected the editing instruction), each annotation took about 1–2 minutes. Raters were recruited online, each holding at least a bachelor's degree. They were shown the original image, the edited image, and the corresponding instruction, and were asked a binary question: *"Evaluate whether the edited image successfully follows the given instruction."*

## P  COUNTING

Among all subtasks, *count_change* is the most challenging. Even the best-performing model (GPT-Image-1) achieves a success rate below 25% at turn 1, while most models remain under 5%. We also provide illustrative examples in Figure 23.

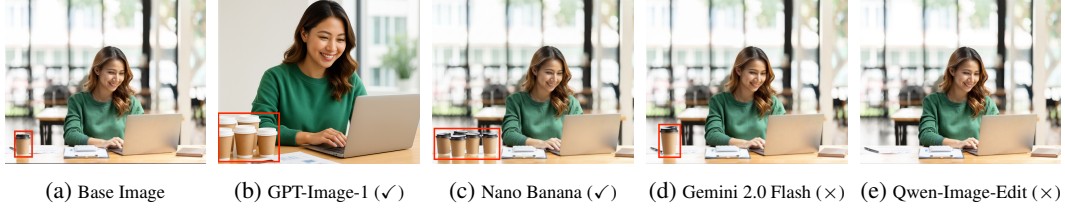

(a) Base Image     (b) GPT-Image-1 (✓)     (c) Nano Banana (✓)     (d) Gemini 2.0 Flash (×)     (e) Qwen-Image-Edit (×)

Figure 23: Example of the *count_change* task: changing the number of paper cups to five.

---

[2] https://www.gradio.app/

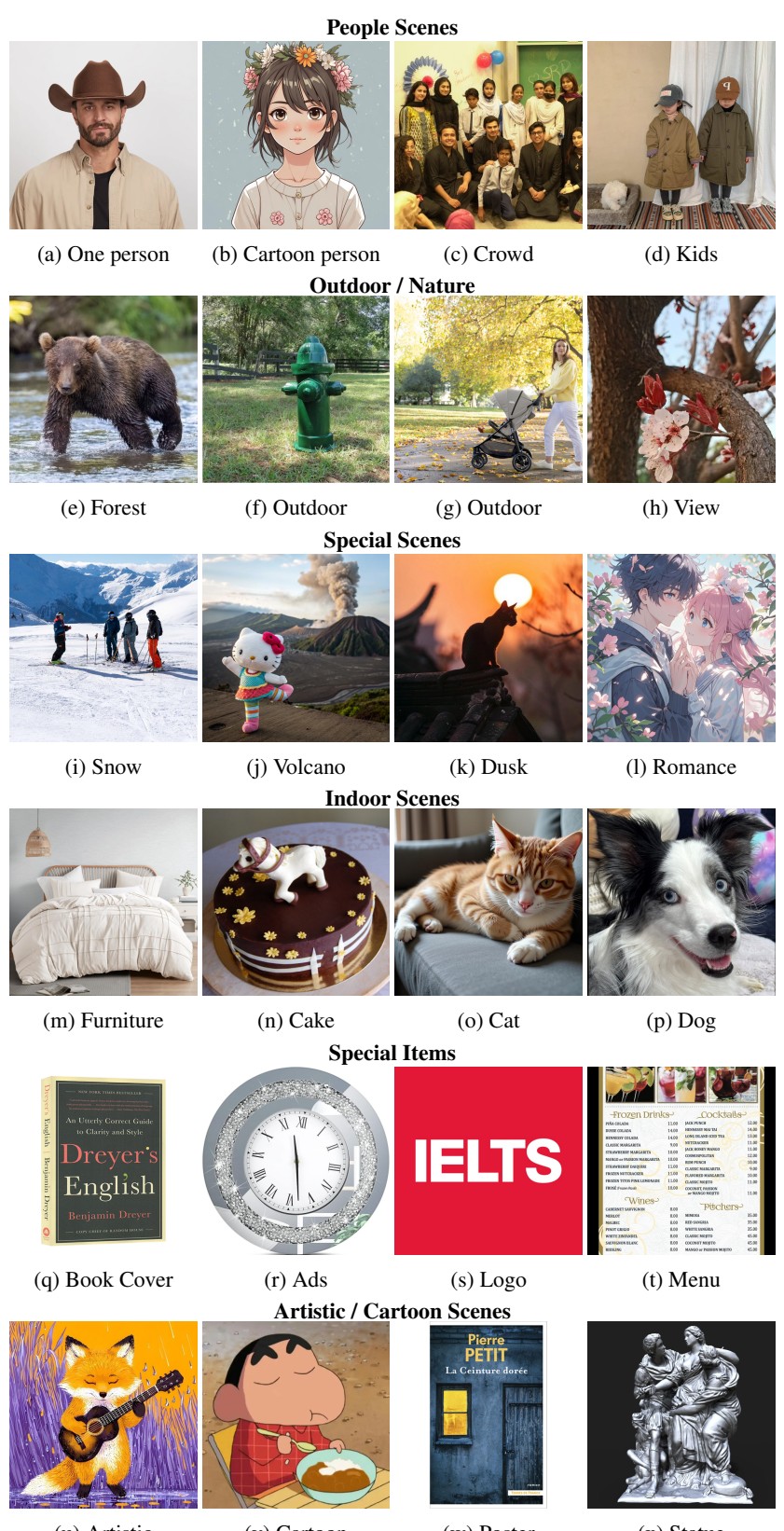

Figure 24: **Scene diversity** of our dataset across multiple environments and content types.

# Q DIVERSITY OF OUR DATASET

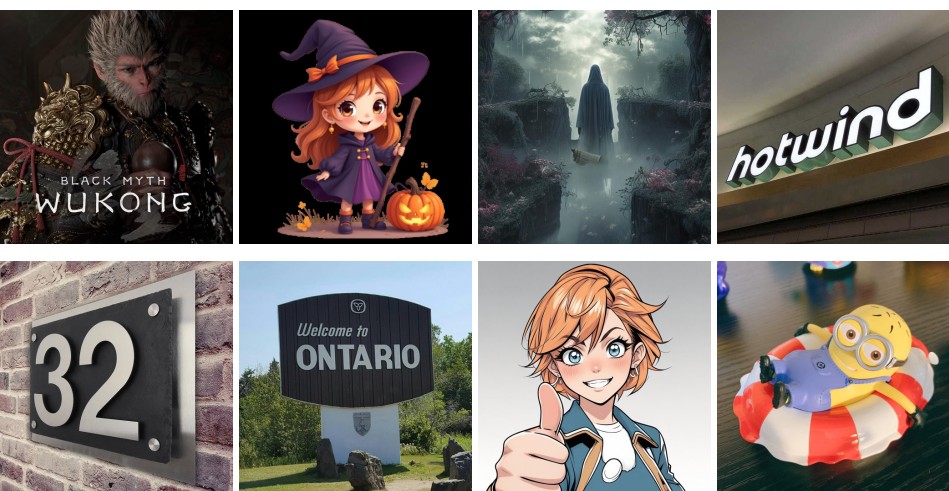

Figure 25: Images of different **lightning** sorted from very dark to very bright.

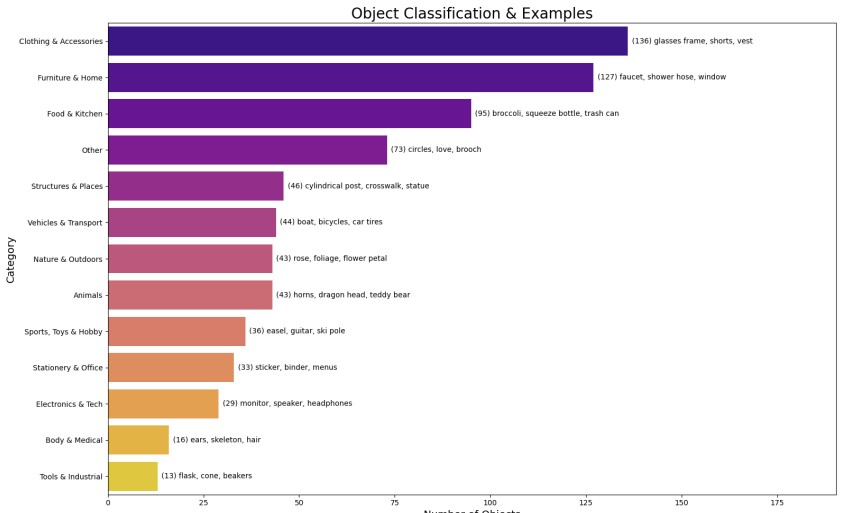

Figure 26: Categories and examples of objects in the base images.

All base images can be found here: the GEdit-Bench's official Huggingface link `https://huggingface.co/datasets/stepfun-ai/GEdit-Bench`.

Our dataset could reflect real-world editing scenarios in the following senses.

**Data Source** We first stress that our images

1. are from real-world user editing cases as stated in GEdit-Bench Liu et al. (2025).
2. include both synthetic and real images
3. are carefully selected by the GEdit team to ensure the diversity.

**Different Scenes** We demonstrate example images of different scenes in Fig. 24 which covers assorted environments and content types, including but not limited to indoor/outdoor, person, special items, and artistic scenes. This diversity helps our dataset to better reflect the real-world user cases.

**Different Lightning**   We demonstrate images of lightning conditions in Fig. 25 sorted from very dark to very bright. Our dataset includes a variety of illumination scenarios that aim to cover the spectrum of very bright to very dark environments. This variation reflects the real-world complex editing scenes.

**Object categories.**   After decomposition, we analyzed and found that there are a total of 724 distinct objects in the dataset. We classify them into 13 categories, ranging from everyday items such as furniture and kitchenware to special entities like vehicles and animals. Specifically, they are (The distribution of these objects is presented in Fig. 26.):

1. Clothing & Accessories
2. Furniture & Home
3. Food & Kitchen
4. Structures & Places
5. Vehicles & Transport
6. Nature & Outdoors
7. Animals
8. Sports, Toys & Hobby
9. Stationery & Office
10. Electronics & Tech
11. Body & Medical
12. Tools & Industrial
13. Other

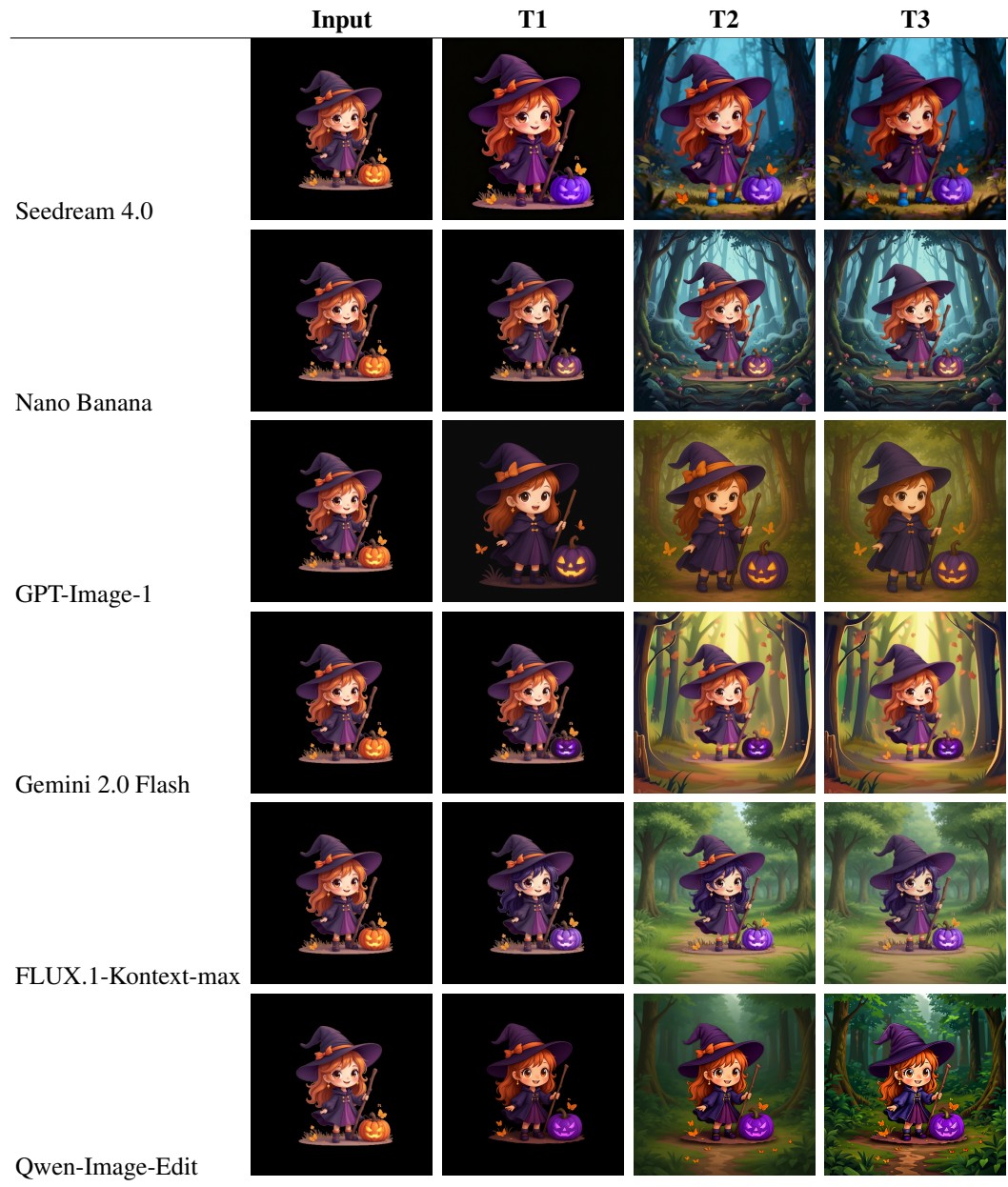

Figure 27: **T1: Change the color of pumpkin to purple; T2: Change the background to forest; T3: Remove fabric orange bow.** Row-wise quality examples for the first six models: Seedream 4.0, Nano Banana, GPT-Image-1, Gemini 2.0 Flash, FLUX.1-Kontext-max, and Qwen-Image-Edit. Each row shows generations for Input and three editing turns.

# R    MORE QUALITY EXAMPLES

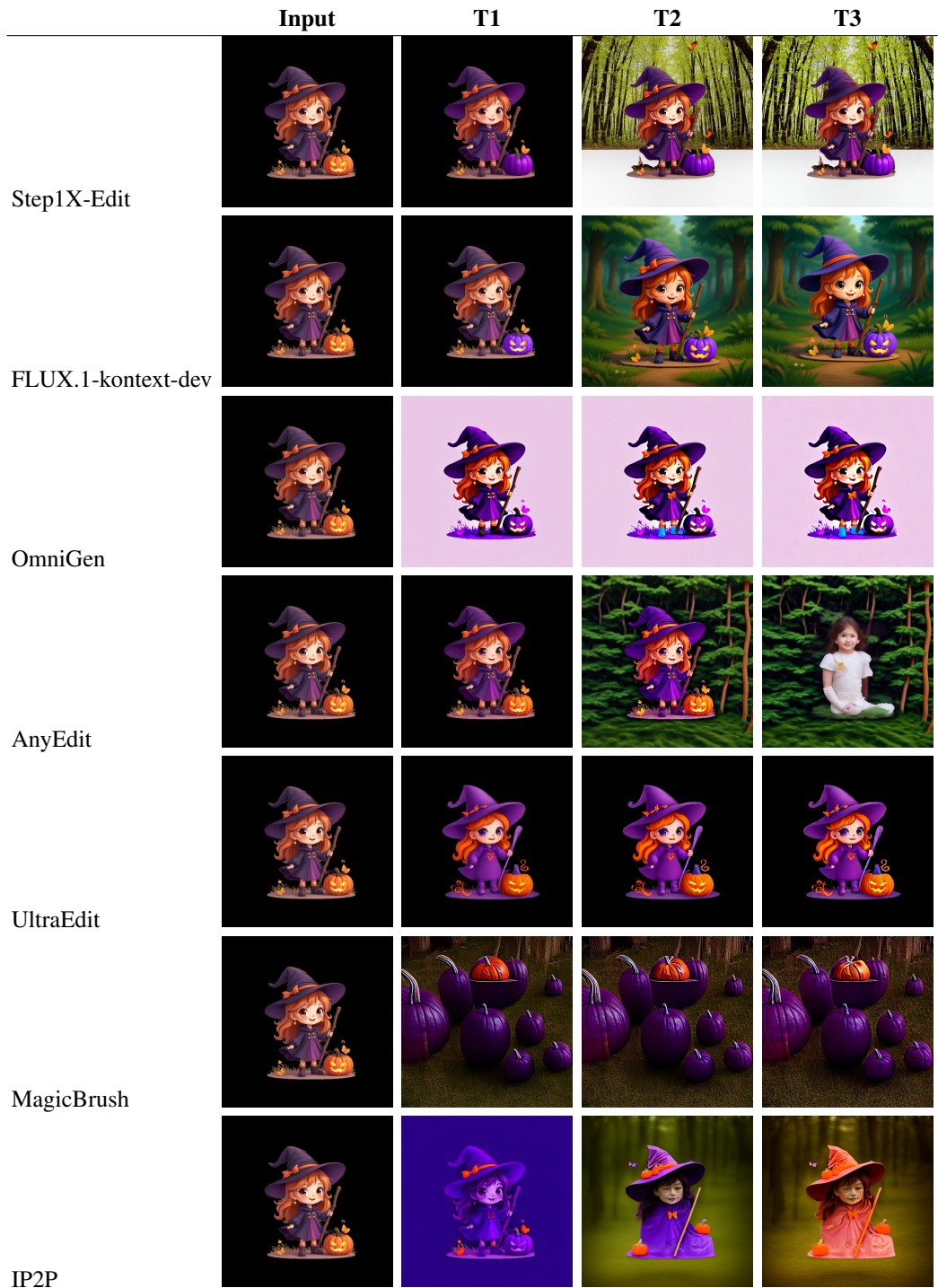

Figure 28: **T1: Change the color of pumpkin to purple; T2: Change the background to forest; T3: Remove fabric orange bow.** Row-wise quality examples for the remaining models: Step1X-Edit, FLUX.1-kontext-dev, OmniGen, AnyEdit, UltraEdit, MagicBrush, and IP2P.

