# OpenReview forum: "EdiVal-Agent: An Object-Centric Framework for  Automated,    Fine-Grained Evaluation of Multi-Turn  Editing"
_ICLR.cc/2026/Conference — ICLR 2026 Poster_

### Official Review · Reviewer_TgZ8 · 2025-10-28

**Soundness:** 3
**Presentation:** 4
**Contribution:** 3
**Rating:** 6
**Confidence:** 4

**Summary:**

The paper introduces EdiVal-Agent, an object-centric, multi-turn framework for evaluating instruction-based image editing. It generates scene-aware instructions and scores edits with three metrics—EdiVal-IF (instruction following via detector+VLM), EdiVal-CC (content consistency for unchanged regions), and EdiVal-VQ (visual quality)—and releases EdiVal-Bench across 9 instruction types and 13 editors.

**Strengths:**

- Clear problem framing & object-centric design: The paper crisply motivates why existing reference-based and VLM-only evaluation is insufficient and positions an object-centric alternative that combines symbolic checks with semantic reasoning.
- Well-specified agentic pipeline with dynamic object pools: The decomposition → instruction generation → evaluation workflow is precise.

**Weaknesses:**

- Tool-dependence and potential evaluator bias: The pipeline fixes Grounding-DINO as the detector, Qwen-2.5-VL as the VLM, and DINOv3/HPSv3 as backbones for CC/VQ. While the authors say the pipeline is tool-agnostic, the reported results (e.g., human agreement advantage) are contingent on this specific stack. Please quantify sensitivity to swapping components (e.g., different detectors/VLMs), or report ablations where each tool is replaced by a strong alternative.
- Metric coupling between IF and CC is under-analyzed: The geometric mean \sqrt{\text{IF}\cdot \text{CC}} is reasonable, but different applications may value preservation much more than instruction compliance (or vice versa). The choice is referenced to prior work but lacks a robustness study: how often do model rankings flip under other aggregations (harmonic/weighted means)?
- Fairness of latency comparison: The table mixes closed-source UI latency with open-source single-GPU latency, which is not directly comparable (unknown batch sizes, backend scaling, caching). The paper acknowledges measurement contexts but still draws speed–quality conclusions. Consider removing cross-setting latency comparisons or normalizing via a controlled local runtime for open models and API budgeted throughput for closed models.
- Multi-turn vs. single-shot comparison design: The conclusion that multi-turn can help when exposure bias is low is plausible, but details on prompt concatenation for single-shot “complex” edits matter (ordering, connectors, explicit constraints like “make X unchanged”). A stricter prompt engineering baseline for single-shot could narrow gaps; please share exact templates and run ablations with stronger single-shot compositions.

**Questions:**

N/A

---

> ### Author Response · Authors · 2025-11-24
> **1/n rebuttal**
>
> [**Highlight:** We updated the pdf with 3 new sections: **3.5 Pareto Front, 3.6 Ablation Study on Complex Editing Compression, and 3.7 Tool Swap Analysis** on Page **10**. We also add more details on **Page 15-23** in the Appendix.]
>
> Thank you very much for your positive review and for recognizing our problem framing and evaluation pipeline. We found your comments highly insightful, and incorporating your suggestions has helped us further strengthen the paper. Below, we try to address your concerns in detail and explain the revisions we are making in response.
>
>
> > W1. Tool-dependence and potential evaluator bias: The pipeline fixes Grounding-DINO as the detector, Qwen-2.5-VL as the VLM, and DINOv3/HPSv3 as backbones for CC/VQ. While the authors say the pipeline is tool-agnostic, the reported results (e.g., human agreement advantage) are contingent on this specific stack. Please quantify sensitivity to swapping components (e.g., different detectors/VLMs), or report ablations where each tool is replaced by a strong alternative.
>
>
> We thank the reviewer for the insightful question and fully agree that we should analyze how swapping different tools at each component affects the final evaluation. In response, we have added **Section 3.7** to systematically study the behavior of tool swapping.
>
> For the **VLM component**, we replace the default Qwen2.5-7B-VL with three strong alternatives: Qwen2-7B-VL, Qwen2.5-32B-VL, and InternVL3-8B.
>
> For **Grounding DINO’s threshold and large-box filter**, our default setting uses a threshold of $0.30$ for *subject\_remove*, $0.40$ for *position\_change*, and $0.35$ for all other tasks. During detection, we also discard any box whose normalized height *and* width are both larger than $0.98$, in order to filter out degenerate predictions that cover almost the entire image. To investigate sensitivity to these choices, we re-run the evaluation under three alternative configurations:
> 1. A global threshold of $0.30$ for all tasks.
> 2. A global threshold of $0.40$ for all tasks.
> 3. The default task-specific thresholds, but with the large-box filter disabled.
>
> For the **detector**, we swap Grounding DINO (default) with OWL-ViT; for the **image feature extractor**, we swap DINOv3 (default) with DINOv2. We focus on the top four models in our benchmark—Seedream 4.0, Nano Banana, GPT-Image-1, and FLUX.1-Kontext-max—and compute:
> - per-turn instruction-following success rates across all $(\text{model}, \text{turn})$ pairs for VLM / threshold / filter / detector swaps, and
> - consistency across $(\text{model}, \text{turn})$ pairs for the DINOv3 vs. DINOv2 feature swap.
>
> The resulting correlations are summarized below:
>
> | Type      | Default          | Variant         | Metric | Pearson | Spearman |
> |-----------|------------------|-----------------|--------|---------|----------|
> | VLM       | Qwen2.5-7B-VL    | Qwen2-7B-VL     | IF     | 0.9544  | 0.9298   |
> | VLM       | Qwen2.5-7B-VL    | Qwen2.5-32B-VL  | IF     | 0.9790  | 0.9544   |
> | VLM       | Qwen2.5-7B-VL    | InternVL3-8B    | IF     | 0.9660  | 0.9228   |
> | Threshold | [0.3, 0.4]       | Threshold = 0.4 | IF     | 0.9817  | 0.9860   |
> | Threshold | [0.3, 0.4]       | Threshold = 0.3 | IF     | 0.9772  | 0.9457   |
> | Filter    | Has filter       | No filter       | IF     | 0.9982  | 0.9930   |
> | Detector  | Grounding DINO   | OWL-ViT         | IF     | 0.8157  | 0.8029   |
> | Feature   | DINOv3           | DINOv2          | CC     | 0.9987  | 1.0000   |
>
>
> As for **EdiVal-VQ**, we found that HPSv3 is the only human preference model trained on images generated after Stable Diffusion 3.5. This suggests that other preference models, such as HPSv2, cannot provide reliable assessments of recent, more advanced generations, as they are **only sensitive to earlier-stage generations that are significantly lower in quality**. In our preliminary experiments, we found that prior preference models tend to treat recent image generations as equally good, failing to capture meaningful quality differences.
>
>
> Overall, tool swaps make the evaluation **highly consistent** with the default configuration: correlations are strong, and the **model ranking remains unchanged**. These results suggest that each tool introduces a (roughly) systematic estimation bias that is applied simultaneously to all models, so while absolute scores may shift, the *relative* ordering of models is stable.

---

> ### Author Response · Authors · 2025-11-24
> **2/n rebuttal**
>
> > W1 [continue]. Tool-dependence and potential evaluator bias: The pipeline fixes Grounding-DINO as the detector, Qwen-2.5-VL as the VLM, and DINOv3/HPSv3 as backbones for CC/VQ. While the authors say the pipeline is tool-agnostic, the reported results (e.g., human agreement advantage) are contingent on this specific stack. Please quantify sensitivity to swapping components (e.g., different detectors/VLMs), or report ablations where each tool is replaced by a strong alternative.
>
> We also emphasize that our **default tool stack** has **high human agreement (81.3%)**. When we switch to more degenerate alternatives—e.g., replacing Grounding DINO with OWL-ViT—even though the ranking is preserved, human agreement drops substantially (from 81.3% to 53.67%). Moreover, as shown in the detector comparison below (where we explicitly cite reported performance for Grounding DINO, OWL-ViT, and GLIP), Grounding DINO is consistently stronger on open-vocabulary detection benchmarks: (Numbers are copied from the paper Grounding DINO 1.5. )
>
> | Method          | Backbone | Pre-training data        | COCO AP_all | LVIS_minival AP_all | LVIS_minival AP_r | LVIS_minival AP_c | LVIS_minival AP_f | LVIS_val AP_all | LVIS_val AP_r | LVIS_val AP_c | LVIS_val AP_f | ODinW35 AP_avg | ODinW13 AP_avg |
> |-----------------|----------|--------------------------|-------------|----------------------|-------------------|-------------------|-------------------|-----------------|---------------|---------------|---------------|----------------|----------------|
> | OWL-ViT         | ViT-L    | O365, OID, VG, LiT       | 42.2        | -                    | -                 | -                 | -                 | 34.6            | 31.2          | -             | -             | -              | -              |
> | GLIP            | Swin-L   | FourODs, GoldG, Cap24M   | 49.8        | 37.3                 | 28.2              | 34.3              | 41.5              | 26.9            | 17.1          | 23.3          | 35.4          | -              | 52.1           |
> | Grounding DINO  | Swin-L   | O365, OID, GoldG         | 52.5        | -                    | -                 | -                 | -                 | -               | -             | -             | -             | 26.1           | 56.9           |
>
> **Does changing Grounding DINO to OWL-ViT or GLIP eliminate failure cases?**
> No. We explicitly re-examine the same failure cases of Grounding DINO **(Figure 17)** under OWL-ViT **(Figure 18)** and GLIP **(Figure 19)**. In these examples, OWL-ViT behaves differently from Grounding DINO but does **not** fix the underlying issue: regardless of whether the object truly exists in the image, OWL-ViT often detects nothing at all. GLIP, on the other hand, exhibits similar false-positive behavior to Grounding DINO, but with denser and less precise bounding boxes (e.g., mis-detecting the tower as “stone gray railing” or “wooden fence”). Thus, switching to OWL-ViT or GLIP does not eliminate these failure modes; it merely turns them into **systematic missed detections** or **more cluttered false positives**.
>
> Taken together, these analyses show that while our framework is flexible enough to support alternative tools, it is still important that any replacement tool should maintain **high agreement with human labels**. Detector accuracy remains crucial: not all tools can be swapped in without significantly degrading alignment with human judgments.

---

> ### Author Response · Authors · 2025-11-24
> **3/n rebuttal**
>
> > W2. Metric coupling between IF and CC is under-analyzed: The geometric mean \sqrt{\text{IF}\cdot \text{CC}} is reasonable, but different applications may value preservation much more than instruction compliance (or vice versa). The choice is referenced to prior work but lacks a robustness study: how often do model rankings flip under other aggregations (harmonic/weighted means)?
>
> Thank you for this thoughtful question. We would like to clarify why we chose the geometric mean as our default aggregation rule.
>
> **Why not the harmonic mean?**
>
> We did consider the harmonic mean but found it overly sensitive to the smaller component in our setting. At Turn 3, $\text{EdiVal-IF}$ typically lies in the range $[0, 50]$, while $\text{EdiVal-CC}$ is usually in $[60, 90]$ (both on a $[0, 100]$ scale). If we define the harmonic mean as
> $H(a,b) = \dfrac{2}{\frac{1}{a} + \frac{1}{b}},$
> then whenever $a$ and $b$ have very different scales, $H(a,b)$ is effectively dominated by the smaller term.
>
> Empirically, when we rank models at Turn 3 using the harmonic mean of $(\text{EdiVal-IF}, \text{EdiVal-CC})$, the ranking is *identical* to ranking purely by $\text{EdiVal-IF}$ (Spearman correlation $= 1$), as shown in the table below. **In other words, $\text{EdiVal-CC}$ has essentially no influence on the ordering if we use the harmonic mean:**
>
> | Model               | Harmonic Rank T3 | EdiVal-IF Rank T3 |
> |---------------------|------------------|-------------------|
> | Seedream 4.0        | 1                | 1                 |
> | Nano Banana         | 3                | 3                 |
> | GPT-Image-1         | 2                | 2                 |
> | Gemini 2.0 Flash    | 5                | 5                 |
> | FLUX.1-Kontext-max  | 4                | 4                 |
> | Qwen-Image-Edit     | 6                | 6                 |
> | Step1X-Edit         | 7                | 7                 |
> | FLUX.1-Kontext-dev  | 8                | 8                 |
> | OmniGen             | 9                | 9                 |
> | AnyEdit             | 10               | 10                |
> | UltraEdit           | 11               | 11                |
> | MagicBrush          | 12               | 12                |
> | IP2P                | 13               | 13                |
>
> Because one of our goals is precisely to *jointly* reflect instruction following and content consistency, we view this behavior as undesirable: the harmonic mean effectively collapses the bi-criteria evaluation back to a single metric.
>
>
> **Geometric vs. arithmetic mean.**
>
> Instead, we use the geometric mean
> $\text{EdiVal-O} = \sqrt{\text{EdiVal-IF} \cdot \text{EdiVal-CC}},$
> which (i) is less sensitive to extreme values than the harmonic mean, and (ii) still penalizes models that perform well on only one axis. This choice also aligns with prior work such as Step1X-Edit~[1], which uses geometric aggregation.
>
> For completeness, we also compare against the arithmetic mean
> $\text{EdiVal-O}_{\text{arith}} = \dfrac{\text{EdiVal-IF} + \text{EdiVal-CC}}{2}.$
> At Turn 3, the rankings induced by geometric vs. arithmetic means are very similar (**Spearman correlation $= 0.96$**), and the top models remain stable:
>
> | Model               | Arithmetic Rank T3 | Geometric Rank T3(ours) |
> |---------------------|--------------------|-------------------|
> | Seedream 4.0        | 1                  | 1                 |
> | Nano Banana         | 2                  | 2                 |
> | GPT-Image-1         | 4                  | 3                 |
> | Gemini 2.0 Flash    | 5                  | 5                 |
> | FLUX.1-Kontext-max  | 3                  | 4                 |
> | Qwen-Image-Edit     | 8                  | 6                 |
> | Step1X-Edit         | 7                  | 7                 |
> | FLUX.1-Kontext-dev  | 6                  | 8                 |
> | OmniGen             | 9                  | 9                 |
> | AnyEdit             | 12                 | 11                |
> | UltraEdit           | 10                 | 10                |
> | MagicBrush          | 11                 | 12                |
> | IP2P                | 13                 | 13                |
>
> Thus, while different aggregation rules (geometric vs. arithmetic) produce slightly different scores, they yield **highly correlated rankings**, and the same group of models (Seedream 4.0, Nano Banana, GPT-Image-1, FLUX.1-Kontext-max) consistently emerges as top performers. We chose the geometric mean as a principled compromise that (a) avoids collapsing to a single axis (as the harmonic mean effectively does here), (b) treats both axes more symmetrically, and (c) is consistent with prior literature [1].
>
> [1] Shiyu Liu et al. *Step1X-Edit: A Practical Framework for General Image Editing.* arXiv:2504.17761, 2025.

---

> ### Author Response · Authors · 2025-11-24
> **4/n rebuttal**
>
> > W2 [continue]. Metric coupling between IF and CC is under-analyzed: The geometric mean \sqrt{\text{IF}\cdot \text{CC}} is reasonable, but different applications may value preservation much more than instruction compliance (or vice versa). The choice is referenced to prior work but lacks a robustness study: how often do model rankings flip under other aggregations (harmonic/weighted means)?
>
> Finally, our framework is explicitly designed to support **user-defined weighting**. Since we report each dimension separately, users are free to post-hoc aggregate these metrics according to their own preferences. Concretely, they can define an **additive (arithmetic)** composite such as
> $\mathrm{EdiVal\text{-}O}= \alpha\cdot\mathrm{EdiVal\text{-}IF}+ \beta\cdot\mathrm{EdiVal\text{-}CC}+ \gamma\cdot \mathrm{EdiVal\text{-}VQ},$
>
> or a **multiplicative (geometric)** composite such as
> $\mathrm{EdiVal\text{-}O}
> = \mathrm{EdiVal\text{-}IF}^{\alpha}
> \cdot \mathrm{EdiVal\text{-}CC}^{\beta}
> \cdot \mathrm{EdiVal\text{-}VQ}^{\gamma},$
>
> where $\alpha,\beta,\gamma \in [0,1]$ and $\alpha + \beta + \gamma = 1$ encode the user’s preferred trade-off between instruction following, consistency, and visual quality. Because we release all three component scores, readers can **easily instantiate any weighted composite** (arithmetic or geometric) and also inspect **per-dimension rankings** if they wish to emphasize one aspect (e.g., reliability vs. fidelity vs. aesthetics).
>
> **Pareto Front Plots across 3 metrics**
>
> Moreover, beyond the choice of aggregation rule, we also provide Pareto-front visualizations to make the trade-offs explicit. In **Section 3.5**, we add Pareto plots in **Figure 10, Figure 14, and Figure 15**, where we analyze the trade-offs between different evaluation dimensions. To ensure that no model “games” the benchmark by excelling in only a single dimension, we plot the Pareto boundary at Turn 3 for all pairwise combinations of our three evaluated dimensions: EdiVal-IF, EdiVal-CC, and EdiVal-VQ (see Figure 10). Additional Pareto plots for Turn 1 and Turn 2 are provided in **Figure 14** and **Figure 15**, respectively.

---

> ### Author Response · Authors · 2025-11-24
> **5/n rebuttal**
>
> > W3. Fairness of latency comparison: The table mixes closed-source UI latency with open-source single-GPU latency, which is not directly comparable (unknown batch sizes, backend scaling, caching). The paper acknowledges measurement contexts but still draws speed–quality conclusions. Consider removing cross-setting latency comparisons or normalizing via a controlled local runtime for open models and API budgeted throughput for closed models.
>
> Thanks for the insightful questions. We have updated the latency numbers in **Table 3** for closed-source models, where we now report **API inference latency**. For open-source models, we use the **default inference settings recommended by each release**, as detailed in **Appendix O.5**. Specifically, for open-source models, we measure latency on a **single A100 80GB GPU** with **batch size = 1**, loading each model using its official recommended setup (links provided in Appendix O.5). Thus, any acceleration techniques (e.g., caching, compilation) that are enabled by default in their official pipelines are implicitly reflected in our measurements.
>
> The updated latency values in Table 3 are shown below. *Italicized* entries correspond to **API-based latency** for closed-source models, while the non-italic entries correspond to **single-GPU latency** for open-source models:
>
> | Model               | Latency (s/img) |
> |---------------------|-----------------|
> | Seedream 4.0        | *14.55*         |
> | Nano Banana         | *9.20*          |
> | GPT-Image-1         | *26.47*         |
> | Gemini 2.0 Flash    | *8.34*          |
> | FLUX.1-Kontext-max  | *10.13*         |
> | Qwen-Image-Edit     | 115.08          |
> | Step1X-Edit         | 20.42           |
> | FLUX.1-Kontext-dev  | 29.21           |
> | OmniGen             | 19.70           |
> | AnyEdit             | 3.93            |
> | UltraEdit           | 3.15            |
> | MagicBrush          | 4.08            |
> | IP2P                | 4.09            |
>
>
>
>
> > W4. Multi-turn vs. single-shot comparison design: The conclusion that multi-turn can help when exposure bias is low is plausible, but details on prompt concatenation for single-shot “complex” edits matter (ordering, connectors, explicit constraints like “make X unchanged”). A stricter prompt engineering baseline for single-shot could narrow gaps; please share exact templates and run ablations with stronger single-shot compositions.
>
> Thank you for the reviewer’s question. We fully agree that the complex-editing prompt template may influence the performance of Complex Editing. As requested, we have added an ablation study in **Section 3.6 (Table 5)** in the revised version.
>
> Concretely, we choose the best open-source model, Qwen-Image-Edit, as our testbed to ensure the ablation is fully reproducible. In our main experiment, we adopt the simplest concatenation strategy: ``{prompt T1}. {prompt T2}. {prompt T3}``.
>
> We then evaluate three alternative variants on Qwen-Image-Edit:
>
> 1. *Random shuffle*: randomly shuffle the three per-turn prompts before concatenation.
> 2. *Sequential connector*: explicitly encode the order as
>    ``First, {prompt T1}, then, {prompt T2}, last, {prompt T3}``.
> 3. *Keep-unchanged objects*: append an explicit constraint,
>    ``{prompt T1}. {prompt T2}. {prompt T3}. Keep {unchanged objects} unchanged.``
>
> **Table 5** reports the resulting complex-editing success rate (\%) at $C = 3$. The numbers are reproduced below:
>
> | Connector Variant    | Complex (C=3) (%) |
> |----------------------|-------------------|
> | Default              | 27.62             |
> | Random shuffle       | 27.10             |
> | Sequential connector | 26.92             |
> | Keep-unchanged       | 25.87             |
>
> These results show that the different compression variants have only a **very mild effect** on performance, suggesting that our main conclusions about complex editing are robust to reasonable choices of prompt template.
>
> **We sincerely hope our responses address your questions/concerns and help increase your confidence in our work. We are happy to offer more clarification if needed.**

---

### Official Review · Reviewer_hVAS · 2025-10-28

**Soundness:** 3
**Presentation:** 3
**Contribution:** 3
**Rating:** 6
**Confidence:** 3

**Summary:**

The paper proposes EdiVal-Agent, an automated, object-centric framework for evaluating instruction-based image editing, including multi-turn scenarios. The agent (i) decomposes an input image into semantically meaningful objects, (ii) generates diverse, context-aware editing instructions across turns while tracking object pools, and (iii) evaluates model outputs with three metrics: EdiVal-IF (instruction following via symbolic checks with open-vocabulary detection plus VLM verification on crops), EdiVal-CC (content consistency for unchanged objects and background with DINO features / L1), and EdiVal-VQ (visual quality via HPSv3). Using this pipeline, the authors build EdiVal-Bench (572 images; 1,716 instructions; 9 types; 3 turns) and benchmark 13 editors spanning diffusion, flow-matching, and in-context AR systems. Key findings include 81.3% human-agreement for EdiVal-IF vs. 75.2% for a strong VLM baseline (Fig. 4, p.6), and clear trade-offs among systems (e.g., Seedream 4.0 best overall EdiVal-O; GPT-Image-1 strong IF but weaker consistency and high latency; Table 3, p.7).

**Strengths:**

Clear problem framing: Multi-turn editing evaluation is under-served; the object-centric perspective addresses spatial grounding, subtle changes, and compositionality (Sec. 1–2).
Methodological novelty: EdiVal-IF combines symbolic checks (presence/absence, spatial relations, counts) with semantic VLM verification on detector-guided crops, improving precision and interpretability (Sec. 2.4; Eqs. (1)–(2), p.5).
Human agreement: 81.3% on IF vs. 75.2% VLM-only; inter-annotator agreement 85.5% bounds achievable performance (Fig. 4, p.6).
Comprehensive benchmark & analysis: 13 editors, 9 instruction types, 3 turns; rich breakdowns (per-task success, consistency via DINO/L1, visual quality HPSv3, exposure drift analysis with luminance p99/p999; Tables 3–4, 8–19; Figs. 5–9, 12–14).
Transparency & reproducibility: Full prompts, thresholds, and pseudo-code (Appendix I), with limitations and failure cases (Appendix E).
Actionable insights: Identifies exposure bias in multi-turn (Fig. 5, p.7; Fig. 12, p.18), weaknesses in spatial/counting tasks (Figs. 6, 15), and model-family trade-offs between beautification vs. preservation (Fig. 3, p.5).

**Weaknesses:**

Detector dependence / error propagation: EdiVal-IF's symbolic branch hinges on Grounding-DINO; false positives/negatives can flip outcomes (failure case in Fig. 11, p.17). A more robust ensemble or confidence-calibrated aggregation could mitigate this.
Limited edit taxonomy: Style change is excluded (Design Scope, p.6), which reduces coverage for prevalent real-world edits; future extensions are suggested but not explored.
Tool ablations & sensitivity: While the agent is "tool-agnostic", results rely on specific choices (Qwen-2.5-VL, DINOv3, HPSv3). A systematic tool swap study (e.g., different open-vocab detectors/VLMs/features) and threshold sensitivity analysis would strengthen claims of generality.
Counting & spatial reasoning remain brittle: Even top systems struggle on count change (<25% at T1) and position change (Table 10, p.29). The paper diagnoses this but does not propose evaluator improvements specialized for numeracy/spatial relation verification beyond center-based rules.
Quality metric choice: HPSv3 is reasonable, but the decision to report separately (not integrated) may complicate a single headline score; some readers may want a tunable composite with explicit weights or Pareto front visualizations.

**Questions:**

Detector robustness: Have you tested alternative detectors (e.g., GLIP, OWL-ViT, recent Grounding-DINO variants) and/or ensembles to reduce false positives observed in Fig. 11? How do EdiVal-IF accuracies shift?
Threshold sensitivity: How sensitive are EdiVal-IF decisions to the 0.3–0.4 detection thresholds and box-size filters (Appendix I.1, p.20)? Could you provide a calibration curve or confidence-weighted IF score?
Style edits: If you added a bounded "style change" category (e.g., limited palette/style banks) with reference-free checks, what metric would you recommend? Any preliminary experiments? (Design Scope, p.6.)
Counting verification: For count change, did you consider tracking-based counting across turns or density estimators to complement detection counts (Table 10 & Appendix L)?
Tool-swap ablation: Can you include a tool-agnostic study (swap VLM, detector, feature backbone) to quantify portability and establish guidance for communities without access to the default stack? (Sec. 2.4 & Appendix I.)
Aggregation design: Why geometric mean for EdiVal-O (p.6)? Did you compare with harmonic mean or a dominance/Pareto analysis to discourage gaming one axis?

**Details Of Ethics Concerns:**

The work focuses on evaluation of editing models. Dataset filtering and safety notes are included (Ethics Statement, p.10; human study notes Appendix K).

---

> ### Author Response · Authors · 2025-11-24
> **1/n rebuttal**
>
> [**Highlight:** We updated the PDF with 3 new sections: **3.5 Pareto Front, 3.6 Ablation Study on Complex Editing Compression, and 3.7 Tool Swap Analysis** on Page **10**. We also add more details on **Page 15-23** in the Appendix.]
>
> We sincerely appreciate the reviewer’s insightful questions and constructive suggestions. The detailed feedback will undoubtedly help us make the revision more solid and comprehensive. Below, we clarify several points raised in the review and present additional experiments to further support our claims. We hope that our responses address the reviewer’s concerns, and we are very grateful again for the careful, thorough, and constructive evaluation of our work.
>
>
> ## Tool Swap Analysis: Detector, VLMs, thresholds, filter, feature extractors
>
> > Q1. Detector robustness: Have you tested alternative detectors (e.g., GLIP, OWL-ViT, recent Grounding-DINO variants) and/or ensembles to reduce false positives observed in Fig. 11? How do EdiVal-IF accuracies shift?
>
> > Q2. Threshold sensitivity: How sensitive are EdiVal-IF decisions to the 0.3–0.4 detection thresholds and box-size filters (Appendix I.1, p.20)?
>
> > Q5. Tool-swap ablation: Can you include a tool-agnostic study (swap VLM, detector, feature backbone) to quantify portability and establish guidance for communities without access to the default stack? (Sec. 2.4 & Appendix I.)
>
>
>
> We thank the reviewer for the insightful question and fully agree that we should analyze how swapping different tools at each component affects the final evaluation. In response, we have added **Section 3.7** to systematically study the behavior of tool swapping.
>
> ### **Tool Swap Analysis**
>
> For the **VLM component**, we replace the default Qwen2.5-7B-VL with three strong alternatives: Qwen2-7B-VL, Qwen2.5-32B-VL, and InternVL3-8B.
>
> For **Grounding DINO’s threshold and large-box filter**, our default setting uses a threshold of $0.30$ for {subject\_remove}, $0.40$ for {position\_change}, and $0.35$ for all other tasks. During detection, we also discard any box whose normalized height *and* width are both larger than $0.98$, in order to filter out degenerate predictions that cover almost the entire image. To investigate sensitivity to these choices, we re-run the evaluation under three alternative configurations:
> 1. A global threshold of $0.30$ for all tasks.
> 2. A global threshold of $0.40$ for all tasks.
> 3. The default task-specific thresholds, but with the large-box filter disabled.
>
> For the **detector**, we swap Grounding DINO (default) with OWL-ViT; for the **image feature extractor**, we swap DINOv3 (default) with DINOv2. We focus on the top four models in our benchmark—Seedream 4.0, Nano Banana, GPT-Image-1, and FLUX.1-Kontext-max—and compute:
> - per-turn instruction-following success rates across all $(\text{model}, \text{turn})$ pairs for VLM / threshold/filter / detector swaps, and
> - consistency across $(\text{model}, \text{turn})$ pairs for the DINOv3 vs. DINOv2 feature swap.
>
> The resulting correlations are summarized below:
>
> | Type      | Default          | Variant         | Metric | Pearson | Spearman |
> |-----------|------------------|-----------------|--------|---------|----------|
> | VLM       | Qwen2.5-7B-VL    | Qwen2-7B-VL     | IF     | 0.9544  | 0.9298   |
> | VLM       | Qwen2.5-7B-VL    | Qwen2.5-32B-VL  | IF     | 0.9790  | 0.9544   |
> | VLM       | Qwen2.5-7B-VL    | InternVL3-8B    | IF     | 0.9660  | 0.9228   |
> | Threshold | [0.3, 0.4]       | Threshold = 0.4 | IF     | 0.9817  | 0.9860   |
> | Threshold | [0.3, 0.4]       | Threshold = 0.3 | IF     | 0.9772  | 0.9457   |
> | Filter    | Has filter       | No filter       | IF     | 0.9982  | 0.9930   |
> | Detector  | Grounding DINO   | OWL-ViT         | IF     | 0.8157  | 0.8029   |
> | Feature   | DINOv3           | DINOv2          | CC     | 0.9987  | 1.0000   |
>
> As for **EdiVal-VQ**, we found that HPSv3 is the only human preference model trained on images generated after Stable Diffusion 3.5. This suggests that other preference models, such as HPSv2, cannot provide reliable assessments of recent, more advanced generations, as they are **only sensitive to earlier-stage generations that are significantly lower in quality**. In our preliminary experiments, we found that prior preference models tend to treat recent image generations as equally good, failing to capture meaningful quality differences.
>
> Overall, tool swaps make the evaluation **highly consistent** with the default configuration: correlations are strong, and the **model ranking remains unchanged**. These results suggest that each tool introduces a (roughly) systematic estimation bias that is applied simultaneously to all models, so while absolute scores may shift, the *relative* ordering of models is stable.

---

> ### Author Response · Authors · 2025-11-24
> **2/n rebuttal**
>
> We also emphasize that our **default tool stack** has **high human agreement (81.3%)**. When we switch to more degenerate alternatives—e.g., replacing Grounding DINO with OWL-ViT—even though the ranking is preserved, human agreement drops substantially (from 81.3% to 53.67%). Moreover, as shown in the detector comparison below (where we explicitly cite reported performance for Grounding DINO, OWL-ViT, and GLIP), Grounding DINO is consistently stronger on open-vocabulary detection benchmarks: (Numbers are copied from the paper Grounding DINO 1.5. )
>
> | Method          | Backbone | Pre-training data        | COCO AP_all | LVIS_minival AP_all | LVIS_minival AP_r | LVIS_minival AP_c | LVIS_minival AP_f | LVIS_val AP_all | LVIS_val AP_r | LVIS_val AP_c | LVIS_val AP_f | ODinW35 AP_avg | ODinW13 AP_avg |
> |-----------------|----------|--------------------------|-------------|----------------------|-------------------|-------------------|-------------------|-----------------|---------------|---------------|---------------|----------------|----------------|
> | OWL-ViT         | ViT-L    | O365, OID, VG, LiT       | 42.2        | -                    | -                 | -                 | -                 | 34.6            | 31.2          | -             | -             | -              | -              |
> | GLIP            | Swin-L   | FourODs, GoldG, Cap24M   | 49.8        | 37.3                 | 28.2              | 34.3              | 41.5              | 26.9            | 17.1          | 23.3          | 35.4          | -              | 52.1           |
> | Grounding DINO  | Swin-L   | O365, OID, GoldG         | 52.5        | -                    | -                 | -                 | -                 | -               | -             | -             | -             | 26.1           | 56.9           |
>
> ### **Does changing Grounding DINO to OWL-ViT or GLIP eliminate failure cases?**
>
> No. We explicitly re-examine the same failure cases of Grounding DINO **(Figure 17)** under OWL-ViT **(Figure 18)** and GLIP **(Figure 19)**. In these examples, OWL-ViT behaves differently from Grounding DINO but does **not** fix the underlying issue: regardless of whether the object truly exists in the image, OWL-ViT often detects nothing at all. GLIP, on the other hand, exhibits similar false-positive behavior to Grounding DINO, but with denser and less precise bounding boxes (e.g., mis-detecting the tower as “stone gray railing” or “wooden fence”). Thus, switching to OWL-ViT or GLIP does not eliminate these failure modes; it merely turns them into **systematic missed detections** or **more cluttered false positives**.
>
> Therefore, we believe that **ensemble methods would not be effective, as the open-set detectors behave similarly and exhibit comparable performance rather than offering complementary strengths.**
>
>
>
> ### **Discussion on confidence-calibrated IF**.
>
>
> We fully agree that **confidence-calibrated aggregation** is are promising direction to further mitigate detector noise. However, in our setting, confidence calibration is non-trivial because each instruction-following decision depends on **multiple detection calls** (involving 2 or more objects), all of which jointly determine the final **symbolic verdict**. Compressing these multiple confidence scores into a single, globally calibrated scalar would require additional modeling assumptions that are not obviously well-justified.
>
> In traditional object detection, confidence-calibrated AP (Average Precision) can be calculated because each detected bbox can correspond to the groundtruth object labels in an image. However, in our setting, we don't have groundtruth labels for each bbox. Instead, the aggregated bboxes' results jointly determine the IF result.
>
> For example, consider a **count-change** instruction such as “Change the count of red apples to 3”:
> - We run Grounding DINO on the **edited image** to detect the “red apple”.
>
> Note that
> 1. We need a **hard threshold** to determine if the read apple exists or not (e.g., bbox_confidence > 0.3). And then we aggregate the result to determine if the number of detected BBoxes is equal to 3.
> 2. With the hard threshold as the criterion, the confidence of the existence of a bbox no longer exists, i.e. it either exists with probability 1 given the threshold or simply does not exist.
>
> This makes it difficult to define a principled, confidence-based calibration scheme for such logical operators.
>
>
>
>
> Taken together, these analyses show that while our framework is flexible enough to support alternative tools, it is still important that any replacement tool should maintain **high agreement with human labels**. Detector accuracy remains crucial: not all tools can be swapped in without significantly degrading alignment with human judgments.

---

> > ### Author Response · Authors · 2025-11-24
> > **3/n rebuttal**
> >
> > ## Style Editing
> >
> > > Q3 – Style edits: If you added a bounded "style change" category (e.g., limited palette/style banks) with reference-free checks, what metric would you recommend? Any preliminary experiments? (Design Scope, p.6.)
> >
> > We appreciate this question and agree that **creative, artistic, style editing is an important direction**. The current benchmark is scoped to tasks where we can define **clear and reproducible criteria** for success (e.g., object presence/absence, attributes, layout, background). Our design choices reflect a trade-off between **coverage** and **reliable automatic evaluation**, rather than a hard conceptual limitation.
> >
> >
> > 1. **Exclusion of style/mood/creative instructions in the current version**
> >
> >
> >
> >
> > We **deliberately exclude** style-transfer, mood-change, and other highly **open-ended instructions** in this version due to evaluation challenges.
> >
> > We provide a preliminary experiment in **Appendix F**, where we generate stylized images (e.g., “Ghibli style” or “Pixar style”) and then ask both GPT-4o and Gemini to classify them. For each generated image, we prompt the VLMs with a yes/no question such as: “Is this image in Ghibli style?” or “Is this image in Pixar style? Please answer only yes or no.” In all cases, both models answered “no”, even for images that humans would generally agree are successful style transfers **(see Fig. 23 and Fig. 24)**. This preliminary experiment highlights a key limitation: current VLMs struggle to reliably judge whether a style-transfer edit has succeeded, even when evaluating images produced by the same model being queried (e.g., GPT-4o answering “no” for its own generations).
> >
> > Beyond this limitation, we identify two additional factors that make style-transfer edits particularly challenging for fair instruction-following evaluation. First, style transfer is an inherently ill-defined and subjective editing category: the boundaries between styles can be vague (e.g., watercolor anime vs Ghibli), which naturally induces lower human agreement. Second, when performing style-transfer edits with closed-source models, we frequently encounter copyright-related refusals (e.g., declining to apply a specific proprietary style), which further complicates systematic evaluation under our framework.
> >
> > In summary, challenges may include the following aspects:
> >
> >    - *Style* does not have clear or discrete boundaries, often leading to **disagreement even among human annotators**.
> >
> >    - Strong VLMs currently **fail to act as reliable style classifiers**, struggling to distinguish between different visual styles.
> >
> >    - Without a robust, widely accepted operational definition of success, **automatic instruction-following scores** for such tasks become noisy and less interpretable.
> >
> >
> >   2. **Not a fundamental limitation, but a current design scope**
> >
> > We do **not** view object-centric evaluation as inherently incompatible with artistic, global, or open-ended editing. Instead:
> >
> > - As the field progresses and the community develops **validated tools** such as reliable **style classifiers** or **scene mood detectors**, these can be seamlessly integrated into our framework.
> > - Many global/creative instructions still involve an **underlying object and region structure** (e.g., *“Apply a watercolor style to the image,”* or *“Make the image look like a vintage film poster.”*). In such cases, an object-centric evaluation is still meaningful, because the edit is still expected to preserve the semantic identity of key objects. These objects can be identified and labeled using LLM-based rewrites or similar tools.
> >
> > For style changes, we propose:
> > - **Disabling background consistency** checks (as background often changes significantly with style).
> > - **Replacing DINOv3** with feature extractors that focus on **semantic identity** rather than low-level appearance—i.e., extractors that can distinguish a dog from a cat, but are tolerant of a real dog vs. a watercolor-rendered dog.
> >
> >
> >
> > **In summary**
> >
> > The current object-centric paradigm is a **deliberate, evaluation-driven choice**, not a fundamental limitation. It provides a **robust foundation** for multi-turn semantic editing and is **designed to evolve** alongside advances in evaluation tools, paving the way for more **reliable** artistic and open-ended editing scenarios in the future.

---

> ### Author Response · Authors · 2025-11-24
> **4/n rebuttal**
>
> ## Counting task
>
> > Q4. Counting verification: For count change, did you consider tracking-based counting across turns or density estimators to complement detection counts (Table 10 & Appendix L)?
>
> We thank the reviewer for the insightful questions. We conduct additional experiments on counting using density maps in **Appendix D**, and we would like to clarify our design choices here.
>
> First, when we generate prompts, the target count is never higher than $10$ by design. Under this regime, Grounding DINO is sufficiently reliable for counting the relevant objects (please see **Figure 20**).
>
> You are absolutely right that there are two other major families of counting methods: **tracking-based counting** and **density-map estimation**.
>
> - **Tracking-based methods** are typically designed for videos, where they exploit multiple frames to track objects over time. In our setting, however, we only have single edited images for counting, so such tracking-based approaches are not applicable.
> - **Density-map estimation methods** (e.g., CSRNet [1,2]) are usually developed for crowd counting of a **fixed** target category (such as humans) on specific datasets, rather than open-vocabulary object counting. When the objects to be counted are not aligned with the training data distribution, these methods can fail dramatically. To make this concrete, we choose an example image containing 3 brown-fur bears and apply CSRNet: CSRNet outputs an estimated count of $44.86$, whereas Grounding DINO correctly predicts $3$ bounding boxes for the target objects. Please see **Figure 20** for this qualitative example.
>
> Overall, while density-based methods are powerful in their intended domain, they are not well-suited for our open-vocabulary, single-image counting setup. In contrast, within our bounded counting range ($\leq 10$), Grounding DINO provides accurate and stable counts for the objects of interest.
>
>
>
> [1] Zhang, Yingying, et al. "Single-image crowd counting via multi-column convolutional neural network." Proceedings of the IEEE conference on computer vision and pattern recognition. 2016.
>
> [2] Li, Yuhong, Xiaofan Zhang, and Deming Chen. "Csrnet: Dilated convolutional neural networks for understanding the highly congested scenes." Proceedings of the IEEE conference on computer vision and pattern recognition. 2018.
>
>
> ## Aggregation design and Pareto front
>
> > Q6. Aggregation design: Why geometric mean for EdiVal-O (p.6)? Did you compare with harmonic mean? Did you compare with the harmonic mean or a dominance/Pareto analysis to discourage gaming one axis?
>
> Thank you for this thoughtful question. We would like to clarify why we chose the geometric mean as our default aggregation rule.
>
>
> ### **Why not the harmonic mean?**
>
> We did consider the harmonic mean, but found it overly sensitive to the smaller component in our setting. At Turn 3, $\text{EdiVal-IF}$ typically lies in the range $[0, 50]$, while $\text{EdiVal-CC}$ is usually in $[60, 90]$ (both on a $[0, 100]$ scale). If we define the harmonic mean as $H(a,b) = \frac{2}{\frac{1}{a} + \frac{1}{b}},$then whenever $a$ and $b$ have very different scales, $H(a,b)$ is effectively dominated by the smaller term. Empirically, when we rank models at Turn 3 using the harmonic mean of $(\text{EdiVal-IF}, \text{EdiVal-CC})$, the ranking is *identical* to ranking purely by $\text{EdiVal-IF}$ (Spearman correlation $= 1$), as shown in the table below. In other words, $\text{EdiVal-CC}$ has no influence on the ordering if we use the harmonic mean:
>
> | Model               | Harmonic Rank T3 | EdiVal-IF Rank T3 |
> |---------------------|------------------|-------------------|
> | Seedream 4.0        | 1                | 1                 |
> | Nano Banana         | 3                | 3                 |
> | GPT-Image-1         | 2                | 2                 |
> | Gemini 2.0 Flash    | 5                | 5                 |
> | FLUX.1-Kontext-max  | 4                | 4                 |
> | Qwen-Image-Edit     | 6                | 6                 |
> | Step1X-Edit         | 7                | 7                 |
> | FLUX.1-Kontext-dev  | 8                | 8                 |
> | OmniGen             | 9                | 9                 |
> | AnyEdit             | 10               | 10                |
> | UltraEdit           | 11               | 11                |
> | MagicBrush          | 12               | 12                |
> | IP2P                | 13               | 13                |
>
> Because one of our goals is precisely to *jointly* reflect instruction following and content consistency, we view this behavior as undesirable: the harmonic mean effectively collapses the bi-criteria evaluation back to a single metric.

---

> ### Author Response · Authors · 2025-11-24
> **5/n rebuttal**
>
> ### **Geometric vs. arithmetic mean.**
>
> Instead, we use the geometric mean
> $$\text{EdiVal-O} = \sqrt{\text{EdiVal-IF} \cdot \text{EdiVal-CC}},$$
> which (i) is less sensitive to extreme values than the harmonic mean, and (ii) still penalizes models that perform well on only one axis. This choice also aligns with prior work such as Step1X-Edit [1], which uses geometric aggregation.
>
> [1] Shiyu Liu et al. *Step1X-Edit: A Practical Framework for General Image Editing.* arXiv:2504.17761, 2025.
>
> For completeness, we also compared against the arithmetic mean
> $$\text{EdiVal-O}_{\text{arith}} = \frac{\text{EdiVal-IF} + \text{EdiVal-CC}}{2}.$$
> At Turn 3, the rankings induced by geometric vs. arithmetic means are very similar (Spearman correlation $= 0.96$), and the top models remain stable:
>
> | Model               | Arithmetic Rank T3 | Geometric Rank T3 |
> |---------------------|--------------------|-------------------|
> | Seedream 4.0        | 1                  | 1                 |
> | Nano Banana         | 2                  | 2                 |
> | GPT-Image-1         | 4                  | 3                 |
> | Gemini 2.0 Flash    | 5                  | 5                 |
> | FLUX.1-Kontext-max  | 3                  | 4                 |
> | Qwen-Image-Edit     | 8                  | 6                 |
> | Step1X-Edit         | 7                  | 7                 |
> | FLUX.1-Kontext-dev  | 6                  | 8                 |
> | OmniGen             | 9                  | 9                 |
> | AnyEdit             | 12                 | 11                |
> | UltraEdit           | 10                 | 10                |
> | MagicBrush          | 11                 | 12                |
> | IP2P                | 13                 | 13                |
>
> Thus, while different aggregation rules (geometric vs. arithmetic) produce slightly different scores, they yield **highly correlated rankings**, and the same group of models (Seedream 4.0, Nano Banana, GPT-Image-1, FLUX.1-Kontext-max) consistently emerges as top performers. We chose the geometric mean as a principled compromise that (a) avoids collapsing to a single axis (as the harmonic mean effectively does here), (b) treats both axes more symmetrically, and (c) is consistent with prior literature.
>
>
>
> ### **Pareto Front Plots across 3 metrics**
>
> Moreover, beyond the choice of aggregation rule, we also provide Pareto-front visualizations to make the trade-offs explicit. In **Section 3.5**, we add Pareto plots in **Figure 10, Figure 14, and Figure 15**, where we analyze the trade-offs between different evaluation dimensions. To ensure that no model “games” the benchmark by excelling in only a single dimension, we plot the Pareto boundary at Turn 3 for all pairwise combinations of our three evaluated dimensions: EdiVal-IF, EdiVal-CC, and EdiVal-VQ (see Figure 10). Additional Pareto plots for Turn 1 and Turn 2 are provided in **Figure 14** and **Figure 15**, respectively.
>
>
>
>
>
>
> **We sincerely hope our responses address your questions/concerns and help increase your confidence in our work. We are happy to offer more clarification if needed.**

---

### Official Review · Reviewer_nxo4 · 2025-10-30

**Soundness:** 3
**Presentation:** 3
**Contribution:** 3
**Rating:** 6
**Confidence:** 4

**Summary:**

This paper tackles a big headache in AI image editing: how do we actually know if multi-turn edit is good? Current methods are flawed, so the authors built EdiVal-Agent, a smart and automated evaluation system. In a nutshell, the agent looks at an image, breaks it down into individual objects, and then generates a series of editing instructions, like a conversation.

It then judges the edited image using three clever new metrics. EdiVal-IF checks if the AI followed the instructions correctly, using a combo of object detectors and vision models. EdiVal-CC checks if the AI messed up the background or other objects it wasn't supposed to touch. Finally, EdiVal-VQ scores the final image's aesthetic quality. Using this agent, they created a benchmark called EdiVal-Bench to test 13 top models, revealing who's best at complex, multi-step edits.

**Strengths:**

1. Object-Centric view, instead of just vaguely looking at the whole picture, the framework focuses on specific objects. This allows it to check with much higher precision whether "the brown horse" was really changed to "a deer" while leaving the other horses alone.
2. Built for Multi-Turn Editing. It's one of the first benchmarks designed specifically for conversational, back-and-forth editing. This is much closer to how people actually use these tools, rather than just giving a single, one-off command.
3. It doesn't just rely on one tool. It smartly combines object detectors with Vision-Language Models and human preference models (for aesthetics). This hybrid approach is more robust and aligns better with human judgment.
4. Fully Automated Pipeline. the agent can create the entire benchmark on its own—from analyzing the image to generating diverse instructions. This makes it super scalable and easy to expand in the future.

**Weaknesses:**

1. This work only support limited instruction types. The agent is a pro at object-based edits (add, remove, change color, etc.), but it can't handle abstract or stylistic requests like "make it more dramatic" or "change the style to cyberpunk." The paper acknowledges this limitation.
2. The entire system is a chain, and it's only as strong as its weakest link. If the object detector (Grounding-DINO) fails to spot an object correctly, the evaluation for that step will be wrong. Its accuracy is capped by the performance of the tools it uses.

**Questions:**

1. The framework is awesome for object-focused edits, but what about creative styles? How would you adapt EdiVal-Agent to reliably evaluate a prompt like "make this photo look like a van Gogh painting," where there isn't a single correct answer?

2. You mentioned that some models suffer from "exposure bias" when editing their own outputs over multiple turns. Do you think this problem could be solved by training these models specifically on multi-turn generative data, or is it a more fundamental flaw in their architecture?

---

> ### Author Response · Authors · 2025-11-24
> **1/n rebuttal**
>
> [**Highlight**: We updated the pdf with 3 new sections: **3.5 Pareto Front, 3.6 Ablation Study on Complex Editing Compression, and 3.7 Tool Swap Analysis** on Page **10**. We also add more details on **Pages 15-23** in the Appendix.]
>
> We thank the reviewer for the careful and constructive review. We genuinely appreciate the sharp insights and find that many of the comments have already helped us improve the paper in the revision. We are also glad that the reviewer acknowledges our contributions, including the object-centric evaluation protocol, the multi-turn benchmark, the rich tool set, and the automatic baselines. Below, we address the reviewer’s concerns and questions; we believe incorporating these discussions into the revised version will make the paper more solid.
>
>
> > **Q1 Creative style edits (e.g., “make this photo look like a van Gogh painting”).**
> > *“The framework is awesome for object-focused edits, but what about creative styles? How would you adapt EdiVal-Agent to reliably evaluate a prompt like ‘make this photo look like a van Gogh painting,’ where there isn't a single correct answer?”*
>
>
> We appreciate this question and agree that **creative, artistic, style editing is an important direction**. The current benchmark is scoped to tasks where we can define **clear and reproducible criteria** for success (e.g., object presence/absence, attributes, layout, background). Our design choices reflect a trade-off between **coverage** and **reliable automatic evaluation**, rather than a hard conceptual limitation.
>
>
> 1. **Exclusion of style/mood/creative instructions in the current version**
>
> We **deliberately exclude** style-transfer, mood-change, and other highly **open-ended instructions** in this version due to evaluation challenges:
>
> - **Style and mood instructions**
>   Our **newly-added Figure 22 and Section F** provide preliminary experiments on style changes, and illustrate why these edits are difficult to evaluate **precisely**:
>   1. *Style* does not have clear or discrete boundaries, often leading to **disagreement even among human annotators**.
>   2. Strong VLMs currently **fail to act as reliable style classifiers**, struggling to distinguish between different visual styles.
>   3. Without a robust, widely accepted operational definition of success, **automatic instruction-following scores** for such tasks become noisy and less interpretable.
>
> - **Creative or ambiguous instruction types**
>   For highly open-ended or ambiguous instructions, we believe our framework can still offer value:
>   - By applying **fine-grained metrics** such as IF (Instruction Following), CC (Content Consistency), and VQ (Visual Quality).
>   - Most creative edits still target identifiable parts of the image. If we think in terms of objects or regions, an LLM can rewrite the instruction to clearly state which object should be changed and how.
>
>
>
> 2. **Not a fundamental limitation, but a current design scope**
>
> We do **not** view object-centric evaluation as inherently incompatible with artistic, global, or open-ended editing. Instead:
>
> - As the field progresses and the community develops **validated tools** such as reliable **style classifiers** or **scene mood detectors**, these can be seamlessly integrated into our framework.
> - Many global/creative instructions still involve an **underlying object and region structure** (e.g., *“Apply a watercolor style to the image,”* or *“Make the image look like a vintage film poster.”*). In such cases, an object-centric evaluation is still meaningful, because the edit is still expected to preserve the semantic identity of key objects. These objects can be identified and labeled using LLM-based rewrites or similar tools.
>
> For style changes, we propose:
> - **Disabling background consistency** checks (as background often changes significantly with style).
> - **Replacing DINOv3** with feature extractors that focus on **semantic identity** rather than low-level appearance—i.e., extractors that can distinguish a dog from a cat, but are tolerant of a real dog vs. a watercolor-rendered dog.
>
>
>
> **In summary**
>
> The current object-centric paradigm is a **deliberate, evaluation-driven choice**, not a fundamental limitation. It provides a **robust foundation** for multi-turn semantic editing and is **designed to evolve** alongside advances in evaluation tools, paving the way for more **reliable** artistic and open-ended editing scenarios in the future.

---

> ### Author Response · Authors · 2025-11-24
> **2/n rebuttal**
>
> > **Q2. Exposure bias in multi-turn editing.**
> > *“You mentioned that some models suffer from ‘exposure bias’ when editing their own outputs over multiple turns. Do you think this problem could be solved by training these models specifically on multi-turn generative data, or is it a more fundamental flaw in their architecture?”*
>
> We sincerely appreciate this thoughtful and incisive question, which directly touches on a core motivation behind our benchmark.
>
> We **do not** view exposure bias in multi-turn editing as a fundamental flaw of the underlying architecture. Instead, our results suggest that **training data and training strategy** play a central role:
>
> - For example, both **FLUX.1-Kontext-max** and **Qwen-Image-Edit** adopt a DiT-style architecture and are trained with flow matching, yet their behavior on our benchmark is quite different:
>   - **Qwen-Image-Edit** exhibits **severe exposure bias**, with performance degrading quickly as it edits its own outputs over multiple turns.
>   - **FLUX.1-Kontext-max**, despite using a similar architecture and learning paradigm, suffers much less from this issue.
>   - Also, we want to point out **Seedream 4.0** (unknown architecture) performs well without suffering from exposure bias, and its previous version 3.0 is flow matching-based.
> - This contrast indicates that the **architecture and base learning algorithm are unlikely to be the main bottlenecks**; instead, data curation, training distribution, and regularization strategies are likely responsible for the observed differences.
>
> We strongly agree with the reviewer that **training on multi-turn generative data** is a promising direction. More broadly, our findings naturally point to several complementary strategies:
>
> 1. **Multi-turn or self-generated training data.**
>    Training on sequences where the model repeatedly edits its *own* outputs can help close the distribution gap between training and evaluation:
>    - Include **self-generated images** in the training data, so the model learns to handle the distribution of its own outputs instead of only clean, single-edit inputs.
>    - Augment the training set with **multi-turn editing trajectories**, either collected from real users or synthesized via scripted pipelines, so that the model explicitly sees multi-turn interaction patterns during training.
>
> 2. **Stronger content-consistency / distribution-preserving objectives.**
>    Our object-centric metrics highlight that some models tend to “drift” away from previously edited or unchanged content over turns. One potential remedy is to:
>    - **Distribution-Level:** Encourage the editor to behave closer to a **distribution-preserving operator**. Let the data distribution be $p(x)$. We would like the editor $f$ to satisfy, at least approximately, the edited images $y$'s distribution $p(y)$,   where $y=f(x)$,   is similar to  $p(x)$, meaning that edited images still lie close to the original data distribution $p(x)$ instead of drifting into out-of-distribution regions.
>    - **Image-Level:** Apply content-consistency losses during the model training, e.g., reducing the pixel-level or feature-level distances of regions that should not be changed before and after an edit.
>    - Such regularization can help ensure that edited images remain near the training data manifold and can reduce cumulative distribution shift across turns, thereby mitigating exposure bias.
>
> In summary:
> - (i) exposure bias appears to be primarily a **mismatch in data and training strategy**, rather than an inherent architectural defect, and
> - (ii) multi-turn training data, self-generated trajectories, and consistency-/distribution-preserving objectives are promising directions to address this issue.
>
>
>
>
>
>
>
> **We sincerely hope our responses address your questions/concerns and help increase your confidence in our work. We are happy to offer more clarification if needed.**

---

### Official Review · Reviewer_FCAS · 2025-11-02

**Soundness:** 3
**Presentation:** 3
**Contribution:** 2
**Rating:** 4
**Confidence:** 4

**Summary:**

This paper introduces EdiVal-Agent, an automated object-centric framework that addresses the unreliable evaluation of multi-turn image editing by first decomposing scenes into objects and then employing a novel hybrid evaluation strategy. This approach strategically combines open-vocabulary detectors for precise symbolic checks and VLMs for semantic verification on localized crops, leading to its core metrics for instruction-following (EdiVal-IF) and content preservation (EdiVal-CC), while separately employing a specialized human preference model for visual quality assessment (EdiVal-VQ) to account for subjective aesthetic judgments. Its primary contributions include the proposed pipeline, the creation of the large-scale EdiVal-Bench for multi-turn editing, and a comprehensive evaluation of 13 editors that provides critical insights—such as diagnosing exposure bias and performance degradation over turns—thereby establishing a new, more reliable, and interpretable standard for the field.

**Strengths:**

1）Automated Evaluation: The proposed EdiVal-Agent introduces a complete, automated pipeline that systematically tackles multi-turn editing evaluation. It decomposes the process into three core stages: image decomposition, instruction generation, and hybrid assessment.
2）Hybrid Assessment Methodology: A tool-specific hybrid strategy is employed for evaluation. This methodology efficiently handles different task types: symbolic tasks (e.g., object detection, counting) are addressed by a specialized object detector; semantic tasks (e.g., color, material changes) are evaluated by a Visual Language Model (VLM) on localized image crops; and overall visual quality is assessed by a dedicated human preference model.
3）Human Alignment: Validation through a substantial human study, comprising 4,576 human judgments, demonstrates the efficacy of the framework. The core metric, EdiVal-IF, achieves an impressive 81.3% agreement with human judgments, significantly outperforming compared baseline methods.
4）Multi-Turn Editing Benchmark: The framework is instantiated as the EdiVal-Bench benchmark. This structured testbed uniquely features multi-turn interaction, an object-centric design, and reference-free evaluation, offering a robust standard for assessing multi-turn editing capabilities.

**Weaknesses:**

1. Limitations in the Benchmark Scale and Data Diversity
--Homogeneous Data Sources: The benchmark relies solely on 572 real-world images. This high level of homogeneity lacks adequate evaluation coverage of synthetic or generated images, restricting its applicability to emerging models and research directions.
--Opaque Scene Coverage: The paper provides no clear specification of image selection criteria, scene distribution characteristics, or quantitative diversity metrics. This lack of transparency makes the benchmark's scene coverage unverifiable.
--Insufficient Complexity for Long-Sequence Tasks: The default 3-turn editing setup is inadequate. It fails to sufficiently test model robustness and coherence in demanding, long-sequence editing scenarios.
--Incomplete Editing Taxonomy: The framework is limited to 9 basic edit types, omitting several essential operations critical for real-world editing, such as object zooming, lighting and environmental adjustments, image cropping, special effects (addition/removal), object duplication, and object rotation.

2. Inherent Biases in Evaluation Methodology
--VLM Dependency Risk: The core image decomposition phase is entirely dependent on proprietary models like GPT-4o. The paper offers no quantitative analysis of how performance would be affected by using different or open-source Visual Language Models, creating significant dependency and reproducibility risks.
--Artificial Framework Advantage: While correctly criticizing the imprecision of VLM-only evaluation, the method's claimed "fine-grained" superiority is derived from a manually engineered evaluation framework, not a fundamental methodological breakthrough that generalizes beyond its custom design.

3. System Architecture and Practicality Deficiencies
--Misleading "Agent" Characterization: The system's fixed workflow and rule-based decision processes fundamentally lack the autonomous reasoning and adaptive capability expected of a true intelligent agent, rendering the "Agent" designation misleading.

4. Superficial Problem Diagnosis and Limited Utility
--Solution Gap: The work identifies existing issues, such as exposure bias, without proposing effective solutions or mitigation strategies, offering a diagnosis but no actionable remedy.
--Limited Guidance Value: The evaluation findings fail to translate into concrete, targeted suggestions for improving model architectures or training protocols, thus diminishing the framework's practical guidance value for future research.

5. Incomplete Evaluation Dimensions
--Blind Spots for Global Editing: The object-centric evaluation paradigm cannot effectively assess crucial non-object-level capabilities, including global attribute editing, style transfer, and overall scene mood alteration.

**Questions:**

1. Data Selection and Representativeness
The paper states that the benchmark contains 572 real-world images. What were the specific, detailed criteria for selecting this dataset? Was consideration given to diversity across essential dimensions such as scenes, lighting conditions, and object categories? Crucially, how can the authors demonstrate that this relatively small dataset is statistically significant and sufficiently representative of the diverse and complex demands of real-world image editing?

2. Blind Spot in Evaluating Generated Images
The benchmark completely excludes synthetically generated images. Given that many advanced editing models are built upon generative techniques, and considering the rapid proliferation of AI-generated content, does the absence of evaluation on this class of images introduce a significant bias in the results? How does this limitation affect the benchmark's long-term relevance and its ability to stress-test state-of-the-art generative models?

3. Editing level
What was the precise rationale for setting the editing depth at 3 turns? Is there experimental evidence demonstrating that three turns are sufficient to expose critical issues in multi-turn interactions, such as error accumulation and state drift, or was this a pragmatic compromise between computational cost and evaluation depth? Furthermore, the current nine editing types do not cover common and fundamental operations like zooming, rotation, or adding special effects. Could this restricted task taxonomy lead to an incomplete or skewed evaluation, potentially misrepresenting the true ranking of model capabilities?

4. Dependency on GPT-4o and Evaluation of Alternatives
The core image decomposition phase relies heavily on the closed-source model, GPT-4o. While the paper claims that other Visual Language Models (VLMs) could serve as alternatives, this assertion lacks supporting data. How stable or robust would the evaluation pipeline outputs be if alternative, open-source VLMs (e.g., Qwen-VL or Llava) were utilized? Does this deep reliance on a single, powerful commercial model introduce the very "evaluation bias" that the proposed framework aims to critique and overcome?

5. Positioning and Autonomy of the "Agent"
The system is named "EdiVal-Agent." However, its described workflow appears to be predefined and strictly rule-driven. To what extent does the system actually possess the autonomous reasoning, dynamic decision-making, and adaptive planning capabilities typically expected of a true intelligent agent, as opposed to functioning as a complex, yet fixed, automation script?

6. Benchmark Evolution and Obsolescence
As a static benchmark, how do the authors plan to mitigate the critical risk of community overfitting? Do they have concrete mechanisms or a roadmap for continuous update and evolution—such as periodically introducing new images, fresh editing types, or adversarial instruction sets—to ensure the benchmark maintains its challenge and long-term relevance?

7. Limitations of the Object-Centric Paradigm
The current "object-centric" paradigm appears inherently limited in its ability to evaluate global attribute editing (e.g., style transfer, mood change) or effectively handle highly ambiguous or creative instructions. Does this imply a fundamental limitation of the framework in assessing more complex and artistic editing tasks? What potential extensions or modifications could be implemented in the future to broaden its scope?

---

> ### Author Response · Authors · 2025-11-24
> **1/n rebuttal**
>
> [Highlight: We updated the PDF with 3 new sections: **3.5 Pareto Front, 3.6 Ablation Study on Complex Editing Compression, and 3.7 Tool Swap Analysis** on Page **10**. We also add more details on **Page 15-23** in the Appendix.]
>
> We thank the reviewer for the careful reading and for recognizing our contribution toward “a new, more reliable, and interpretable standard for the field.” Below, we address each concern point by point.
>
>
> > Highlights of our response:
>
> 1. We do not manually select the images ourselves, as our base images are directly inherited from the widely adopted **GEdit-Bench**.
> 2. The base images include **both AI-generated and real images**, as provided by GEdit-Bench.
> 3. Please refer to **Figures 11, 12, and 13** for various examples of **lighting conditions**, **scenes**, and the distribution across 13 **object categories** in our dataset.
> 4. Feel free to check **Page 10** of our paper PDF, where we have added additional experiments (highlighted in blue), including swapping VLMs, feature extractors, and detectors, as requested by other reviewers.
>
> > Q1. Data Selection and Representativeness
>
> We believe the reviewer has **a misunderstanding** here. The reviewer asked about the selection criterion of our base images.   However, we want to point out that we don't select the images by ourselves, as our base images are inherited directly from **GEdit-Bench**, as stated in the paper.
>
> We choose GEdit-Bench because it is already **a widely used** standard benchmark for **single-turn editing**, and several recent SOTA editing models/technique reports adopt it as their main evaluation protocol, such as Step1X-Edit, Qwen-Image-Edit, OpenGPT-4o-Image, Uniworldv1, Bagel, and OmniGen2. Note that Qwen-Image-Edit is currently the SOTA open-sourced editor.
>
>
> [1] Step1X-Edit: A Practical Framework for General Image Editing.
>
> [2] Qwen-Image-Edit: Qwen-Image Technical Report.
>
> [3] OpenGPT-4o-Image: A Comprehensive Dataset for Advanced Image Generation and Editing.
>
> [4] Uniworld: High-Resolution Semantic Encoders for Unified Visual Understanding and Generation.
>
> [5] Bagel: Emerging Properties in Unified Multimodal Pretraining.
>
> [6] OmniGen2: Exploration to Advanced Multimodal Generation.
>
> ---
>
> All base images can be found here: the  GEdit-Bench's official Huggingface link https://huggingface.co/datasets/stepfun-ai/GEdit-Bench.
>
> **Our dataset could reflect real-world editing scenarios in the following senses:**
>
> 1. **Selection Criteria Implemented by the GEdit-Bench Team:** In their paper, they specify that:
>      - The images are collected from **real user editing requests**, which are designed to support more authentic evaluation of image editing models.
>       - The images contain **both AI-generated and real images**. In other words, **the phrase “real-world images” in our paper refers to the *source of user editing requests* rather than the exclusion of synthetic imagery**. See Fig. 1 (two horses) and the last figure in the Appendix (little witch girl) for AI-generated image examples.
>       - The team carefully curated the dataset to reflect real-world editing needs and **ensure diversity in editing scenarios**, as stated in their paper's section 4.1.
>
> 2. **Diversity in scenes:** Please check out our newly-added **Figure 11** in the paper, which covers assorted environments and content types, including but not limited to indoor/outdoor, person, special items, and artistic scenes. This diversity helps our dataset to better reflect the real-world user cases.
>
> 3.  **Diversity in lightning:** Please check out our newly-added **Figure 12** in the paper for image examples of different lightning sorted from very dark to very bright. Our dataset includes a variety of illumination scenarios that aim to cover the spectrum of very bright to very dark environments. This variation reflects the real-world complex editing scenes.
>
> 4.  **Diversity in object categories:**  After decomposition, we analyzed and found that there are a total of **724 distinct objects** in the dataset. We classify them into 13 categories, ranging from everyday items such as furniture and kitchenware to entities like vehicles and animals. Specifically, they are  Clothing & Accessories, Furniture & Home, Food & Kitchen, Structures & Places, Vehicles & Transport, Nature & Outdoors, Animals, Sports, Toys & Hobby, Stationery & Office, Electronics & Tech, Body & Medical, Tools & Industrial, and Other. The distribution of these objects is presented in the newly-added **Fig. 13**. Please check **all the figures** in our paper, including the two mentioned above, to get a qualitative sense of the object diversity in our dataset.

---

> ### Author Response · Authors · 2025-11-24
> **2/n rebuttal**
>
> 5. **Size of our dataset.**
>
>       a. Since our images are directly inherited from GEdit-Bench, our **single-turn dataset** is comparable to GEdit's for single-turn experiments. Therefore, we believe it is sufficient.
>
>       b. Representativeness in image editing depends not only on dataset size but also on the **coverage of editing types and visual scenarios**. EdiVal-Agent is designed to capture a variety of realistic situations.
>
>      - A **broad taxonomy of edit types** (9 in total), covering object-level operations (add/remove/replace, color/material, position/count, text) and global changes (background change).
>      - A mixture of **foreground-focused** and **background-focused** edits, and edits that explicitly test **content preservation** (unchanged objects and regions).
>      - Base images inherited from **real-world user editing requests**, which contain a variety of scenes (portraits, objects, indoor/outdoor, products, etc.) as mentioned above.
>
> Above all, we believe that our dataset reflects real-world editing scenarios through **its authentic user editing requests, diverse image sources (including both real and AI-generated images), and wide coverage of scenes, lighting conditions, and object categories. With a rich taxonomy of edit types and a thoughtful design that balances foreground and background edits,** EdiVal-Agent offers strong representativeness for image editing tasks, both in quality and scope.
>
>
>
> > Q2. Blind Spot on synthetic images
>
>
> We thank the reviewer for raising this important point on evaluating synthetic images, but we believe there is **another misunderstanding**, as explained below:
>
> 1. **Synthetic images are already included in our base dataset.**
>    The base images in EdiVal-Bench are **directly inherited from GEdit-Bench**, which is built from **real-user editing requests** and already contains a mix of **natural photos and AI-generated content**: See Fig. 1 (two horses) and the last Figure (little witch girl) for AI-generated image examples.
>
> 2. **Multi-turn editing naturally involves synthetic images.** In multi-turn settings **(when turn > 1)**, each new edit is applied to the AI-edited output from the previous turn, meaning that subsequent inputs are synthetic by nature.
>
> 2. **The current benchmark is sufficient to evaluate SOTA editing models, and EdiVal is designed to be future-proof.** We believe the current setup is already capable of challenging strong models—for example, Nano Banana achieves only **56.24%** on our T3 EdiVal-O evaluation. Furthermore, EdiVal is a fully automated pipeline that takes a set of base images (from any domain, including real or synthetic) as input. **Based on that, it can generate benchmarks tailored to any future test scenarios.** The current setup presented in the paper should be seen as just **one instantiation of the framework**. Since the entire process is automated, creating new benchmarks requires minimal manual effort.
>
>
> > Q3. Editing Level: why choose max_turn = 3?
>
> Our choice of 3 turns is motivated by **both empirical evidence and practical considerations**:
>
> 1. **Empirical Evidence: 3 turns already expose critical issues and interesting findings.**
> - **The best editor is still far from saturation at 3 editing turns.**
>   As shown in **Table 3**, the top-performing model, Seedream 4.0, achieves only ~42% multi-turn instruction-following success at turn 3. Other models perform substantially worse. This indicates that even at 3 turns, the task remains highly challenging and reveals clear performance gaps.
>
> - **Clear failure modes emerge within 3 turns.**
>   Both **Figure 1** and **Figure 5** reveal exposure bias failure modes in models such as Qwen-Image-Edit, where marginal success rates degrade sharply across turns. In addition, GPT-Image-1 performs poorly in content consistency while Nano Banana lags slightly behind other strong closed-sourced models in instruction following.
>
> - **Multi-turn editing vs. single-shot complex editing.**
>   In models capable of in-context editing (e.g., Nano Banana), multi-turn editing yields **higher success rates** (35.35% vs. 28.14%) than single-shot complex editing, aligning with a step-by-step “chain of edits” paradigm—analogous to chain-of-thought reasoning.
>
> - **Performance trends can be inferred from marginal behavior.**
>   As shown in **Figure 5**, closed-source models maintain relatively stable marginal success rates across turns, suggesting similar behavior for longer chains. In contrast, flow-matching models (e.g., Qwen-Image-Edit, Step1x-Edit) show sharp performance drops, which are likely to continue.
>   This pattern also extends to content consistency and visual quality—e.g., the **Flux-series** model maintains strong consistency across all turns.
>
> In short, 3 turns are already sufficient to expose **multi-turn pathologies**—longer horizons are not needed to surface core weaknesses.

---

> > ### Author Response · Authors · 2025-11-24
> > **3/n rebuttal**
> >
> > 2. **Practical Considerations: A trade-off between evaluation depth and computational/monetary cost.**
> > - A 3-turn chain in our benchmark involves roughly **1.7k editor generation calls**, resulting in significant compute time. For example, Qwen-Image-Edit takes **~28 hours** to complete this on a single A100 GPU (without advanced parallelism). The final evaluation stage takes **~2.5 hours**.
> >
> > - Running significantly longer chains would be **infeasible for a community benchmark**, where users may run evaluations multiple times during model optimization or adaptation.
> >
> > - For commercial/API-based models, the **monetary cost** become substantial. For example, generating 1.7k edits with GPT-Image-1 costs **~$600**, accounting for failed API retries.
> >
> >
> >
> > We therefore chose 3 turns as a **practical default**: deep enough to reveal multi-turn failure modes, while keeping the evaluation affordable and reproducible.
> >
> > 3. **Not a limitation of the framework.**
> >   - The choice of 3 turns is **not a hard constraint** of EdiVal-Agent.  The object pool management, instruction generation loop, and evaluation metrics **naturally scale** to more turns by simply increasing the `MAX_TURN` hyperparameter.
> >
> >    - As editing models improve and computational budgets grow, users can seamlessly extend our framework to **5, 6, or more turns** without any structural change.
> >
> >
> > In summary, the 3-turn setting offers a **balanced choice**: it exposes key model limitations while keeping evaluation feasible for widespread adoption. Crucially, the underlying methodology generalizes to longer interaction chains, allowing future work to push the boundaries of multi-turn image editing as the field progresses.
> >
> >
> > > Q4. Dependency on GPT-4o: decomposition phase relies heavily on the closed-source model, GPT-4o
> >
> >
> >
> >
> > We chose GPT-4o for **its superior visual understanding and creative writing capabilities** compared to available open-source alternatives.
> >
> > We do not consider this a bias, but rather a reflection of **model superiority**. In our case, the goal of **decomposition** stage is not to ensure fairness across all models, but to ensure **precision and reliability** in breaking down image content. According to benchmarks published on OpenAI's announcement https://openai.com/index/hello-gpt-4o/, GPT-4o outperforms models like LLaMA 3 (400B) in multimodal understanding and generation, especially in tasks requiring fine-grained visual understanding **(decomposition stage)** and creative instruction synthesis **(instruction generation stage)**. These strengths make GPT-4o particularly well-suited to our setting, which demands **accurate** interpretation of complex image components and **diverse** editing goals.
> >
> > Beyond GPT-4o, we conducted a **tool swap analysis** for other components used in our system, including different VLMs, object detectors, and feature extractors. Please refer to **page 10 of our updated PDF** for detailed results. In summary, variants of our toolchain configurations exhibit strong correlations (In most cases, Pearson, Spearman >0.95) with our default setup, demonstrating the robustness and generalizability of our agentic system.
> >
> >
> > > Q5. Positioning and Autonomy of the “Agent"
> >
> >
> > We appreciate this question and agree that clarifying the role and capabilities of **“EdiVal-Agent”** is important.
> >
> >
> > 1. **What we mean by “agent” in this work**
> > We use *agent* to refer to a system that:
> >    - **State**: Maintains three object pools containing all/unchanged/available objects and their associated attributes within an image.
> >    - **Action**: Generates an appropriate instruction by analyzing the current state—for example, identifying which objects are available to edit at this turn and reasoning about what instruction is suitable for editing that object.
> >    - **State Transition**: Dynamically updates the object pools after each action.
> >
> > Concretely, EdiVal-Agent is not a static script that always emits the same sequence of instructions.   Its decisions depend on the *current* image content and edit history, rather than being derived from a fixed, image-independent template.
> >
> > 2. **Where the system is rule-driven and where it is adaptive**
> > We agree that the **high-level control loop** is predefined: the agent always follows the same structure of  “decompose → prompt generation → evaluation”.  This is by design, to ensure **reproducibility and controllability** in a benchmark setting.  However, within this loop:
> >    - The selected **objects**, **relations**, and **concrete instructions** vary across images and turns, guided by LLM-based reasoning over the current object pools.
> >    - The **evaluation queries** are dynamically constructed based on the updated object pools and prior instructions—they are not hard-coded per image.
> >
> > Thus, EdiVal-Agent combines **rule-based orchestration** (for high-level workflow control) with **LLM-driven reasoning and tool use** (for scene understanding and instruction generation).

---

> ### Author Response · Authors · 2025-11-24
> **4/n rebuttal**
>
> 3. **Scope of autonomy and claims**
> We do **not** claim that EdiVal-Agent is a fully general or open-ended intelligent agent. Its autonomy is scoped to our specific task:
>    - It **autonomously proposes** multi-turn edit instructions conditioned on the input image and edit history.
>    - It autonomously **constructs evaluation queries** to assess instruction following, content consistency, and visual quality.
>
> The goal is to build a **reliable, object-centric evaluation agent/workflow**, not a general-purpose agent. We will make this more explicit in the revision to prevent potential misunderstandings.
>
> 4. **Independence of technical contributions from naming**
> Regardless of terminology, the core technical contributions are:
>    - The **stateful object pools** maintained across turns,
>    - The **multi-turn instruction generation** conditioned on that state,
>    - And the **fine-grained evaluation metrics** (*EdiVal-IF*, *EdiVal-CC*, *EdiVal-VQ*) built upon this agentic workflow, which are tailored for multi-turn editing evaluation.
>
> **In summary**, EdiVal-Agent is best viewed as a **stateful, LLM- and tool-orchestrated evaluation agent with a predefined control loop**, rather than a fixed script or a fully general intelligent agent. If the reviewer agrees with this positioning, we will clarify it explicitly in the revised manuscript.
>
>
> > Q6. Benchmark Evolution and Obsolescence: mitigating overfitting, roadmap for continuous update and evolution, and maintaining long-term relevance and its challenges
>
>
> We fully agree that benchmark obsolescence and overfitting are important concerns. Our design explicitly separates **(i) a concrete benchmark instantiation (EdiVal-Bench)** from **(ii) a general, automatic evaluation pipeline (EdiVal-Agent)**, with the **latter** intended as the main vehicle for **long-term evolution**.
>
> 1. **Automated Pipeline vs. Instantiated Benchmark.**
>    EdiVal-Agent is a **fully automatic pipeline** that takes as input: a set of base images (from any domain, including real and synthetic) and a set of edit types (with success criteria), and produces:  Multi-turn instruction chains, and Object-centric evaluation metrics (EdiVal-IF/CC/VQ).
>
>    The **EdiVal-Bench**  in the paper should  **not** be viewed as the only possible instantiation. Because the whole process is automated, new benchmarks can be generated with minimal manual effort.
>
> 2. **Concrete mechanisms for evolution.**
>    We envision several straightforward ways to keep the benchmark challenging and up to date:
>    - **New base images.** Researchers can add images from new domains (e.g., higher-resolution product photos, stylized AI art, medical or scientific imagery where appropriate) and run the same pipeline to obtain new multi-turn scenarios.
>    - **New random seeds/instruction sets.** For the same image pool, we can re-run EdiVal-Agent with different random seeds to produce **fresh instruction chains**, effectively generating new test splits without redesigning the framework.
>    - **Longer horizons.** As models become stronger and evaluation budgets grow, the number of turns can be increased (i.e., setting the hyperparameter MAX_TURN to a larger number), directly stressing longer-horizon consistency while reusing the same object-centric machinery.
>    - **Additional edit types.** When reliable evaluators for new operations (e.g., style classifiers) become available, they can be added as new edit types by specifying prompt templates and success rules, without changing the core pipeline.
>
> 3. **Mitigating overfitting in practice.**
>    While directly memorizing thousands of (image, instruction) triples is unlikely to be a realistic strategy for large editing models, we still want to discourage benchmark-specific tuning. The automatic nature of EdiVal-Agent allows:
>    - Periodic release of **new benchmark versions** with expanded or shifted image pools and instruction seeds.
>    - Community-maintained **hidden test suites** (e.g., held-out images/instructions) that are generated by the same pipeline but not publicly released, enabling unbiased reporting on leaderboards.
>
> 4. **Roadmap in the paper.**
> We thank the reviewer for raising this point. We would like to:
>    - Position EdiVal-Bench as the **first instantiation** of a fully automated evaluation pipeline.
>    - Add a short subsection in the discussion section outlining this **evolution roadmap** (new images, new instruction seeds, longer horizons, and additional edit types) as mentioned above.
>
> If the reviewer agrees with the readmap mentioned above, we promise to incorporate these clarifications in the revision.
>
> In summary, EdiVal-Agent is designed precisely to avoid being a one-off static benchmark: it is an **automatic, extensible evaluation pipeline** that can continuously generate new testbeds as models and use cases evolve, thereby mitigating overfitting and maintaining long-term relevance.

---

> ### Author Response · Authors · 2025-11-24
> **5/n rebuttal**
>
> > Q7. Limitations of the Object-Centric Paradigm
>
>
> We appreciate this question and agree that **complex, artistic editing is an important direction**. The current benchmark is scoped to tasks where we can define **clear and reproducible criteria** for success (e.g., object presence/absence, attributes, layout, background). Our design choices reflect a trade-off between **coverage** and **reliable automatic evaluation**, rather than a hard conceptual limitation.
>
>
>
> 1. **Our framework already includes global changes**
>
> The `background_change` edit type explicitly targets **global scene modifications**, such as changes in environment, season, or overall background. In our design, when a `background_change` edit occurs at turn *t*, **background-consistency scoring is disabled** from that turn onward, while all other evaluation metrics remain applicable.
>
> The present benchmark emphasizes **object- and background-level semantic changes** because these are the types of edits for which we can currently guarantee **high evaluation reliability** without relying on human annotation.
>
>
>
> 2. **Exclusion of style/mood/creative instructions in the current version**
>
> We **deliberately exclude** style-transfer, mood-change, and other highly **open-ended instructions** in this version due to evaluation challenges:
>
> - **Style and mood instructions**
>   Our **newly-added Figure 22 and Section F** provide preliminary experiments on style changes, and illustrate why these edits are difficult to evaluate **precisely**:
>   1. *Style* does not have clear or discrete boundaries, often leading to disagreement even among human annotators.
>   2. Strong VLMs currently **fail to act as reliable style classifiers**, struggling to distinguish between different visual styles.
>   3. Without a robust, widely accepted operational definition of success, **automatic instruction-following scores** for such tasks become noisy and less interpretable.
>
> - **Creative or ambiguous instruction types**
>   For highly open-ended or ambiguous instructions, we believe our framework can still offer value:
>   - By applying **fine-grained metrics** such as IF (Instruction Following), CC (Content Consistency), and VQ (Visual Quality).
>   - By leveraging **object-centric perspectives**, since many creative edits still involve identifiable objects or regions, which can be extracted using LLM-based rewriters.
>
>
>
> 3. **Not a fundamental limitation, but a current design scope**
>
> We do **not** view object-centric evaluation as inherently incompatible with artistic, global, or open-ended editing. Instead:
>
> - As the field progresses and the community develops **validated tools** such as reliable **style classifiers** or **scene mood detectors**, these can be seamlessly integrated into our framework.
> - Many global/creative instructions still involve an **underlying object and region structure** (e.g., *“Apply a watercolor style to the image,”* or *“Make the image look like a vintage film poster.”*). In such cases, **object-centric evaluation remains meaningful**, as the edit should preserve the **semantic identity of objects**, which can be extracted through LLM rewrites or similar tools.
>
> For style changes, we propose:
> - **Disabling background consistency** checks (as background often changes significantly with style).
> - **Replacing DINOv3** with feature extractors that focus on **semantic identity** rather than low-level appearance—i.e., extractors that can distinguish a dog from a cat, but are tolerant of a real dog vs. a watercolor-rendered dog.
>
>
>
> **In summary**
>
> The current object-centric paradigm is a **deliberate, evaluation-driven choice**, not a fundamental limitation. It provides a **robust foundation** for multi-turn semantic editing and is **designed to evolve** alongside advances in evaluation tools, paving the way for more artistic and open-ended editing scenarios in the future.
>
>
>
>
> **We thank the reviewer for their comprehensive and insightful feedback. We hope that the newly added experiments and clarifications greatly address the reviewer’s concerns. If so, we would be grateful if the reviewer could consider increasing their score.**

---

### Author Response · Authors · 2025-11-27
**Dear Reviewers**

Dear Reviewers,

We would appreciate any feedback on our rebuttal, if there is any.

Best,

Authors

---

### Author Response · Authors · 2025-11-28
**Revision summary for reviewers**

Dear PCs, SACs, ACs, and Reviewers,

We sincerely appreciate the constructive and thoughtful feedback provided during the review and rebuttal process. Given that only **four days** remain in the discussion period, we have made a set of focused revisions to strengthen the paper. The changes in the revised submission are highlighted in blue.

The main updates are as follows:

- **Section 3.5 & Appendix E:** Prompt compression ablation
- **Section 3.6 & Appendix B:** Metric aggregation and Pareto analysis
- **Section 3.7, Appendix B & Appendix C:** Tool-swap analysis for the VLM, detector, detector thresholds / large-box filtering, and image feature backbone
- **Appendix A:** Dataset coverage and quantitative analysis
- **Appendix C:** Detector swap comparison and qualitative failure cases
- **Appendix D:** Preliminary experiments on counting tasks
- **Appendix F:** Preliminary experiments on style transfer

Given the limited remaining discussion window, we hope these revisions address your main concerns. We are, of course, very happy to continue the discussion and clarify any remaining issues.

Best regards,
Authors

---

### Meta-Review · Area_Chair_Jqox · 2025-12-10

**Summary:**

This paper proposes EdiVal-Agent for automatically evaluating the performance of instruction-based image editing. This framework targets an object-centric perspective and validates the editing performance in three steps. It first decomposes the input image into several objects and then generates multiple editing instructions. Subsequently, it evaluates the editing results using three new metrics: EdiVal-IF, EdiVal-CC, and EdiVal-VQ, which assess instruction following, content consistency, and overall editing quality, respectively. The authors also propose a benchmark named EdiVal-Bench to test prevailing editing models to evaluate their multi-turn editing performance.

The paper received initial scores of 6, 6, 6, and 4. Most reviewers agreed with the contribution of the proposed automatic pipeline for multi-turn editing. Reviewers expressed concerns about the representativeness of the constructed benchmark, as well as the robustness and sensitivity of different components within the pipeline and the newly introduced evaluation metrics. During the rebuttal, the authors included more details about the benchmark illustration and provided a detailed analysis of the tool swap experiments across the whole pipeline, together with explanations of the designs of the three new metrics.

Overall, the proposed EdiVal-Agent and EdiVal-Bench represent a good contribution to the development of generative AI, especially in the field of multi-turn image editing. Therefore, I recommend acceptance.

**Reviewer Concerns:**

The main concerns raised by the reviewers can be categorized as follows, for which the authors have provided additional results and explanations for clarification.

- The Representativeness of the Proposed Benchmark

    Reviewer FCAS expressed concerns regarding the scale and diversity of the proposed EdiVal-Bench, encompassing data sources, scene coverage, and complexity for long-sequence tasks. The authors addressed this by explaining the original image sources of the benchmark and demonstrating data diversity through new figures in the paper. Moreover, they justified the choice of the maximum number of turns for multi-turn editing tasks as adequate for evaluating current editing methods and practicality. I believe that these concerns can be addressed during the rebuttal with more detailed visualizations and explanations.

- The Robustness and Sensitivity of Components in EdiVal-Agent

    All reviewers expressed concerns about the robustness or sensitivity of each component in the chained evaluation pipeline. The authors have supplemented with additional ablation studies, such as swapping key components and exploring various thresholds and filtering mechanisms to analyze the impact of each module. Their results indicate that alternative configurations yield outcomes consistent with the default setup. However, as noted by Reviewer nxo4 and demonstrated in the appendix, a failure of the object detector (Grounding-DINO in this paper ) to correctly identify an object will lead to an erroneous evaluation for that step. The authors acknowledge this limitation and suggest that the fidelity of EdiVal-Agent could be further improved by employing more reliable open-vocabulary detection models.

- Scope of Editing Types

    Reviewers nxo4 and hVAS raised questions regarding the scope of editing types covered, noting that the proposed EdiVal-Agent is designed for object-centric evaluation and excluded the style change. The authors acknowledge that evaluating style transfer is a challenge. However, they contend that the object-centric framework is extensible and can potentially accommodate more global or open-ended editing instructions as the field advances. Having reviewed the paper and the full discussion, I find that while the current version of EdiVal-Agent has limitations in evaluating open-ended instructions especially for style-editing tasks, this does not diminish the core contribution of the work. Future integration of reliable style classifiers or scene aesthetics detectors could address this specific limitation.

- The reliability of newly proposed metrics

    Reviewer hVAS and TgZ8 raised specific questions regarding the proposed metrics. Reviewer hVAS requested a Pareto analysis, while Reviewer TgZ8 raised concerns about the potential coupling between EdiVal-IF and EdiVal-CC. Additionally, Reviewer TgZ8 suggested that the latency comparison results be reorganized to ensure a fair assessment. In response, the authors have  updated the latency numbers for closed-source models, provided a Pareto front analysis in the revised paper and detailed their rationale for selecting the geometric mean as the default aggregation rule. I believe that with these additional explanations and analyses, the concerns pertaining to the newly introduced metrics have been adequately addressed.

**Reviewer Scores:**

After reading the initial reviews of all reviewers and the rebuttal results provided by the authors, I think Reviewer nxo4, hVAS and TgZ8 will maintain their positive scores as their questions are ansered by the detailed added illustration. The review from Reviewer FCAS appear somewhat lengthy and disorganized. But I anticipate that, after carefully reviewing the paper and the rebuttal provided by the authors, FCAS would also consider adjusting the score accordingly.

---

### Decision · Program_Chairs · 2026-01-26

Accept (Poster)